# Training-Free Modality-Agnostic Concept Sliders: Fine-Grained Control via Diffusion Models of Images, Audio, and Video

## Abstract

Diffusion models have become state-of-the-art generative models for images, audio, and video, yet enabling *fine-grained controllable generation*, i.e., continuously steering specific concepts without disturbing unrelated content, remains challenging. Concept Sliders (CS) offer a promising direction by discovering semantic directions through textual contrasts, but they require per-concept training and architecture-specific fine-tuning (e.g., LoRA), limiting scalability to new modalities. In this work, we introduce a simple yet effective approach that is fully *training-free* and *modality-agnostic*, achieved by partially estimating the CS formula during inference. To support modality-agnostic evaluation, we extend the CS benchmark to include both video and audio, establishing the first suite for fine-grained concept generation control with multiple modalities. We further propose three evaluation properties along with new metrics to improve evaluation quality. Finally, we identify an open problem of scale selection and non-linear traversals and introduce a two-stage procedure that automatically detects saturation points and reparameterizes traversal for perceptually uniform, semantically meaningful edits. Extensive experiments demonstrate that our method enables plug-and-play, training-free concept control across modalities, improves over existing baselines, and establishes new tools for principled controllable generation.

## 1 Introduction

Diffusion models have emerged as state-of-the-art generative models, capable of producing realistic and diverse outputs across images, audio, and video (Rombach et al., 2022; Yang et al., 2023; Ho et al., 2022). Beyond generating high-quality samples, a central task is *controllable generation*, the ability to steer the generative process along user-specified signals (Liu et al., 2023; Ho et al., 2022). In particular, *text-to-x*, where x is a certain modality, has emerged as a powerful control signal for generative models, offering an intuitive human interface and enabling semantically aligned control (Zhang et al., 2023a;b). This text-guided capability plays a central role in creative applications, allowing users to produce high-quality content without requiring technical knowledge or professional design skills. An abundance of methods have been proposed to enable flexible, accurate generation and editing of creative content (Gal et al., 2023a; Chiu et al., 2025; Gaintseva et al., 2025). However, relying solely on text prompts makes it difficult to precisely edit specific attributes and **modulate them continuously** without affecting unrelated components, for example, adjusting a person's age without altering their identity, or changing weather intensity without modifying the scene location. This limitation restricts creators' ability to fully realize their intended specific edits.

To enable **fine-grained, text-driven control** of specific attributes, Gandikota et al. (2024) introduced *Concept Sliders* (CS), which derives semantic directions from textual contrasts between opposing concepts to target localized edits (e.g., adjusting a person's apparent age). However, CS *requires per-concept training*, which is impractical in interactive settings where users may request diverse, unpredictable concepts that cannot be pre-trained. Moreover, CS and others such as (Sridhar & Vasconcelos, 2024; Choi et al., 2021) often depend on external classifiers or architecture-specific fine-tuning (e.g., LoRA), which may not be available for new backbones, may become infeasible for future models, or may demand deep, model-specific integration. These constraints hinder adoption in *new modalities* (video, audio) and in rapidly evolving open-source models, motivating *architecture- and modality-agnostic, training-free* methods that retain fine-grained control without per-concept training or architectural modifications.

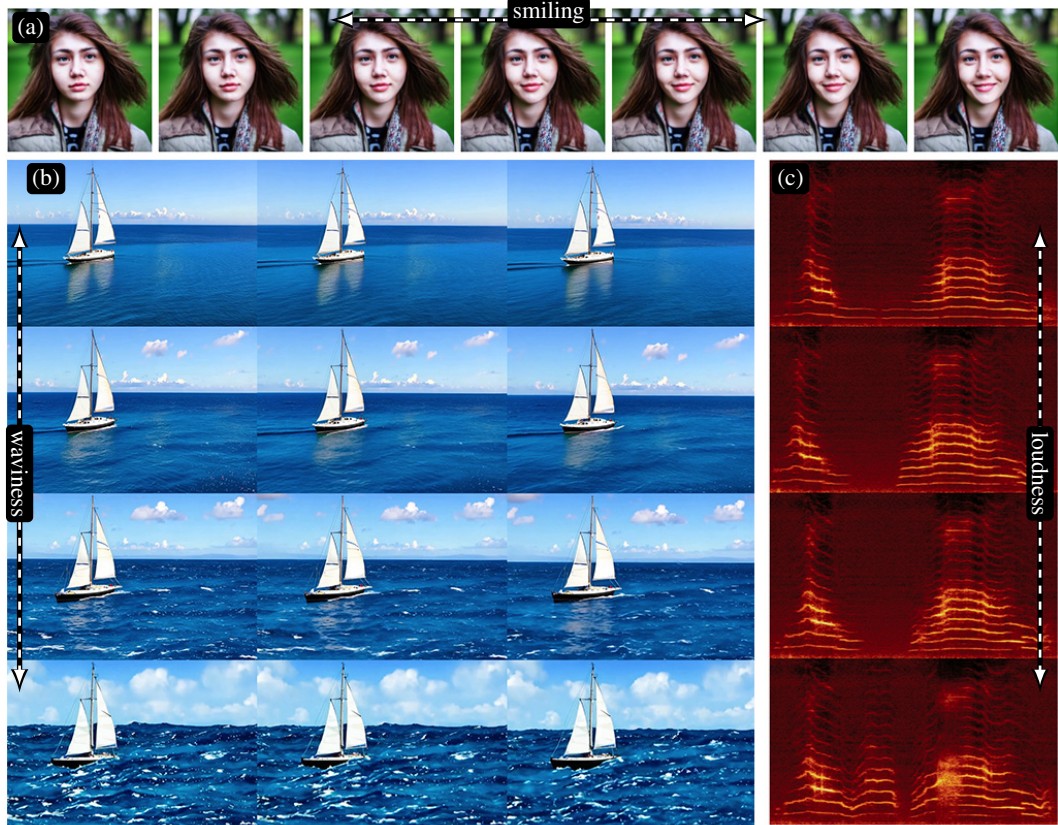

Figure 1: **Sliders:** (a) image: gradually increasing/decreasing smile intensity; (b) video: progressively increasing ocean waviness in a sailing scene; (c) audio: spectrograms of a cat's meow with rising energy in dominant frequencies from top to bottom, indicating successful concept control.

In this work, we propose a simple and elegant CS-inspired solution that is **training-free and can be seamlessly adapted to new modalities and architectures.** Rather than training a model to directly predict the CS update, we partially estimate the CS formula during inference. This design decouples the method from particular model architecture or modality, thereby enabling plug-and-play concept control method. While our approach is theoretically applicable for any modality, current public benchmarks focus primarily on images. To address this gap, we fundamentally extend the Gandikota et al. (2024) image benchmark to include standardized *image*, *video* and *audio* concept editing, constituting, to the best of our knowledge, the first modality-agnostic benchmarking for this task. We hope that this benchmark will help drive the development of architecture or modality-agnostic and training-free methods for fine-grained concept manipulation. Finally, on the extended benchmark, our extensive experiments demonstrate that the proposed simple solution **enables fully training-free concept controlling while achieving improved results**, albeit with a modest increase in computational cost during inference.

Extending the benchmark to multiple modalities raises non-trivial evaluation questions: How should we measure *fine-grained* control, and can a unified protocol work across modalities? Revisiting work by Gandikota et al. (2024), we note that CS evaluations rely largely on ΔCLIP; however, we find that this metric often misaligns with human perception, and its interpretability, i.e., clarifying *why* a slider is 'good', can be vague, especially for users without technical experience. We therefore introduce three **modality-agnostic properties** that an effective slider should satisfy: *range* (extent of controllable variation), *smoothness* (consistency and monotonicity of intermediate transitions), and *preservation* (extent of non-target content change). To capture these properties, we introduce two new metrics grounded in textual alignment and perceptual similarity.

In our experimentation, we observe a fundamental challenge shared by our method, the original CS approach, and other variants: all require a *scale* that specifies how much to increase or decrease

a given property. Two issues arise. (i) It is unclear where a concept *saturates* (e.g., what scale corresponds to the "maximum age" of a person). (ii) Because diffusion traversals are inherently non-linear, linearly stepping the scale often yields non-linear semantic progress, which confuses users and hinders intuitive interaction (e.g., skipping age ranges unless many fine steps are used, which is computationally expensive). To resolve this, we introduce **Automatic Saturation and Traversal Detection (ASTD)**, a two-stage procedure that (1) estimates saturation by combining *conceptual* and *perceptual* scores with a user-tunable trade-off, and (2) fits a data-driven *reparameterization curve* that rescales the traversal so intermediate steps correspond to perceptually uniform, semantically meaningful changes. Empirically, ASTD has been found to improve the overall score (up to $\sim 2\times$).

In summary, we present a simple, architecture- and modality-agnostic approach to fine-grained concept control that requires neither per-concept training nor architectural adaptations, enabling seamless use across images, video, and audio. We extend the CS benchmark into the first modality-agnostic evaluation suite for fine-grained control and introduce metrics that more faithfully capture fine-grained control. We further tackle scale selection, saturation, and non-linear traversals with **ASTD**, a two-stage procedure that detects saturation points and reparameterizes traversals for perceptually uniform changes. Together, these contributions provide a principled, practical, and extensible framework for training-free, fine-grained concept manipulation across modalities. An interactive presentation of our benchmark and method is available at: https://anonymous.4open.science/w/FreeSliders-4CD3.

## 2 RELATED WORK

**Controllable generation.** Controllability has long been recognized as a fundamental challenge in generative modeling. Early methods for text-to-image diffusion focused on prompt engineering and classifier guidance, which allow only coarse influence over the output distribution (Dhariwal & Nichol, 2021). Recent subsequent approaches introduced more explicit control mechanisms, such as attention manipulation (Meng et al., 2022; Hertz et al., 2023) and instruction-based fine tuning (Brooks et al., 2023). Another prominent line of work focuses on personalization, enabling user-specific concepts to be embedded into pre-trained models through techniques such as Textual Inversion (Gal et al., 2023a), DreamBooth (Ruiz et al., 2023), and LoRA-based fine-tuning (Gal et al., 2023b; Tewel et al., 2023) or others (Gaintseva et al., 2025). While these methods achieve impressive results, they typically require additional training or are tailored to specific modalities or architectures, which limits their scalability and ease of use. These limitations motivate the search for more lightweight and broadly applicable mechanisms of control.

**Finding semantic directions.** A complementary research direction seeks to identify semantic directions in the latent or parameter space of generative models, enabling interpretable traversal along meaningful attributes. In the context of GANs, several works demonstrated that linear directions correspond to factors such as pose, age, or expression (Härkönen et al., 2020; Jahanian et al., 2020; Shen et al., 2020). Recent works have extended this paradigm to diffusion models, proposing methods such as Concept Sliders (Gandikota et al., 2024), Prompt Sliders (Sridhar & Vasconcelos, 2024), CASteer (Gaintseva et al., 2025), NoiseCLR (Dalva & Yanardag, 2024), and Concept Steerers (Kim & Ghadiyaram, 2025). While these methods highlight the promise of semantic directions for fine-grained control, they generally rely on training additional modules or on architecture-specific assumptions (e.g., LoRA adapters, text encoders, or cross-attention layers), and have been explored almost exclusively in the image domain. Our work builds on this line of inquiry but departs fundamentally by showing that effective semantic directions can be discovered in a training-free manner and applied consistently across text-to-image, text-to-video, and text-to-audio models.

A related work is Huang et al. (2024), which proposes an editing-based steering method that discovers concept directions using a memory bank of intermediate diffusion states and DDIM inversion. Their approach is fundamentally designed for post-hoc image editing rather than generation from noise (our setting), requires storing and manipulating a bank of trajectories, and is tied to a specific sampler. In contrast, our method operates directly at inference time without memory banks and targets concept control during generation. While Huang et al. (2024) also demonstrate applications on images and audio, they do not provide quantitative evaluation for audio, nor do they consider video. Our work fills these gaps by introducing quantitative multimodal benchmarks (image, video, and audio), which we view as crucial for fair comparison and future methodological progress.

# 3 ARCHITECTURE-AGNOSTIC, TRAINING-FREE CONCEPT SLIDERS

In this section, we provide an introduction to diffusion models and the original CS framework. We then present our simple partial inference-time estimation approach.

## 3.1 BACKGROUND

**Diffusion models** (Ho et al., 2020; Song et al., 2021) are a class of generative models that synthesize data by learning to reverse a fixed forward noising process. Conceptually, this process begins with a clean data sample, $x_0$, and gradually adds Gaussian noise over a sequence of $T$ timesteps. As $t$ approaches $T$, the data sample $x_T$ is transformed into a sample from a simple prior distribution, typically an isotropic Gaussian, $\mathcal{N}(0, \mathbf{I})$. The generative process learns to reverse this transformation, starting from a random sample $x_T$ and progressively denoising it to produce a realistic sample $x_0$. The core of this reverse process lies in the estimation of the *score function*, which is the gradient of the log-probability of the noisy data $x_t$ with respect to itself. A neural network $s$, parameterized by $\theta$, is trained to approximate this score:

$$s_\theta(x_t, t, c) \approx \nabla_{x_t} \log p(x_t | c) , \tag{1}$$

where $t$ is the timestep and $c$ is conditioning information, such as a text. The score function effectively provides a vector field that points towards regions of higher data density, guiding the generation process. The ability to modify the score function $s_\theta$ during denoising allows for fine-grained control over the generated output, which forms the basis for concept-based methods.

**Concept-based methods** such as Concept Sliders (Gandikota et al., 2024) and Prompt Sliders (Sridhar & Vasconcelos, 2024) manipulate the model's score function to identify *concept directions*. Specifically, Concept Sliders (CS) learn a score function that combines the score at time $t$ with a scaled concept direction, derived from the difference between two conditioning prompts. Formally,

$$\nabla_{x_t} \log p_\omega(x_t \mid c_{\text{base}}) \propto \nabla_{x_t} \log p_\theta(x_t \mid c_{\text{base}}) + \eta \left[ \nabla_{x_t} \log p_\theta(x_t \mid c_+) - \nabla_{x_t} \log p_\theta(x_t \mid c_-) \right] , \tag{2}$$

where $\eta$ is a user-controlled scaling factor, $c_{\text{base}}$ is the base conditioning prompt, and $c_+$ and $c_-$ represent the positive and negative concepts, respectively. The parameters $\theta$ and $\omega$ correspond to the pre-trained diffusion model weights and the trainable LoRA network weights, respectively. Intuitively, this adjustment shifts the generation trajectory toward or away from the target concept while preserving the global structure and identity of the generated sample. In practice, a dataset containing positive, negative, and base examples is prepared for each target concept (e.g., base: "a car", positive: "brand new", negative: "wrecked"; the resulting target concept is "car condition"). Then, the LoRA adapter is fine-tuned to minimize

$$\min_\omega \left\| \epsilon_\omega(x_t, c_{\text{base}}, t) - \epsilon_\theta^{\text{mod}} \right\| , \tag{3}$$

where $\epsilon_\theta^{\text{mod}} := \epsilon_\theta(x_t, c_{\text{base}}, t) + \eta \left[ \epsilon_\theta(x_t, c_+, t) - \epsilon_\theta(x_t, c_-, t) \right]$ and $\epsilon_\omega, \epsilon_\theta$ are the outputs of the denoising networks. In practice, $\omega$ corresponds to LoRA weights that are fine-tuned per concept, modifying the pre-trained diffusion model weights, i.e., $\theta_{\text{new}} = \theta + \alpha\omega, \alpha \in \mathbb{R}^+$. Controlling the intensity $\eta$ of the target concept can be achieved by modifying $\alpha$ during inference. See more information about CS in App. A.6.

As described in the previous sections, current concept sliding methods require a dedicated *training phase* involving: (1) collecting datasets for each new concept, and (2) fine-tuning a diffusion model with concept-specific parameters. While effective, these steps are often infeasible for casual creators who may not have the technical expertise, resources, datasets and time to train models.

## 3.2 OUR APPROACH

In the original CS framework, the modified score in Eq. 2 is approximated by training LoRA adapters to match the target modification. At inference, all three required scores

$$\nabla_{x_t} \log p_\theta(x_t \mid c_{\text{base}}), \quad \nabla_{x_t} \log p_\theta(x_t \mid c_+), \quad \nabla_{x_t} \log p_\theta(x_t \mid c_-)$$

are implicitly encoded in the learned LoRA model. In our *training-free* approach, we compute Eq. 2 directly from the frozen, pre-trained model without fine-tuning. This is achieved by performing three forward passes per denoising step: one for the base concept, one for the positive concept, and one for the negative concept. Then, the diffusion processes reads:

$$\epsilon(x_t, t) = \begin{cases} \epsilon_\theta(x_t, c_{\text{base}}, t) , & \text{if } t \leq k , \\ \epsilon_\theta^{\text{mod}}(x_t, c_{\text{base}}, c_+, c_-, t) , & \text{if } t > k . \end{cases} \tag{4}$$

We first run $k$ steps using only the *base concept*. After step $k$, we apply $\epsilon_\theta^{\mathrm{mod}}$ at each remaining timestep. While the choice of $k$ is heuristic, we observed several practical guidelines for selecting it. Please see App. C.5 for further discussion. This two-stage process first fixes the *base* concept (the main theme) and then enables fine-grained control over the desired target attribute. Unlike Choi et al. (2021), we do not use any classifier. Moreover, in contrast to Gandikota et al. (2024); Kim & Ghadiyaram (2025); Sridhar & Vasconcelos (2024), we do not train additional weights and directly estimate the score. Although our changes are simple, to the best of our knowledge, this setting has not been explored in prior CS-based work. The impact is twofold: (i) by focusing only on the **sampling** (Karras et al., 2022), it fully decouples the method from both model architecture and data modality, and (ii) at the cost of modest extra inference compute, it removes the need for any training. We analyze its inference-time overhead and complexity in Sec. 6.2, highlighting the trade-offs. See App. A.1 for more details about our inference algorithm procedure.

## 4 EVALUATION OF CONCEPT SLIDERS

In this section, we first address two guiding questions: *How do we measure fine-grained control?* and *how can we establish a unified protocol across modalities?* We then revisit current evaluation practices and, finally, propose metric and protocol refinements to close the gaps we identify.

### 4.1 PROPERTIES OF SLIDER EVALUATION

First, we align terminology: a *slider* is any method that, given a target concept, enables explicit control over it. While the motivation for fine-grained control is well established, what constitutes a *good* slider is less explored. Before measuring quality, we propose clear, modality-agnostic properties that serve as objective evaluation criteria:

**Range.** The range of a slider refers to the extent of concept variation it can achieve within a given model. For example, for an "age" concept, one sliding method might produce outputs spanning from young children to individuals over 100 years old, while another, using the same underlying model, may only reach appearances resembling 70–80 years old.

**Smoothness.** The Smoothness of a slider describes its ability to generate *intermediate* conceptual instances in a smooth and perceptually meaningful way. In the age example, a slider with a wide range (e.g., 20–100 years old) might still fail to produce realistic intermediate stages such as 50 or 60 years old. Thus, a model can have a good range without necessarily being smooth.

**Preservation.** Preservation refers to a slider's ability to maintain the original content and identity of the input while modifying only the target concept. For example, when adjusting a person's age, we aim to retain their original facial identity rather than producing an entirely different individual. This property can sometimes impose a trade-off: certain concepts inherently require altering or discarding aspects of the original identity (e.g., transforming a human into a demon).

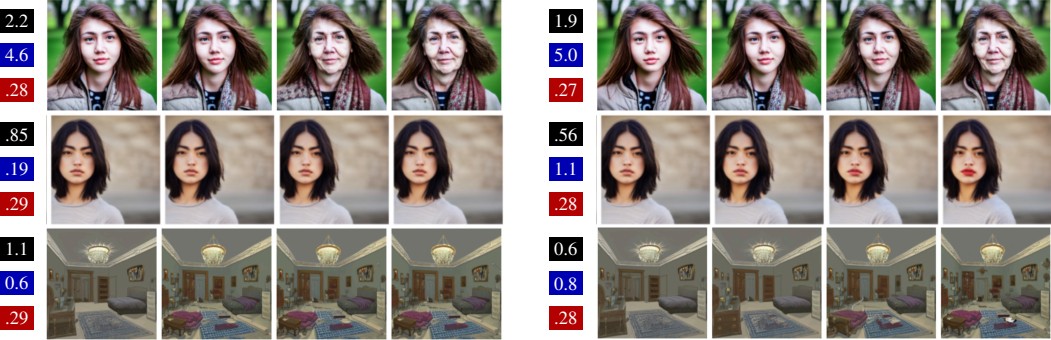

Figure 2: Examples illustrating limitations of the $\Delta$CLIP metric (black boxes): a high score despite abrupt transitions and flat regions at the slider ends (left), and a lower score despite a clear and smooth concept change across intervals (right). Blue and red boxes denote the CR and CSM metrics, respectively, introduced in Sec. 4.3.

## 4.2 LIMITATIONS OF CURRENT EVALUATION

Revisiting the standard evaluation protocol of Gandikota et al. (2024), two key metrics are used: (1) **LPIPS** (Zhang et al., 2018), which measures perceptual similarity between the original and edited images; and (2) $\Delta$**CLIP**, which quantifies the change in CLIP (Radford et al., 2021) similarity between the original and edited images with respect to a target text prompt, thereby estimating alignment with the desired concept. Under our taxonomy, LPIPS naturally serves as a validator of the *preservation* property and is widely adopted in other benchmarks for the same purposes. Therefore, we retain it as our preservation measure. However, closer inspection of $\Delta$CLIP (see App. A.5 for full formulation) alongside visual results reveals it often overlooks aspects of *range* and *smoothness*, and in some situations may give misleading scores. This limitation stems from the fact that $\Delta$CLIP was not designed to capture smoothness or ranges of opposite directions.

In Fig. 2, we compare two sliders side by side. The left column attains a higher $\Delta$CLIP (black box; higher is better) than the right, yet the right column produces a qualitatively better slider. Rows show edits to *age* (top), *glasses* (middle), and *cluttered room* (bottom). These examples reveal that $\Delta$CLIP misses both *range* (extent of change) and *smoothness* (consistency of transitions): in several cases, the visuals remain nearly unchanged or shift abruptly (see App. C.3 for details).

## 4.3 QUANTIFYING SLIDER PROPERTIES WITH NEW METRICS

Building on the previous section, we introduce metrics that alleviate the earlier limitations and explicitly measure the three desired properties. We summarize them here; full definitions and a comparison to $\Delta$CLIP appear in App. A.4, A.5.

**Semantic Preservation (SP)** ↓**:**   Let $x_\eta$ denote the sample generated with slider scale $\eta$, and let $x_0$ be the original (neutral) sample. Given a set of scales $G$ (excluding $\eta = 0$), we define the preservation score as $P(G) = \frac{1}{|G|} \sum_{\eta \in G} d(x_\eta, x_0)$, where $d$ is a perceptual distance measure, such as LPIPS for images, LPAPS (Iashin & Rahtu, 2021; Paissan et al., 2023) for audio, or Video-LPIPS (Voleti et al., 2022) for video. This measures the average perceptual deviation of the slider outputs from the original instance. Lower values indicate better preservation of non-target content. This metric complements the others: a trivial method that makes no changes at all will achieve perfect preservation, but very poor range and smoothness. This perceptual measure has been used in the standard CS benchmark.

**Conceptual Range (CR)** ↑**:**   To measure how well the slider stretches a concept space, we measure the alignment scores at the endpoints. Let $a(x, c)$ denote the alignment between a generated sample $x$ and a concept prompt $c$. Define: $\text{CR} = 0.5(\text{CR}_{\text{pos}} + \text{CR}_{\text{neg}})$, where

$$\text{CR}_{\text{pos}} = a(x_{\eta_{\max}}, c_+) - a(x_{\eta_{\min}}, c_+) , \quad \text{CR}_{\text{neg}} = a(x_{\eta_{\min}}, c_-) - a(x_{\eta_{\max}}, c_-) .$$

Here $c_+$ and $c_-$ are the positive and negative prompts, and $\eta_{\min}, \eta_{\max} \in G$ are the minimum and maximum scales. Alignment models $a$ include CLIP for images, ViCLIP (Wang et al., 2023) for video, and CLAP (Wu & De la Torre, 2023; Chen et al., 2022) for audio. This metric quantifies the extent of the semantic change. As illustrated in Fig. 2, larger concept shifts yield higher CR. Additionally, unlike $\Delta$CLIP, which measures change only between a neutral point and a single target scale, our metric captures the range property by leveraging both positive and negative directions to represent the full span of variation.

**Conceptual Smoothness (CSM)** ↓**:**   Finally, we assess how uniformly the slider distributes concept changes across its scale values. Rather than considering raw alignment scores, we examine the distribution of score increments. For each $\eta \in G$, let $A = \{\, a(x_\eta, c_r) \mid \eta \in G, \ r = \text{sign}(\eta) \,\}$. We normalize $A$ to the interval $[0, 1]$ by rescaling to the maximum attainable alignment score for the concept and assuming the minimum score is 0. We split $A$ into its negative and positive subsets and order each subset separately along the slider scale. For each ordered subset, we compute consecutive gaps $g_i = A_{i+1} - A_i, i = 1, \ldots, |A| - 1$. and define $\text{CSM} = \text{std}(\{g_i\})$. A lower CSM indicates a smoother and uniform traversal of the concept space; higher values reflect irregular jumps or skipped semantic regions. Thus, a smooth slider maps successive steps to perceptually coherent, evenly spaced changes. As shown in Fig. 2, smoother transitions yield lower (better) CSM.

## 5 SATURATION AND TRAVERSAL DETECTION

The range and smoothness of a slider depend heavily on adjusting the scaling factor $\eta$ for each concept. In prior work, $\eta$ values and intermediate steps are fixed and uniformly distributed across

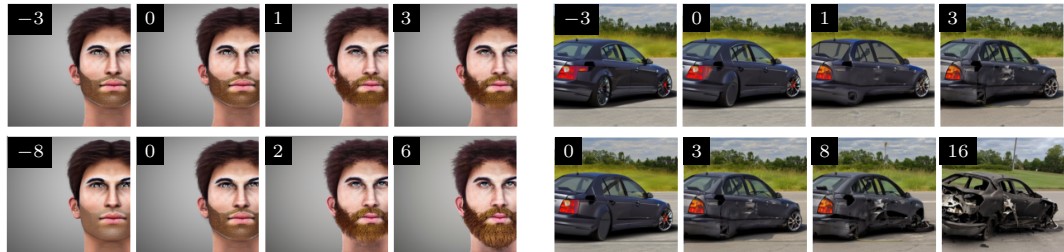

Figure 3: Default CS scales often yield suboptimal results as shown in the top row, while the bottom row shows ASTD-optimized scales and step sizes. Black boxes mark sampled $\eta$ values.

concepts. This leads to two key issues (Fig. 3): (1) different concepts operate effectively at different $\eta$ scales, and (2) linear traversal in $\eta$ often induces *nonlinear* conceptual progress, resulting in non-smooth changes. We therefore propose a two-step procedure to automatically improve slider quality.

**Step 1: Saturation Detection.** Determining when a concept saturates is non-trivial. Let $a(x, c)$ denote an *alignment score* (e.g., CLIP) measuring the presence or intensity of a concept in a generated image. A naïve approach would declare saturation when $a(x_\eta, c)$ no longer increases with $\eta$. However, under classifier-free guidance, degraded images can sometimes yield high scores while losing perceptual meaning. To counter this, let $d(x_0, x_\eta)$ be a similarity measure to the base image (e.g., LPIPS). This ensures edits do not deviate too far from the original content. Since some concepts inherently require larger perceptual changes, we define a user-adjustable trade-off:

$$r(x, \eta) \; = \; \frac{a(x_\eta, c)}{d(x_0, x_\eta)} \, .$$

The ratio $r$ balances preservation ($r < 1$) and concept intensity ($r > 1$). In practice, we set $r = 1$ (Fig. 4, left, black line). We then evaluate $r(x, \eta)$ at $\eta \in \{0, 0.5, 1, 2, 4, 8, 16\}$ and choose the largest $\eta$ above the reference line as the saturation point (e.g., $\eta = 8$ in the figure). The same process is applied to negative scales. Importantly, exploring larger ranges is often necessary: without saturation detection, the original CS (restricted to $[-3, 3]$ during evaluation) may entirely miss the effective editing range (Fig. 3). While one could naïvely sweep many scales, this is computationally prohibitive and not user-friendly, motivating our principled approach.

**Step 2: Traversal Adjustment.** After identifying the saturation point, we analyze the curve $\eta \mapsto a(x, \eta)$ to improve traversal smoothness. As shown in Fig. 4, conceptual progression is better captured by a low-degree polynomial than by uniform $\eta$ sampling. We therefore fit a monotone reparameterization that maps the alignment axis to the scale axis, resampling $\eta$ so that equal steps correspond to perceptually uniform changes between 0 and the saturation point. This adjustment yields smoother, more coherent transitions along the concept axis (Fig. 3), enhancing usability for both the original CS framework and our method.

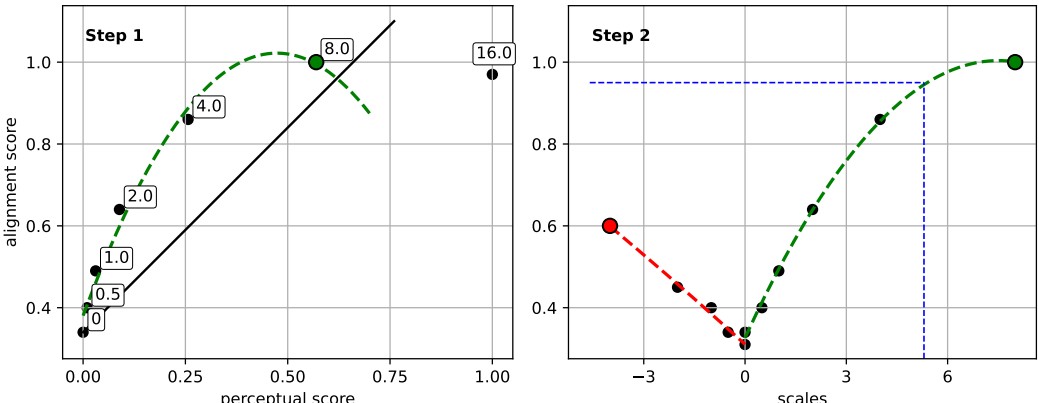

Figure 4: Automatic Saturation and Traversal Detection (ASTD) for concept "smiling". **Left:** saturation detection via $r(x, \eta)$. **Right:** traversal reparameterization for smoother progression.

# 6 RESULTS

This section has two parts. First, we present standardized, independent benchmarks for images, video, and audio. Second, we report additional experiments: (1) adaptability to different backbone models; (2) validation of ASTD's effectiveness; (3) computational analysis of our method and ASTD; and (4) demonstration of compositional control. Further implementation details for baselines, prompts, concepts, data, and metrics are given in App. A.

## 6.1 MULTI-MODAL QUANTITATIVE BENCHMARK

We standardize the *Image* benchmark from Gandikota et al. (2024) and extend it to *Video* and *Audio*. Each modality covers **ten** concepts. For each concept, we generate 100 sliders in the image benchmark and 10 sliders in the video and audio benchmarks, evaluating each at seven scale values selected as either the default or the ASTD-optimized option, whichever performs better. Importantly, all methods achieve higher overall scores with ASTD. All methods use the same backbone per modality. We first compute the per-concept average across sliders, then report the mean across concepts as the final metric. We use the metrics from Sec. 4.3. For compact reporting, we also provide a single *Overall Score* (OS), OS $= \frac{\text{CR}}{\epsilon + \text{SP}} + (1 - \text{CSM})$,

where CR is *Conceptual Range*, SP is *Preservation*, and CSM is *Smoothness*; $\epsilon = 1$ stabilizes the ratio for small SP. This design captures the trade-off between conceptual alignment (numerator) and semantic preservation (denominator), augmented by smoothness over the traversal. Qualitative comparisons appear in App. C.1.

**Evaluating image concepts.** We use *Stable Diffusion v1.4* as the backbone and compare against strong modality-agnostic and/or training-free baselines (Gandikota et al., 2024; Sridhar & Vasconcelos, 2024; Hertz et al., 2023; Huang et al., 2024). Our method is *training-free*, *modality-agnostic*, and *architecture-independent*. Thus, its important to note, that in those aspects, it has inherent advantages upon the other methods. We also implement a Text Embedding (TE) baseline that applies semantic contrast in text-embedding space rather than score space (see App. A.3.1). In Tab. 1 (Image), our method achieves the largest *CR*, the lowest *SP*, competitive *CSM*, and thus the best overall score.

**Evaluating video concepts.** We use *CogVideoX-2B* (Yang et al., 2024). Videos contain both static and dynamic elements: e.g., for *river flow strength*, water motion should vary while background remains stable; for *car style*, dynamics should be preserved while appearance changes. We therefore report both *Dynamic* and *Static* metrics (implementation details in the appendix); table markers **(D)** and **(S)** denote these subsets. Final scores follow the image setup. We compare against adapted video versions of CS and Prompt Sliders; Prompt-to-Prompt was not feasible due to CogVideoX's special attention mechanism. In Tab. 1 (Video), our method attains higher OS and is competitive or superior on each metric for both static and dynamic evaluations.

**Evaluating audio concepts.** We use *Stable Audio Open* (Evans et al., 2025). Final scores follow the same protocol, and we compare against CS and Prompt-to-Prompt. In Tab. 1 (Audio), our method attains better *CSM* and *SP*, competitive *CR*, and the best OS among all methods.

## 6.2 ADDITIONAL EXPERIMENTS

**Qualitative comparison across backbones.** Modality-agnostic methods like ours are well-suited to the rapidly evolving landscape of foundation models, where both backbones and generative denoising paradigms are changing rapidly. To qualitatively demonstrate adaptability, we compare LTX-Video (HaCohen et al., 2024) (a flow-matching–based model with a transformer backbone) and CogVideoX (Yang et al., 2024) (a diffusion-based model with a transformer backbone) in Fig. 5. Notably, adopting our method on a new backbone requires only minor adjustments to the sampling procedure. We include a brief discussion of the equivalence between different denoising formulations and their relation to our concept sliding in App. A.7. Furthermore, we also demonstrate the adaptation of both our inference-time approach and ASTD to Stable Diffusion 3 in the supplementary materials; see App. C.4.

**Performance analysis and ablation.** Training CS is expensive in both peak memory and wall-clock time (WCT), as seen in Tab. 2. In comparison, our method requires no training and operates only at inference time. Relative to CS inference, our approach achieves **similar memory usage**.

Table 1: Benchmarking results across image, video, and audio concepts, in comparison to several strong baselines. Unavailable entries are shown as "–".

| | Metric | Huang et al. (2024) | CS | Prompt Sliders | Prompt-to-Prompt | TE | Ours |
|---|---|---|---|---|---|---|---|
| Image | CR ↑ | .405 ± 2.14 | 2.54 ± .776 | .079 ± .374 | .566 ± .554 | .927 ± .803 | 2.85 ± .861 |
| | CSM ↓ | 0.36 ± −0.09 | .276 ± .009 | .292 ± .014 | .257 ± .013 | .285 ± .010 | .283 ± .009 |
| | SP ↓ | .150 ± .011 | .062 ± .031 | .065 ± .032 | .125 ± .029 | .019 ± .013 | .018 ± .015 |
| | Overall Score ↑ | 0.28 | 3.11 | .782 | 1.24 | 1.62 | 3.56 |
| Video | CR (S) ↑ | – | 1.17 ± .44 | −.173 ± .59 | – | 1.30 ± .55 | 1.52 ± .60 |
| | CR (D) ↑ | – | .013 ± .01 | −.001 ± .01 | – | .021 ± .01 | .024 ± .01 |
| | CSM (S) ↓ | – | .308 ± .01 | .328 ± .02 | – | .306 ± .01 | .311 ± .01 |
| | CSM (D) ↓ | – | .460 ± .01 | .457 ± .02 | – | .442 ± .01 | .436 ± .01 |
| | SP (S) ↓ | – | .761 ± .02 | .112 ± .05 | – | .041 ± .01 | .032 ± .01 |
| | SP (D) ↑ | – | −1.56 ± .94 | −.931 ± .84 | – | −1.19 ± .67 | −.996 ± .54 |
| | Overall Score ↑ | – | 2.29 | 1.04 | – | 2.40 | 2.70 |
| Audio | CR ↑ | – | 5.78 ± 1.42 | – | 6.75 ± 2.23 | 6.13 ± 1.72 | 6.04 ± 2.04 |
| | CSM ↓ | – | .351 ± .016 | – | .386 ± .026 | .393 ± .018 | .344 ± .024 |
| | SP ↓ | – | 2.72 ± 0.32 | – | 4.19 ± 0.59 | 4.21 ± 0.32 | 2.13 ± 0.42 |
| | Overall Score ↑ | – | 2.20 | – | 1.91 | 1.78 | 2.58 |

CogVideoX  LTX-Video

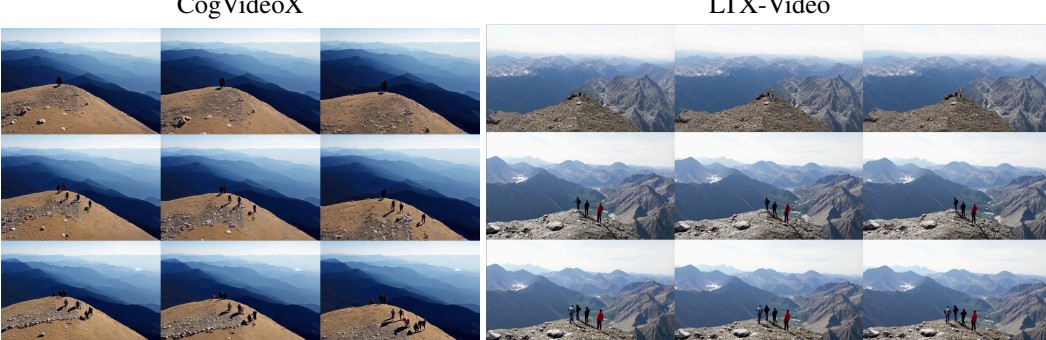

Figure 5: Video sliders for the concept "mountain hikers" with different backbones.

In terms of time, it is important to note that, asymptotically, our computational overhead matches that of a trained Concept Slider (CS) and incurs only a factor of 1.4 more compute due to the extra evaluations. In our setup, the additional cost arises from three extra network evaluations used in a cleaning step; however, they are invoked only after $k$ steps ($k = 15$ in most cases) out of 50 steps in video/audio generation. Additionally, batching evaluations and ongoing advances in fast diffusion/flow sampling techniques (e.g., Frans et al. (2024)) are directly compatible with our method and can further reduce its compute overhead, substantially narrowing the remaining computational gap and making our approach well-suited for modern scalable systems. In this work we focus on the methodological contribution itself and leave these engineering optimizations for future exploration.

Finally, to analyze ASTD, we recall that it requires generating a small set of samples to run its two-stage procedure. In our setup, we evaluate each concept at 13 scales and generate 30 samples, i.e., an *inference* cost equivalent to 390 sample generations. To further improve its time performance, we empirically find that using half the number of sampling steps produces essentially the same final calibration, yielding an approximate $2\times$ speed-up; we refer to this variant as ASTD2x. ASTD/ASTD2x is applied *on top of* existing methods to yield improved results and can be seen as a stand-alone component; in our measurements, running ASTD2x is faster than CS training (although could come on top of it) and requires only slightly more memory.

We validate that ASTD consistently improves overall scores across modalities and methods, demonstrating robustness to diverse setups. Tab. 3 summarizes the results: the left panel reports CS and our method *without* ASTD, while the right panel shows that all overall scores (OS) increase when ASTD is enabled, by nearly $2\times$ in images (I) and audio (A). Additionally, we observe that ASTD2x attains an Overall Score $OS(I)$ very similar to that of ASTD, thus not only reducing computation time by $2\times$ but also preserving the same quality. We hypothesize that these results indicate the alignment and semantic-preservation measurements used in the different stages of ASTD remain robust even under this reduced-quality regime, since most concepts are sufficiently semantically dominant even when the visual quality is somewhat degraded.

Table 2: Runtime and memory comparisons of CS, our approach, and the ASTD add-on.

|  | CS-Train | CS-Infer | Ours | ASTD2x |
|---|---|---|---|---|
| WCT (s/m) | 22 (m) | 6.5 (s) | 9.5 (s) | 16 (m) |
| Mem (GB) | 6.7 | 4.4 | 5.0 | 6.8 |

Table 3: ASTD contribution ablation.

|  | Ours (w/o) | CS (w/o) | Ours (w/) | CS (w/) |
|---|---|---|---|---|
| OS (I) ↑ | 3.04 | 2.26 | **3.56** | **3.11** |
| OS (V) ↑ | 2.48 | 2.28 | **2.70** | **2.29** |
| OS (A) ↑ | 1.83 | 1.82 | **2.58** | **2.20** |

**Slider composition.** Our method naturally supports compositional edits, enabling multiple sliders to be applied jointly. As shown in Fig. 6, we simultaneously adjust the *age* and *smile* concepts, demonstrating controlled manipulation across multiple attributes. A full compositional grid illustrating these interactions across all scale values is included in Fig. 8. All CS-style methods are subject to the underlying entanglement of the backbone diffusion model. For example, if changing "age" also changes "gender," or altering "hair color" also shifts "gender," the corresponding attribute vectors may interfere; conversely, if the backbone is inherently disentangled, such interference should not occur. In our experiments, we did not observe clear cases of such interference, but entanglement remains an open research question and an important avenue for future work.

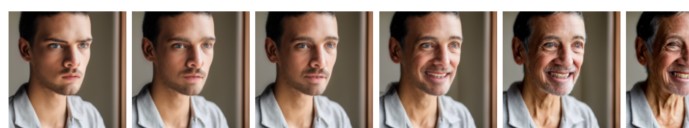

Figure 6: Slider composition: the concepts "age" and "smile" combined together.

## 7 DISCUSSION

In this work, we demonstrate that fine-grained concept control in diffusion models can be achieved *without* per-concept training or architecture-specific modifications. By directly estimating the Concept Slider update during inference, our method enables plug-and-play, modality-agnostic sliders across images, video, and audio. This design trades a modest increase in inference cost for substantial gains in generality, ease of deployment, and performance, surpassing strong baselines in controllable generation. Nevertheless, several limitations remain. Long sequences (e.g., video or audio) introduce higher computational demands, alignment models such as CLIP, ViCLIP, and CLAP may propagate biases, and our two-stage saturation detection requires additional samples. These challenges highlight opportunities for improvement through inference-time acceleration (e.g., few-step solvers, distillation), better handling of alignment biases, and more efficient saturation modeling.

Looking forward, promising directions include human-in-the-loop traversal shaping, principled multi-concept disentanglement, region-specific alignment of concepts, richer composition, and extensions to emerging modalities such as 3D, robotics, and cross-lingual generation. Additionally, ASTD presents a promising way to significantly improve results, but it is currently based on heuristics; exploring its theoretical foundations could be highly valuable. More broadly, lowering technical barriers to controllable generation has the potential to democratize creative and scientific workflows, but it also raises concerns regarding misuse and fairness. Standardized, transparent benchmarks and metrics, combined with safeguards for sensitive attributes, are essential to ensure that training-free, architecture-independent concept sliders are deployed responsibly, interpretably, and ethically.

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

# A  EXPERIMENTAL DETAILS

## A.1  ALGORITHM PSEUDO CODE

---

**Algorithm 1** Our Inference-Time Method with Subroutines

---

**Require:** Initial latents $x_0$, prompt pair $(c_{\text{neutral}}, c_+, c_-)$, timesteps $T$, scales $S$, intervention timestep $k$, guidance scales $g_1$

1: Initialize noise scheduler with $T$ steps $\qquad\qquad$ ▷ Step 1: Diffuse using neutral prompt until $k$
2: **for** $t = 0$ to $k - 1$ **do**
3: $\qquad \epsilon \leftarrow \text{PredictNoise}(x, c_{\text{neutral}}, g_1, \text{use\_cfg=True})$
4: $\qquad x \leftarrow \text{StepScheduler}(x, \epsilon, t)$
5: **end for**
$\qquad\qquad\qquad\qquad\qquad\qquad\qquad\qquad\qquad$ ▷ Step 2: Guided diffusion for each scale
6: **for** each scale $\mu$ in $S$ **do**
7: $\qquad x_\mu \leftarrow x$
8: $\qquad$ **for** $t = k$ to $T$ **do**
9: $\qquad\qquad \epsilon_{\text{neutral}} \leftarrow \text{PredictNoise}(x_\mu, c_{\text{neutral}}, g_1, \text{use\_cfg=True})$
10: $\qquad\qquad \epsilon_+ \leftarrow \text{PredictNoise}(x_\mu, c_+, \text{use\_cfg=False})$
11: $\qquad\qquad \epsilon_- \leftarrow \text{PredictNoise}(x_\mu, c_-, \text{use\_cfg=False})$
12: $\qquad\qquad \epsilon_{\text{mod}} \leftarrow \epsilon_{\text{neutral}} + \mu \cdot (\epsilon_+ - \epsilon_-)$
13: $\qquad\qquad x_\mu \leftarrow \text{StepScheduler}(x_\mu, \epsilon_{\text{mod}}, t)$
14: $\qquad$ **end for**
15: **end for**
$\qquad\qquad\qquad\qquad\qquad\qquad\qquad\qquad\qquad\qquad$ ▷ Subroutine: Predict Noise
16: **procedure** PREDICTNOISE($x, c, \gamma$=None, use\_cfg=False)
17: $\qquad x_{\text{model}} \leftarrow \begin{cases} \text{duplicate}(x), & \text{if use\_cfg} = \text{True} \\ x, & \text{otherwise} \end{cases}$
18: $\qquad \epsilon_\theta \leftarrow \text{ModelPredict}(x_{\text{model}}, c, t, \text{scheduler})$
19: $\qquad$ **if** use\_cfg = True **then**
20: $\qquad\qquad$ Split $\epsilon_\theta$ into $(\epsilon_{\text{uncond}}, \epsilon_{\text{text}})$
21: $\qquad\qquad \epsilon_\theta \leftarrow \epsilon_{\text{uncond}} + \gamma \cdot (\epsilon_{\text{text}} - \epsilon_{\text{uncond}})$
22: $\qquad$ **end if**
23: $\qquad$ **return** $\epsilon_\theta$
24: **end procedure**

---

## A.2  BENCHMARKING DATA

This section contain all the prompts used for quantitative results.

### A.2.1  IMAGES

```
- base: "A realistic image of a person."
  positive: "A realistic image of a person, smiling widely, very
      happy."
  negative: "A realistic image of a person, frowning, very sad."

- base: "A realistic image of a person."
  positive: "A realistic image of a person, very old, aged,
      wrinkly."
  negative: "A realistic image of a person, detailed facial
      features, clear skin."

- base: "A realistic image of a car."
  positive: "A realistic image of a car, damaged, broken
      headlights, dented car, with scrapped paintwork."
  negative: "A realistic image of a car, mint condition, brand new
      , shiny."
```

```
- base: "A realistic image of a room."
  positive: "A realistic image of a room, cluttered, disorganized,
      dirty, jumbled, scattered."
  negative: "A realistic image of a room, super organized, clean,
      ordered, neat, tidy."

- base: "A realistic image of a person."
  positive: "A realistic image of a person, with big bushy beard."
  negative: "A realistic image of a person, clean shaven."

- base: "A realistic image of a person."
  positive: "A realistic image of a person, wearing glasses."
  negative: "A realistic image of a person, clear face."

- base: "A realistic image of a person."
  positive: "A realistic image of a person, very fat, chubby,
      overweight, obese."
  negative: "A realistic image of a person, very skinny, thin,
      slim, lean."

- base: "A realistic image of a person."
  positive: "A realistic image of a person, with makeup, cosmetic,
      concealer, mascara, lipstick."
  negative: "A realistic image of a person, bare face."

- base: "A realistic image of a person."
  positive: "A realistic image of a person, with shocked look,
      surprised, stunned, amazed."
  negative: "A realistic image of a person, dull, uninterested,
      bored, incurious."

- base: "A realistic image of a table of food."
  positive: "A realistic image of a table of food, cooked and
      prepped, in dishes."
  negative: "A realistic image of a table of food, raw, natural,
      not prepped in any way.
```

### A.2.2  VIDEO

```
- base: "A white sailboat in the ocean."
  positive: "A white sailboat in the stormy wavy ocean."
  negative: "A white sailboat in the calm flat ocean."

- base: "A river flowing through the valley."
  positive: "A river rushing rapidly through the valley, with very
      big waves."
  negative: "A river gently trickling through the valley, with
      very calm flat water."

- base: "A butterfly on a leaf."
  positive: "A butterfly flapping its wings very fast."
  negative: "A butterfly standing on a leaf not moving."

- base: "A silhouette of a person walking on the beach."
  positive: "A silhouette of a person jogging very fast on the
      beach."
  negative: "A silhouette of a person strolling slowly down the
      beach."
```

```
- base: "A mountaintop from afar with a few people hiking down it
    ."
  positive: "A mountaintop from afar with a lot of people hiking
      down it, very busy."
  negative: "A mountaintop from afar with a no people hiking down
      it, empty."

- base: "A car cruising down the street."
  positive: "A shiny sports car cruising down the street."
  negative: "An old fashioned faded car cruising down the street."

- base: "A large mountain in the distance."
  positive: "An all white snow-capped large mountain in the
      distance."
  negative: "A brown dry and rocky large mountain in the distance
      ."

- base: "A tall tree in a windy park."
  positive: "A tall green leafy tree in a windy park, very green
      and with lots of leaves."
  negative: "A tall dry brown bare tree with in a windy park, no
      leaves on branches."

- base: "A cat sitting on a windowsill licking its paws."
  positive: "A very fat fluffy, well-groomed cat sitting on a
      windowsill licking its paws."
  negative: "A very skinny, thin, scruffy cat sitting on a
      windowsill licking its paws."

- base: "A lighthouse in the sea, surrounded by water."
  positive:  "A lighthouse in the sea, surrounded by dark splashy
      water with lots of waves."
  negative:  "A lighthouse in the sea, surrounded by bright calm,
      flat water."
```

### A.2.3   AUDIO

```
- base: "A cat meowing"
  positive: "A cat meowing loudly"
  negative: "A cat meowing softly"

- base: "Ocean wave sounds"
  positive: "Stormy ocean wave sounds"
  negative: "Tranquil ocean wave sounds"

- base: "Car engine sound"
  positive: "Revving car engine"
  negative: "Idling car engine"

- base: "Choir singing a classical piece"
  positive: "Choir singing a classical piece in a high pitch"
  negative: "Choir singing a classical piece in a low pitch"

- base: "Door slamming shut"
  positive: "Wooden door slamming shut"
  negative: "Metal door slamming shut"

- base: "Crowd clapping"
```

```
positive: "Crowd clapping with echo"
negative: "Crowd clapping with no echo"

- base: "Solo guitar"
  positive: "Solo electric guitar"
  negative: "Solo acoustic guitar"

- base: "Dog barking"
  positive: "Big dog barking"
  negative: "Small dog barking"

- base: "Rain falling"
  positive: "Heavy rain falling"
  negative: "Light rain falling"

- base: "Crowd of people talking"
  positive: "Crowd of people shouting"
  negative: "Crowd of people wispering"
```

## A.3    BASELINES IMPLEMENTATION DETAILS

### A.3.1    TEXT EMBEDDING VARIANT

As an alternative to our inference-time method, we explore a variant that directly manipulates the different text embeddings, which we refer to as Text-Embeds (TE). Given text encoder outputs for the neutral, positive, and negative prompts, we construct modified embeddings by linear interpolation in embedding space. At step $k$ of the slider, the manipulated embedding is defined as $e_{mod} = e_{\text{neutral}} + \mu \left( e_{\text{positive}} - e_{\text{negative}} \right)$, where $\mu$ represents the scale in the slider. Until step $k$, the diffusion process is conditioned on the neutral embedding $e_{\text{neutral}}$, and from step $k$ onward the manipulated embedding $e_{mod}$ is used for the full denoising process. In contrast, our inference-time method requires three forward passes, one for each embedding (neutral, positive, and negative), after which the manipulation is applied directly to the noise predictions. While TE is computationally simpler, it is also less expressive, as the conditioning signal remains static once $e_{mod}$ is fixed. In practice, we find that TE can yield smooth semantic progressions but often lacks the fine-grained control and robustness of our main approach.

$$\epsilon(x_t, t) \;=\; \begin{cases} \epsilon_\theta(x_t, e_{neutral}, t), & \text{if } t \leq k, \\ \epsilon_\theta(x_t, e_{mod}, t), & \text{if } t > k. \end{cases} \tag{5}$$

### A.3.2    PROMPT TO PROMPT

To implement Prompt-to-Prompt (P2P) editing, we follow the original framework for controlling cross-attention maps during diffusion. Given a neutral prompt and an edited prompt (e.g., "a person" vs. "a smiling person"), we first generate a base image with the neutral prompt and record its latent representation and attention maps using an attention controller. During editing, the diffusion process is initialized from the neutral latent, and the cross-attention activations of the edited prompt are gradually interpolated with those of the neutral prompt over the denoising steps. To constrain edits to specific semantic regions, we employ LocalBlend, which constructs spatial masks by pooling attention maps of selected target words and restricting modifications to those areas. Additionally, AttentionReweight amplifies the contribution of selected target tokens (e.g., "smiling") by scaling their attention weights with an equalizer, enabling smooth variation of the edit along a slider. In practice, we apply a refinement controller (AttentionRefine) to align tokens between the prompts, followed by AttentionReweight with different scales to generate images at varying interpolation strengths. By reusing the latent from the neutral prompt, the identity and overall structure of the original image are preserved, while the desired semantic attributes are gradually introduced. All modifications are taken directly from the original P2P codebase. In our experiments, we observed that for some concepts, the images remain effectively unchanged along the slider despite the attention manipulations.

### A.3.3  PROMPT SLIDERS

The PromptSliders baseline is based on a training-based textual inversion approach, where a dedicated embedding is learned for each concept. During training, only this new concept embedding is updated, while the rest of the model, including the VAE, U-Net, and most of the text encoder, remains fixed. Once trained, the learned embedding can be used at inference time by inserting it into the text encoder, allowing the model to generate images conditioned on the concept. By adjusting the contribution of the learned embedding through a slider, the approach enables continuous and controllable manipulation of the concept in the generated images, providing a flexible way to explore variations without modifying the underlying diffusion model.

### A.3.4  CONCEPT SLIDERS

The Concept Sliders baseline implements continuous control over a learned concept using LoRA adapters on top of a frozen base model. In this approach, a low-rank adapter (LoRA) is trained on a specific concept, modifying only a subset of the transformer's attention layers (to_q, to_k, to_v, to_out.0) while keeping the rest of the model fixed. During training, for each concept, batches of latent images are generated and denoised using the base model conditioned on three types of prompt embeddings: neutral, positive, and negative. The training objective is a mean-squared error between the predicted noise under a target prompt with the LoRA applied and a combination of predicted noises under neutral, positive, and negative prompts without the LoRA. Formally, if $\epsilon_\theta(\mathbf{x}_t, t, c)$ denotes the noise predicted by the transformer at timestep $t$ for latents $\mathbf{x}_t$ conditioned on text embeddings $c$, the target latent is computed as $\epsilon_{\text{target}} = \epsilon_\theta(\mathbf{x}_t, t, c_{\text{neutral}}) + (\epsilon_\theta(\mathbf{x}_t, t, c_+) - \epsilon_\theta(\mathbf{x}_t, t, c_-))$, and the loss minimizes $\text{MSE}(\epsilon_{\text{pred}}, \epsilon_{\text{target}})$ where $\epsilon_{\text{pred}}$ is the LoRA-modified prediction. Once trained, these LoRA weights can be loaded at inference to manipulate the concept continuously. During inference, for a given latent initialization $\mathbf{x}_0$ and scale $\mu$, the noise prediction is computed as

$$
\epsilon_t = \begin{cases} \epsilon_\theta(x_t, c), & t > t_0 \\ \epsilon_{\theta+\mu\omega}(x_t, c), & t \leq t_0 \end{cases},
\tag{6}
$$

where $\epsilon_\theta$ is the denoising network of the base model, $\omega$ are the concept-specific LoRA weights, and $\mu$ is a scaling factor controlling the intensity of the concept and $t_0$ is the diffusion timestep at which the LoRA-modified weights begin to be applied, while all steps before $t_0$ use the original model weights.

### A.4  METRICS IMPLEMENTATION DETAILS

Each metric uses either the perceptual similarity score $d(x_i, x_j)$, which measures the closeness between generated outputs, or the semantic alignment score $a(x, c)$, which quantifies how well an output matches the input prompt. The metrics that build on these scores are described in section 4.3, while the exact definition of $d$ and $a$ depends on the modality and is described in the following sections.

### A.4.1  IMAGE

The perceptual similarity, $d(x_\mu, x_0)$, is calculated using the Learned Perceptual Image Patch Similarity (LPIPS) metric Zhang et al. (2018), which compares the reference image $x_0$ (corresponding to the neutral scale) with a manipulated image $x_\mu$ at scale $\mu$. LPIPS captures perceptual differences between images, reflecting visual fidelity and preserving global structure.

For semantic alignment $a(x, c)$, we compute the CLIP-based score (Radford et al., 2021), which measures the cosine similarity between the image embedding of $x$ and the text embedding of the corresponding prompt $c$.

### A.4.2  VIDEO

In video, both static and dynamic aspects need to be measured, which we capture by defining separate scores for each. The *static* perceptual similarity $d(S)(x_\mu, x_0)$ is measured using LPIPS (Zhang et al., 2018) with an AlexNet backbone (Voleti et al., 2022), which primarily captures more static

differences between frames, reflecting changes in identity or appearance. *Dynamic* differences between videos $d(D)(x_\mu, x_0)$, like movement or speed, is measured using *Motion Alignment* (Xu et al., 2022), which yields a score sensitive to dynamic changes in motion patterns.

Semantic alignment with the target concept is assessed using text-video CLIP-based scores. *CLIP-Text Alignment* (Radford et al., 2021) computes the similarity between individual frame embeddings and the textual prompt embedding, capturing more static semantic consistency, denoted by $a(S)(x, c)$, whereas *ViCLIP-Text Alignment* (Wang et al., 2023) computes a spatio-temporal embedding of the video and compares it to the text prompt embedding, capturing dynamic semantic alignment across frames, denoted by $a(D)(x, c)$.

In section 4.3, the metrics refer to the scores as $d$ and $a$, for videos, these should be understood as $d(S), a(S)$ for the static versions and $d(D), a(D)$ for the dynamic versions of the metrics.

### A.4.3 AUDIO

For generated audio, perceptual similarity, $d(x_\mu, x_0)$, is computed with the LPAPS metric following (Manor & Michaeli, 2024), using the LAION-AI CLAP library (Wu* et al., 2023) with the `music_speech_epoch_15_esc_89.25.pt` weights (Iashin & Rahtu, 2021; Paissan et al., 2023). We compare the reference audio $x_0$ (neutral scale) to the manipulated audio $x_\mu$ at manipulation scale $\mu$.

Semantic alignment is measured with the CLAP score $a(x, c)$, defined as the cosine similarity between the audio embedding of $x$ and the text embedding of the corresponding prompt $c$ (Wu & De la Torre, 2023; Chen et al., 2022). Following (Manor & Michaeli, 2024), we compute CLAP using the LAION-AI CLAP library (Wu* et al., 2023) with the `music_speech_audioset_epoch_15_esc_89.98.pt` weights.

### A.5 DELTA CLIP

By using the corresponding alignment score $a(x, c)$ for each modality, let $a(x_\mu, c_+)$ denote the CLIP score of the image at scale $\mu$ compared the positive prompt $c_+$. The $\Delta CLIP$ score for each scale $\mu$ is calculated by $\Delta\text{CLIP}(\mu) = \big|a(x_\mu, c_+) - a(x_0, c_+)\big|$, and the overall $\Delta CLIP$ for the slider is the average across all non-neutral scales $S$: $\Delta\text{CLIP}_{\text{avg}} = \frac{1}{|\mathcal{S}\setminus\{0\}|}\sum_{\mu\in\mathcal{S}\setminus\{0\}}\Delta\text{CLIP}(\mu), \mu \in S$.

### A.6 CONCEPT-SCORE INTERPRETATION AND COMPOSITION.

The assumption that the *difference* between noise predictions encodes a concept direction is precisely the modeling assumption underlying Concept Sliders (CS) (Gandikota et al., 2024). In CS, the target score for a concept slider is

$$\nabla_{x_t} \log p(x_t \mid c_{\text{base}}) \;+\; \varepsilon\Big(\nabla_{x_t} \log p(x_t \mid c_+) - \nabla_{x_t} \log p(x_t \mid c_-)\Big), \tag{7}$$

and the LoRA parameters are trained to approximate this ideal update. Our method instead *computes* this expression directly at inference time with the frozen model, rather than learning an adapter to approximate it. In this sense, we introduce no new modeling assumption: we operate on exactly the same quantity that CS aims to learn.

From a probabilistic perspective,

$$\nabla_{x_t} \log p(x_t \mid c_+) - \nabla_{x_t} \log p(x_t \mid c_-) \;=\; \nabla_{x_t} \log \frac{p(x_t \mid c_+)}{p(x_t \mid c_-)}, \tag{8}$$

i.e., this is the gradient of the log-likelihood ratio between the two conditional distributions. This direction is the steepest local change that simultaneously increases compatibility with $c_+$ and decreases compatibility with $c_-$, aligning with the intuitive notion of a semantic "direction" between opposing concepts. Similar contrastive score differences are used in classifier guidance, classifier-free guidance, and recent steering methods (e.g., NoiseCLR and related works cited in Sec. 2.2).

This formulation also makes the composition of several concepts natural (Gandikota et al., 2024). For a set of concept triples $\mathcal{C} = \{(c_{\text{base}}, c_+, c_-)\}$, the composed update can be written as

$$\nabla_{x_t} \log p(x_t \mid c_{\text{base}}) \;+\; \eta \sum_{c\in\mathcal{C}} \Big(\nabla_{x_t} \log p(x_t \mid c_+) - \nabla_{x_t} \log p(x_t \mid c_-)\Big). \tag{9}$$

Intuitively, each term in the sum defines a "concept direction" (each element of $\mathcal{C}$ is a base/positive/negative tuple) that pushes the sample toward $c_+$ and away from $c_-$. Composing multiple concepts therefore corresponds to adding their directions in score space.

This log-likelihood–ratio view provides a simple theoretical basis for both single-concept steering and multi-concept composition in our training-free formulation and in CS-style methods more broadly.

### A.7 CONNECTION BETWEEN SCORE-BASED GUIDANCE AND FLOW-MATCHING VELOCITY FIELDS

Diffusion-based models and flow-matching (FM) models differ in the quantity predicted by the network, but both ultimately define a *vector field* that drives a sample $x_t$ toward the data manifold. Classical diffusion models predict a noise vector $\epsilon_\omega(x_t, c, t)$, while rectified flow and FM models predict a velocity field $v_\omega(x_t, c, t)$ that transports $x_t$ along a continuous-time flow. Despite these differences, guidance methods such as classifier-free guidance (CFG) and Concept Sliders operate by forming *linear combinations* of conditional vector fields. Below we make this correspondence explicit.

**From noise prediction to velocity fields.** In standard diffusion models, the score function satisfies

$$s_\omega(x_t, c, t) = \nabla_{x_t} \log p(x_t \mid c) \approx \frac{1}{\sigma_t}\big(\epsilon_\omega(x_t, c, t) - \epsilon_{\text{base}}(x_t, t)\big),$$

which shows that linear combinations of scores correspond to linear combinations of noise predictions. Concept Sliders rely on this property by combining the score (or equivalently, the predicted noise) under three prompts:

$$s_{\text{CS}}(x_t) = s_{\text{base}}(x_t) + \varepsilon\big(s_+(x_t) - s_-(x_t)\big).$$

**Flow matching: identical structure under a different parameterization.** In flow-matching and rectified flow models, the network predicts a velocity

$$v_\omega(x_t, c, t) = \frac{\mathrm{d}x_t}{\mathrm{d}t},$$

and generation is performed by integrating this vector field forward. Crucially, diffusion noise and FM velocity fields are linearly related:

$$v_\omega(x_t, c, t) = a_t\, x_t + b_t\, \epsilon_\omega(x_t, c, t),$$

for model-dependent coefficients $(a_t, b_t)$.[1] Thus, any linear combination of noise predictions implies the same linear combination of velocities:

$$v_{\text{base}} + \varepsilon(v_+ - v_-) \quad\Longleftrightarrow\quad \epsilon_{\text{base}} + \varepsilon(\epsilon_+ - \epsilon_-).$$

**Concept Sliders for flow matching.** Because the generative update in FM is governed directly by $v_\omega$, our inference-time formulation applies:

$$v_{\text{CS}}(x_t) = v_{\text{base}}(x_t) + \varepsilon\Big(v_+(x_t) - v_-(x_t)\Big).$$

This is exactly the same expression used for noise-based diffusion models, with the predicted velocities replacing predicted noise. The sampler then integrates this modified velocity field instead of the original one.

---

[1] For rectified flows, $a_t = 1$ and $b_t$ is a known scalar depending on the schedule; see Liu et al. (2022) for details.

**General applicability to vector-field generative models.** The above derivation highlights that the only requirement for our method is the ability to query the model under different conditional prompts and *linearly combine the resulting vector fields*. Therefore, our approach is directly applicable to:

- score-based diffusion models (noise prediction),

- rectified flow and flow-matching models (velocity prediction),

- any future generative model that exposes a differentiable conditional vector field $F_\omega(x_t, c, t)$.

This generality explains why our method works seamlessly across Stable Diffusion 3, LTX-Video, and classical SD-based backbones without architectural modification.

## B  DIFFUSION MODELS IMPLEMENTATION DETAILS

### B.1  IMAGE

We use Stable Diffusion 1-4 Rombach et al. (2022) for images with resolution $512 \times 512$. The diffusion process consists of 50 total timesteps, and for the methods where it is relevant a splitting point at timestep $k = 15$, and for methods with LoRA weights the starting point to combine the weights is $startnoise = 800$. The guidance scale used in the diffusion for the neutral prompt is $g_1 = 7.5$ for classifier-free guidance, and the scheduler used is the LMSDiscreteScheduler. Stable Diffusion 1.5 was additionally used for qualitative visualizations included in several illustrative figures, while all quantitative results were obtained exclusively with Stable Diffusion 1.4.

### B.2  VIDEO

For quantitative results we use CogVideoX-2b Yang et al. (2024) with a resolution of $480 \times 720$, with 24 frames. The diffusion process consists of 50 total timesteps, and for the methods where it is relevant a splitting point at timestep $k = 12$, and for methods with LoRA weights the starting point to combine the weights is $start-noise = 800$. The guidance scale used in the diffusion for the neutral prompt is $g_1 = 7.5$ for classifier free guidance and the scheduler used is CogVideoXDPMScheduler. For qualitive results and comparisons we also used LTX-Video HaCohen et al. (2024) as a backbone, with resolution of $1344 \times 768$ with a total of 97 frames. The diffusion consists of 30 steps overall with splitting points tested at $k = 10$.

### B.3  AUDIO

We use Stable Audio Open 1.0 Evans et al. (2025) with an implementation based on the `stable-audio-tools` library and a Heun sampler (Karras et al., 2022). The sampler hyper-parameters are $\sigma_{\min} = 0.3$, $\sigma_{\max} = 500$, steps $= 36$, $\rho = 3$, $s_{\text{churn}} = 0$, and a guidance scales of $g_1 = 7$. We generate $10\,\text{s}$ of audio at a 44,100 Hz sampling rate. For all methods where it is relevant, we branch at $k = 4$. In our setting, we use $\epsilon_\theta^{\text{mod}}$ in both the first- and second-order steps of the sampler.

## C  ADDITIONAL RESULTS

### C.1  QUALITATIVE EXPERIMENTS

**Images** Examples for comparison between methods can be found in Figs. 11, and comparisons of methods with and without the ASTD add-on can be found in Fig. 12.

**Video** Examples for video sliders with the LTX-Video backbone can be found in Fig. 14, 15. Comparison between our method and CS can be found in Fig. 13.

Table 4: ASTD ablation on SD3.

| | Conceptual Range ↑ | Conceptual Smoothness ↓ | Semantic Preservation ↓ | Overall Score ↑ |
|---|---|---|---|---|
| **w/ ASTD** | **2.13** | **0.27** | 0.018 | **1.36** |
| **w/o ASTD** | 1.33 | 0.29 | **0.017** | 0.60 |

## C.2  QUANTITATIVE EXPERIMENTS

### C.2.1  IMAGES

**Concepts:**  *Age*, *Chubby*, *Smiling*, *Surprised*, *Lipstick*, *Glasses*, *Beard*, *Car damage*, *Cooked food*, and *Cluttered room*

The results were computed over the 10 concepts, each one generated sliders over 100 different seeds, and with 7 scales. The generic scales (without the ASTD add-on) are [-3, -2, -1, 0, 1, 2, 3], while the scales found with the ASTD add-on vary per concept and per method. See table 5 for full results for all concepts.

### C.2.2  VIDEO

**Concepts:**  *mountain hikers*, *car type*, *mountain type*, *tree leafiness*, *cat size*, *shore waves near a lighthouse*, *ocean waves on a sailboat*, *river current*, *butterfly wing flapping*, and *walking speed on the shore*

For video sliders, the results were computed over 10 concepts, that vary in changing static and dynamic features. each concept was evaluated over 10 different seeds with 7 scales per slider. The generic scales are [-3, -2, -1, 0, 1, 2, 3], while the scales found with ASTD vary for each concept and method. See table 6 for all results.

### C.2.3  AUDIO

**Concepts:**  *cat meow*, *ocean waves*, *car engine*, *choir (classical)*, *door slam*, *crowd clapping*, *solo guitar*, *dog bark*, *rainfall*, and *people talking*

The results were computed over the 10 concepts, each one generated sliders over 10 different seeds, and with 7 scales. The generic scales (without the ASTD add-on) are [-3, -2, -1, 0, 1, 2, 3], while the scales found with the ASTD add-on vary per concept and per method. See table 7 for full results for all concepts.

## C.3  DELTA CLIP LIMITATIONS CONT.

In Fig. 2, we show several cases where a visually better slider receives a lower ΔCLIP score, revealing its inability to capture the range and smoothness of transitions. One limitation is shown in Fig. 10b, and occurs when the slider produces an abrupt change at the beginning, after which the images remain constant. Although ΔCLIP assigns a high score due to the large initial jump in CLIP similarity, it doesn't reflect the lack of smooth intermediate progression. Another limitation arises when the change is pronounced only in the negative direction but not in the positive direction, causing the metric to overlook asymmetries and doesn't succeed in capturing both directions, like in Fig. 10a. In contrast, the proposed metrics in Section 4.3 capture both range and smoothness, aligning more closely with the visual results.

## C.4  STABLE DIFFUSION 3 EXPERIMENTATION

We provide additional qualitative examples of concept sliding on SD3 in the supplementary file `stable_diffusion_3.pdf`. We further experiment with SD3 on the image benchmark, evaluating whether ASTD is also effective for this backbone. As shown in Tab.4, ASTD again proves highly beneficial.

## C.5 Choice of $k$

As detailed in the method section, our inference procedure first takes $k$ steps with the base concept only, and then uses the full guided update for the remaining diffusion steps. The choice of $k$ is arbitrary. For all models and concepts, we use a single default $k$ to maintain fairness and consistency. Across modalities, we select $k$ proportional to the total number of inference steps of the backbone; heuristically, we found that $k \approx 30\%$ of the full trajectory offers a good balance between preservation and expressive range across models.

In our experiments, we observed several heuristics relating the choice of $k$ (the number of timesteps where the concept slider is active) to the trade-off between semantic preservation, conceptual range, and smoothness. When we **reduce** $k$, semantic preservation tends to degrade, while conceptual range and smoothness improve. Intuitively, applying the concept direction only for a *small* portion of the trajectory gives the model less opportunity to "anchor" the sample to the original semantic identity, while the concept shift is amplified as the trajectory is driven further by the guided steps. In contrast, when we **increase** $k$, we observe the opposite behavior: preservation improves, whereas range and smoothness often degrade. This aligns with the interpretation that applying the concept direction over *many* steps repeatedly pulls the sample back toward the base manifold, suppressing large conceptual deviations but retaining more of the original semantics.

For future work, we believe that exploring an adaptive or concept-dependent choice of $k$ is a promising direction. Furthermore, we provide visual illustrations of these heuristics in the new supplementary material (`k_comparisons.pdf`).

## C.6 Slider Composition

Our method naturally supports compositional edits. As shown in Fig. 7, we simultaneously adjust the *smile*, *glasses*, and *lipstick* concepts, demonstrating controlled manipulation across multiple attributes. To create the combination of the concepts, the prompts of every concept are concatenated one after another and are used jointly in the method as a single prompt, forming the basis for the slider composition. In addition, in Fig. 8 there is a visual representation of combinations of composition with different scales in each concept.

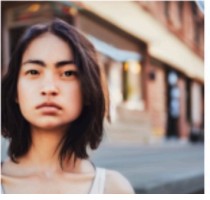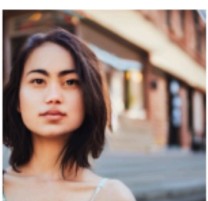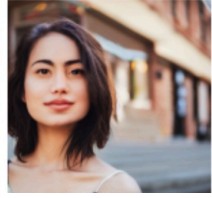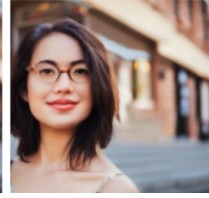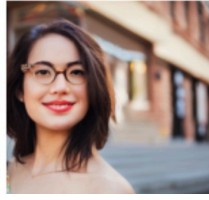

Figure 7: Slider composition: Three concepts "smile", "glasses" and "lipstick" combined together.

## C.7 Correlation to Different Aligner Baselines

Our proposed metric relies on CLIP/ViCLIP-based alignment scores, which are known to exhibit biases and sensitivity to prompts. To assess the robustness of our approach to the choice of aligner, we conducted a correlation analysis using an alternative baseline aligner. In Fig. 9, we visualize the relationship between the scores obtained using CLIP and those computed with BLIP Li et al. (2022). The analysis was performed across 30 examples, each evaluated with 7 different scale levels per concept. Overall, the Pearson correlation between CLIP- and BLIP-derived scores was $r = 0.65$ with statistical significance of $p < 0.001$, indicating a moderate but reliable alignment between the two score sources.

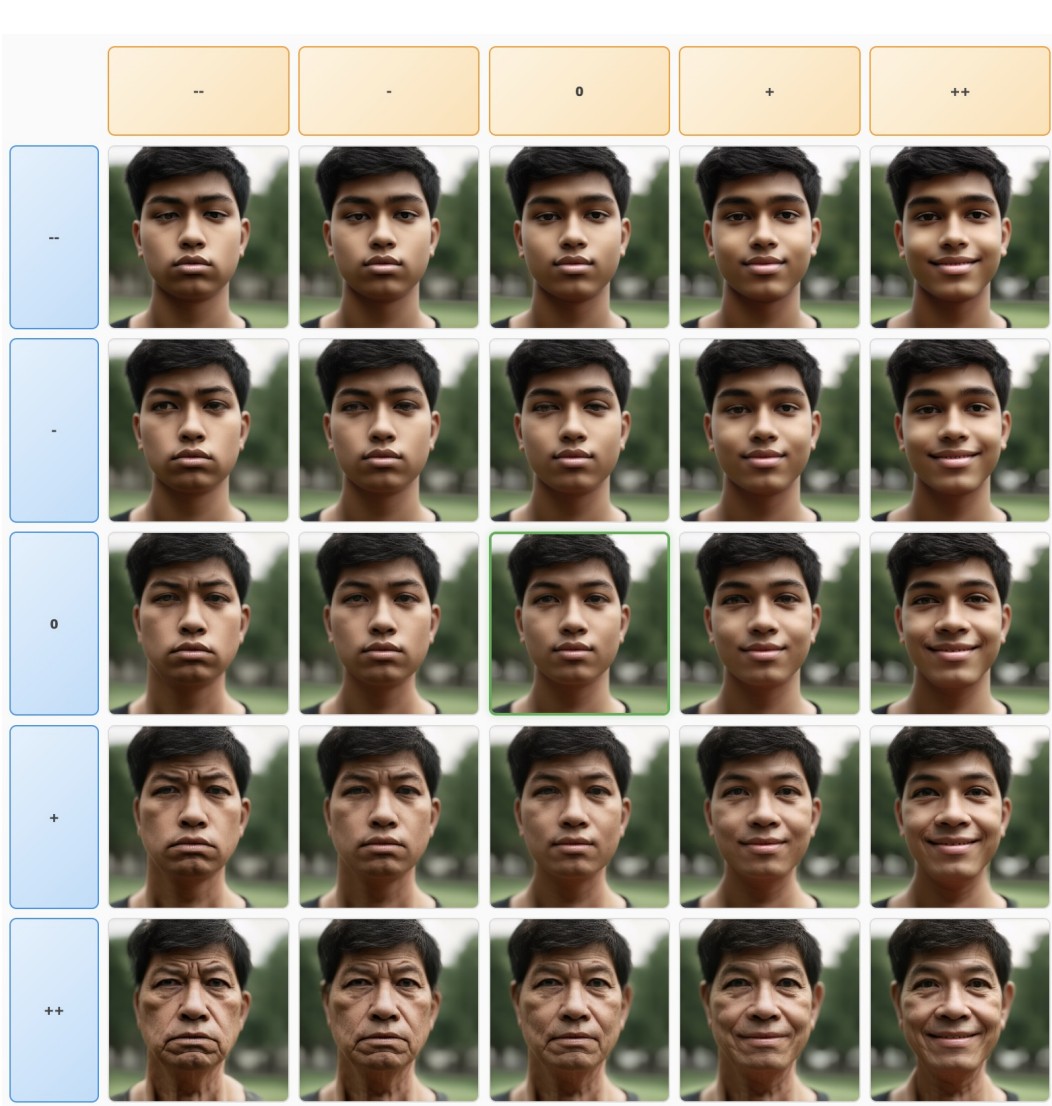

Figure 8: Slider composition grid: The concept "age" increases across the rows, while the concept "smiling" increases across the columns, showing how different combinations of the two concepts are mixed.

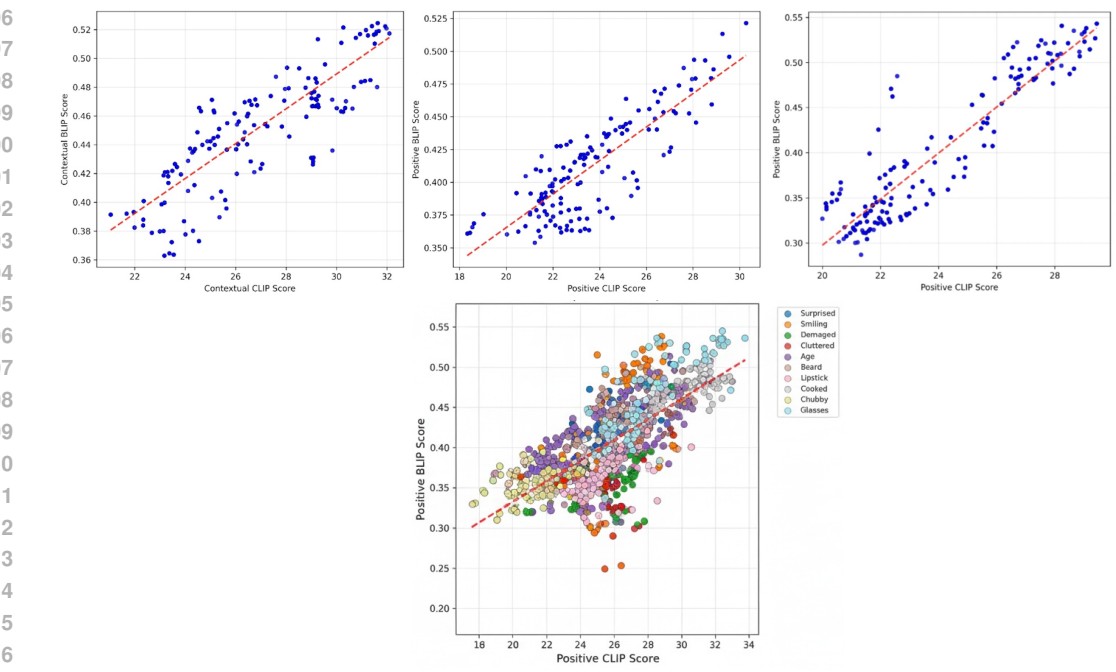

Figure 9: Correlation visualization using two aligners. The top image shows examples per concept while the lower image represents the overall correlation across all concepts.

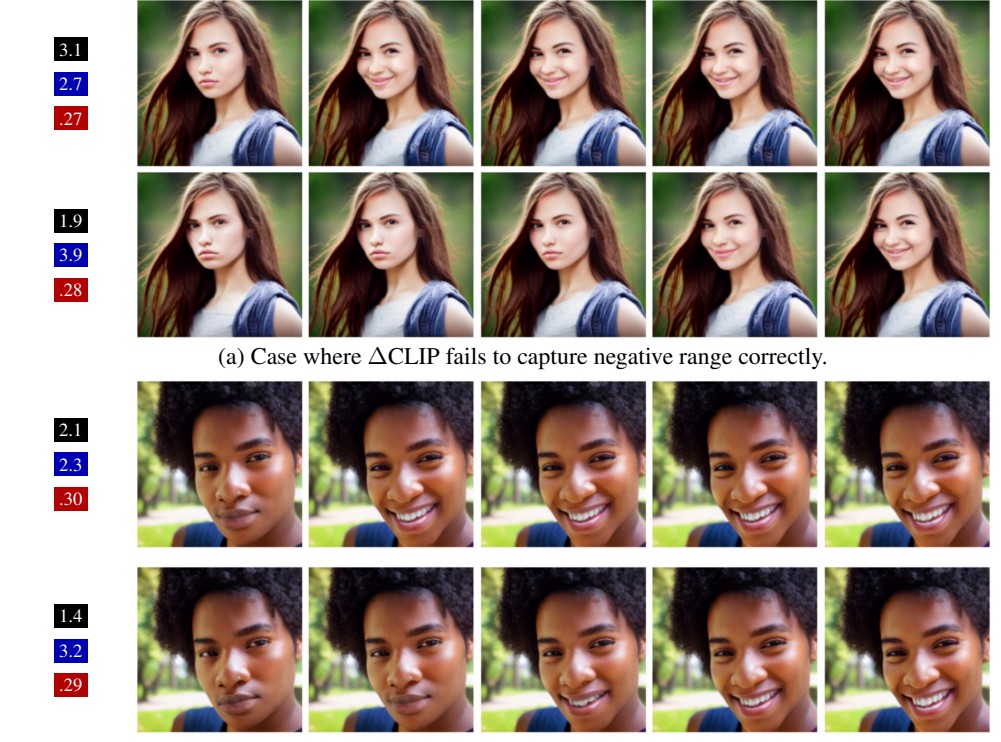

(a) Case where $\Delta$CLIP fails to capture negative range correctly.

(b) Case where $\Delta$CLIP fails to reflect interval smoothness.

Figure 10: Examples of limitations for the $\Delta$CLIP metric (black boxes) that CR and CSM metric results (blue and red boxes, respectively) capture.

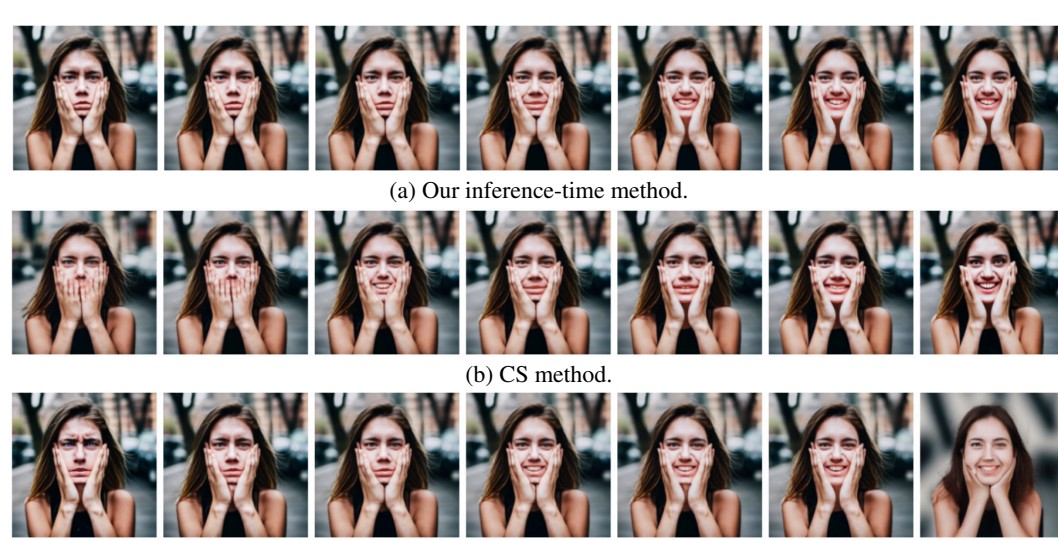

(a) Our inference-time method.

(b) CS method.

(c) Text Embeddings (TE) variant.

Figure 11: Image slider of concept "smiling" with different methods.

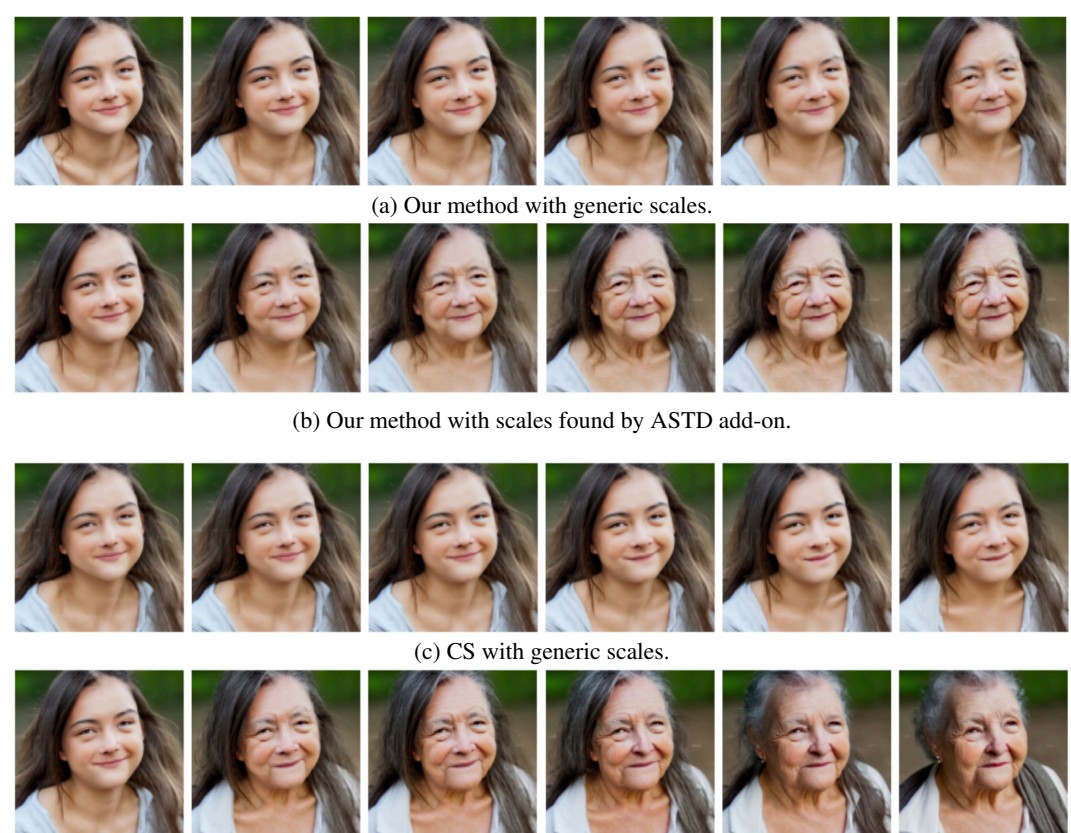

(a) Our method with generic scales.

(b) Our method with scales found by ASTD add-on.

(c) CS with generic scales.

(d) CS with scales found by ASTD add-on.

Figure 12: Comparison of our method and CS with and without ASTD.

1404
1405
1406
1407
1408
1409
1410
1411
1412
1413
1414
1415
1416
1417
1418
1419
1420
1421
1422
1423
1424
1425
1426
1427
1428
1429
1430
1431
1432
1433
1434
1435
1436
1437
1438
1439
1440
1441
1442
1443
1444
1445
1446
1447
1448
1449
1450
1451
1452
1453
1454
1455
1456
1457

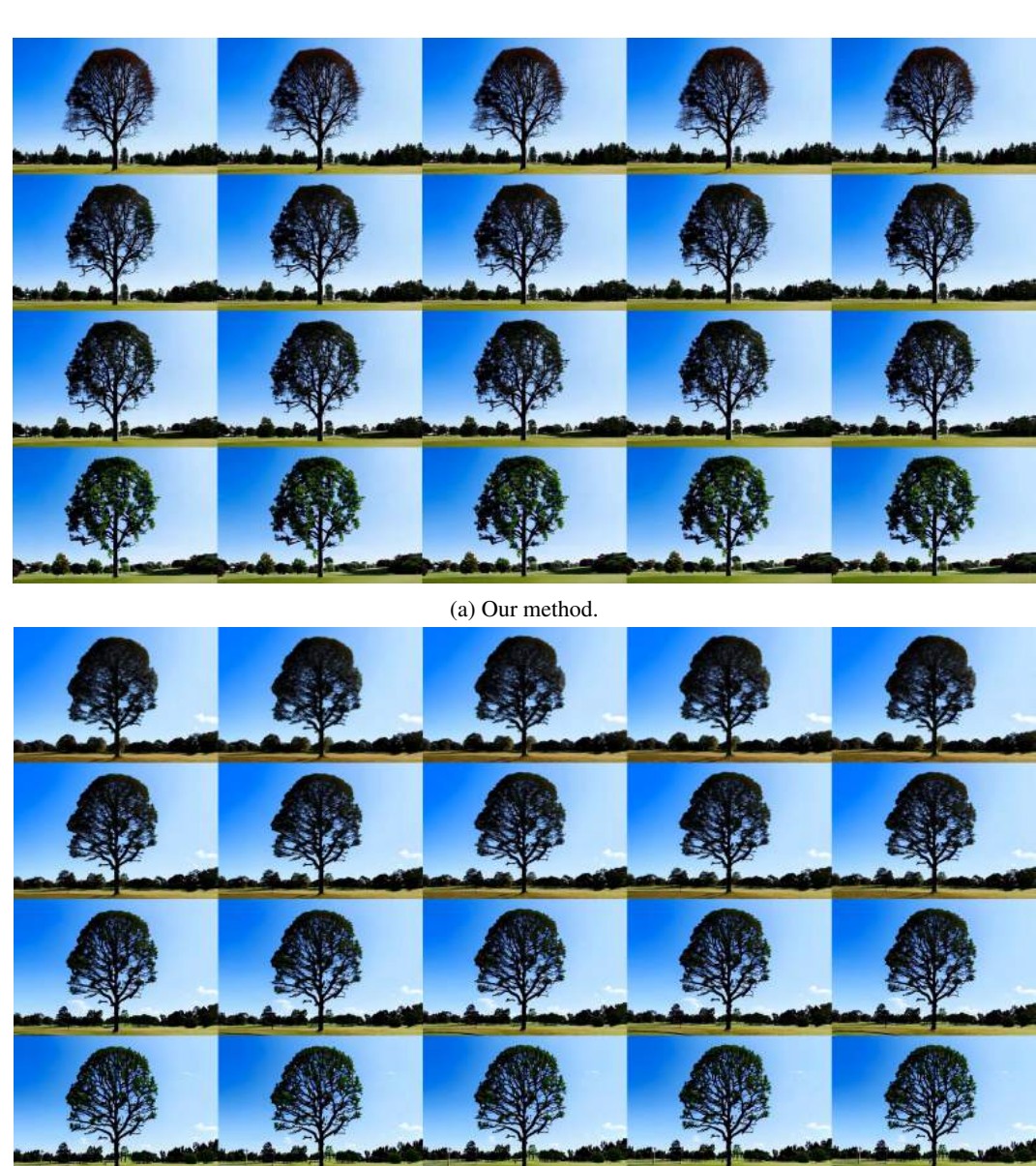

(a) Our method.

(b) CS method.

Figure 13: Video Slider with our inference-time method (top) and with CS method (bottom) with CogVideoX backbone over concept "leafy tree". The rows depict increasing scales and the columns different frames.

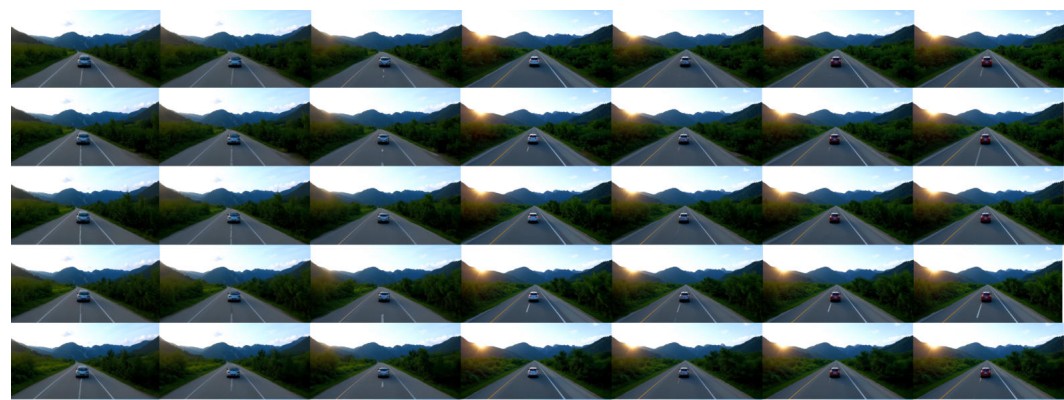

Figure 14: Video slider on concept "car style" computed with LTX-Video as backbone. The columns represent different scales on the slider changing the car style and the rows are the frames of the video.

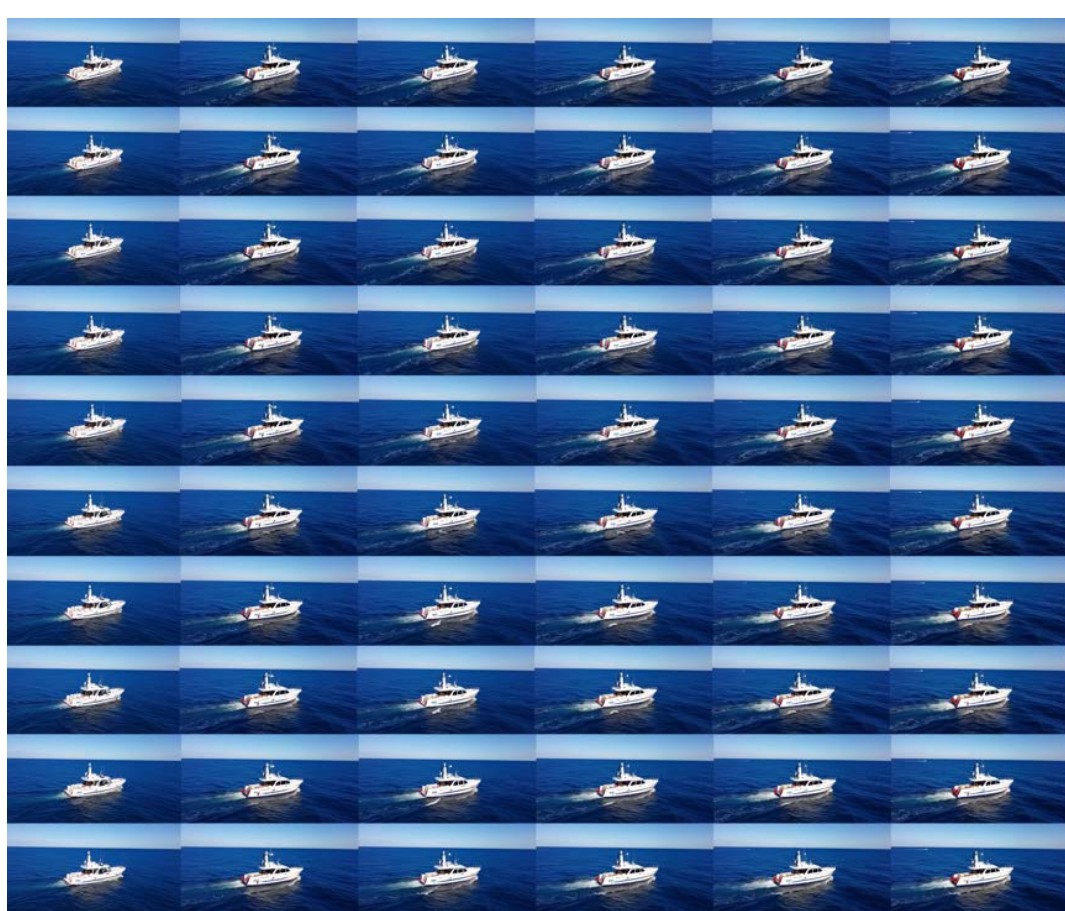

Figure 15: Video slider on concept "sailboat waves" computed with LTX-Video as backbone. The columns represent different scales on the slider changing the waves intensity and the rows are the frames of the video.

Table 5: Overall results for images evaluation, for short R=Conceptual Range, S=Smoothness, P=Preservation, the values with a dash are sliders where the images weren't changed along the slider.

| | CS w/o ASTD | CS w/ ASTD | Ours w/o ASTD | Ours w/ ASTD | TE w/o ASTD | TE w/ ASTD | PromptSliders w/o ASTD | PromptSliders w/ ASTD | P2P w/o ASTD | P2P w/ ASTD |
|---|---|---|---|---|---|---|---|---|---|---|
| **Concept: Age** | | | | | | | | | | |
| R↑ | $2.27 \pm 1.24$ | $4.64 \pm 1.08$ | $3.95 \pm 1.39$ | $5.10 \pm 1.37$ | $2.53 \pm 1.30$ | $1.79 \pm 0.62$ | $-0.01 \pm 0.02$ | $0.11 \pm 0.03$ | $0.00 \pm 0.00$ | $0.00 \pm 0.00$ |
| S↓ | $0.28 \pm 0.01$ | $0.27 \pm 0.01$ | $0.28 \pm 0.01$ | $0.27 \pm 0.01$ | $0.28 \pm 0.01$ | $0.28 \pm 0.01$ | $0.28 \pm 0.01$ | $0.29 \pm 0.01$ | $0.30 \pm 0.00$ | $0.29 \pm 0.01$ |
| P↓ | $0.02 \pm 0.03$ | $0.06 \pm 0.01$ | $0.02 \pm 0.01$ | $0.03 \pm 0.03$ | $0.02 \pm 0.02$ | $0.01 \pm 0.01$ | $0.04 \pm 0.02$ | $0.04 \pm 0.01$ | $0.00 \pm 0.00$ | — |
| **Concept: Beard** | | | | | | | | | | |
| R↑ | $1.69 \pm 1.57$ | $3.59 \pm 0.87$ | $2.12 \pm 1.63$ | $3.63 \pm 1.03$ | $0.97 \pm 1.37$ | $0.28 \pm 1.26$ | $-0.54 \pm 0.05$ | $0.00 \pm 0.00$ | $0.00 \pm 0.00$ | $0.00 \pm 0.00$ |
| S↓ | $0.30 \pm 0.01$ | $0.29 \pm 0.01$ | $0.29 \pm 0.01$ | $0.30 \pm 0.01$ | $0.30 \pm 0.01$ | $0.30 \pm 0.01$ | $0.28 \pm 0.01$ | $0.24 \pm 0.01$ | $0.28 \pm 0.01$ | $0.27 \pm 0.01$ |
| P↓ | $0.02 \pm 0.03$ | $0.06 \pm 0.01$ | $0.01 \pm 0.01$ | $0.03 \pm 0.02$ | $0.03 \pm 0.02$ | $0.01 \pm 0.01$ | $0.19 \pm 0.03$ | $0.00 \pm 0.00$ | $0.00 \pm 0.00$ | $0.19 \pm 0.01$ |
| **Concept: Chubby** | | | | | | | | | | |
| R↑ | $1.67 \pm 1.60$ | $4.10 \pm 1.44$ | $3.89 \pm 1.61$ | $6.67 \pm 1.68$ | $3.64 \pm 1.83$ | $1.43 \pm 1.19$ | $-0.93 \pm 0.05$ | $-0.39 \pm 0.19$ | $0.67 \pm 1.45$ | $0.78 \pm 1.46$ |
| S↓ | $0.29 \pm 0.01$ | $0.29 \pm 0.01$ | $0.28 \pm 0.01$ | $0.28 \pm 0.01$ | $0.29 \pm 0.01$ | $0.30 \pm 0.01$ | $0.32 \pm 0.01$ | $0.31 \pm 0.01$ | $0.30 \pm 0.01$ | $0.30 \pm 0.01$ |
| P↓ | $0.02 \pm 0.02$ | $0.07 \pm 0.02$ | $0.01 \pm 0.01$ | $0.03 \pm 0.02$ | $0.03 \pm 0.02$ | $0.01 \pm 0.01$ | $0.17 \pm 0.03$ | $0.03 \pm 0.01$ | $0.20 \pm 0.01$ | $0.19 \pm 0.01$ |
| **Concept: Cluttered Room** | | | | | | | | | | |
| R↑ | $2.01 \pm 0.72$ | $2.34 \pm 0.77$ | $1.14 \pm 0.86$ | $0.14 \pm 0.21$ | $1.19 \pm 1.05$ | $0.36 \pm 0.50$ | $-0.35 \pm 0.09$ | $-0.31 \pm 0.12$ | $0.81 \pm 0.84$ | $2.13 \pm 0.46$ |
| S↓ | $0.30 \pm 0.01$ | $0.28 \pm 0.01$ | $0.28 \pm 0.01$ | $0.27 \pm 0.01$ | $0.29 \pm 0.01$ | $0.30 \pm 0.01$ | $0.30 \pm 0.01$ | $0.30 \pm 0.01$ | $0.28 \pm 0.01$ | $0.29 \pm 0.00$ |
| P↓ | $0.10 \pm 0.04$ | $0.17 \pm 0.02$ | $0.03 \pm 0.02$ | $0.01 \pm 0.00$ | $0.07 \pm 0.03$ | $0.05 \pm 0.03$ | $0.22 \pm 0.03$ | $0.23 \pm 0.04$ | $0.09 \pm 0.01$ | $0.19 \pm 0.02$ |
| **Concept: Cooked Food** | | | | | | | | | | |
| R↑ | $0.47 \pm 0.62$ | $0.00 \pm 0.00$ | $0.57 \pm 0.38$ | $0.11 \pm 0.18$ | $0.65 \pm 0.66$ | $0.00 \pm 0.00$ | $0.17 \pm 0.07$ | $0.00 \pm 0.00$ | $0.23 \pm 0.34$ | $0.24 \pm 0.14$ |
| S↓ | $0.28 \pm 0.00$ | $0.28 \pm 0.00$ | $0.28 \pm 0.00$ | $0.28 \pm 0.00$ | $0.29 \pm 0.00$ | $0.29 \pm 0.00$ | $0.28 \pm 0.02$ | $0.29 \pm 0.02$ | $0.30 \pm 0.01$ | $0.29 \pm 0.00$ |
| P↓ | $0.03 \pm 0.02$ | — | $0.01 \pm 0.00$ | $0.00 \pm 0.00$ | $0.03 \pm 0.01$ | — | $0.23 \pm 0.03$ | — | $0.05 \pm 0.01$ | $0.10 \pm 0.01$ |
| **Concept: Damaged Car** | | | | | | | | | | |
| R↑ | $0.65 \pm 0.94$ | $1.37 \pm 0.96$ | $1.04 \pm 0.82$ | $0.43 \pm 0.35$ | $0.01 \pm 0.74$ | $0.14 \pm 0.44$ | $0.25 \pm 0.04$ | $0.02 \pm 0.22$ | $1.15 \pm 1.09$ | $1.34 \pm 1.17$ |
| S↓ | $0.28 \pm 0.00$ | $0.28 \pm 0.00$ | $0.29 \pm 0.01$ | $0.30 \pm 0.00$ | $0.29 \pm 0.01$ | $0.29 \pm 0.01$ | $0.28 \pm 0.01$ | $0.29 \pm 0.01$ | $0.28 \pm 0.01$ | $0.29 \pm 0.01$ |
| P↓ | $0.04 \pm 0.03$ | $0.08 \pm 0.01$ | $0.01 \pm 0.01$ | $0.01 \pm 0.01$ | $0.04 \pm 0.02$ | $0.01 \pm 0.01$ | $0.17 \pm 0.01$ | $0.19 \pm 0.03$ | $0.13 \pm 0.02$ | $0.08 \pm 0.01$ |
| **Concept: Glasses** | | | | | | | | | | |
| R↑ | $3.46 \pm 0.85$ | $3.81 \pm 0.61$ | $3.30 \pm 0.88$ | $3.48 \pm 0.93$ | $3.26 \pm 1.00$ | $2.86 \pm 1.27$ | $0.56 \pm 0.26$ | $0.47 \pm 0.15$ | $0.00 \pm 0.00$ | $0.00 \pm 0.00$ |
| S↓ | $0.27 \pm 0.01$ | $0.27 \pm 0.01$ | $0.28 \pm 0.01$ | $0.28 \pm 0.01$ | $0.28 \pm 0.01$ | $0.20 \pm 0.01$ | $0.32 \pm 0.01$ | $0.29 \pm 0.01$ | $0.28 \pm 0.01$ | $0.23 \pm 0.01$ |
| P↓ | $0.03 \pm 0.03$ | $0.04 \pm 0.00$ | $0.01 \pm 0.01$ | $0.02 \pm 0.01$ | $0.02 \pm 0.01$ | $0.02 \pm 0.01$ | $0.05 \pm 0.03$ | $0.03 \pm 0.01$ | — | — |
| **Concept: Lipstick** | | | | | | | | | | |
| R↑ | $1.22 \pm 0.92$ | $0.11 \pm 0.27$ | $2.42 \pm 1.08$ | $2.26 \pm 1.05$ | $2.28 \pm 1.22$ | $0.40 \pm 0.67$ | $-2.75 \pm 0.81$ | $-0.00 \pm 0.04$ | $0.00 \pm 0.00$ | $0.00 \pm 0.00$ |
| S↓ | $0.27 \pm 0.01$ | $0.29 \pm 0.01$ | $0.27 \pm 0.01$ | $0.27 \pm 0.01$ | $0.27 \pm 0.01$ | $0.28 \pm 0.01$ | $0.29 \pm 0.01$ | $0.29 \pm 0.01$ | $0.30 \pm 0.01$ | $0.21 \pm 0.01$ |
| P↓ | $0.02 \pm 0.02$ | $0.00 \pm 0.00$ | $0.01 \pm 0.01$ | $0.01 \pm 0.01$ | $0.05 \pm 0.03$ | $0.01 \pm 0.01$ | $0.13 \pm 0.03$ | $0.00 \pm 0.00$ | $0.00 \pm 0.00$ | $0.06 \pm 0.01$ |
| **Concept: Surprised** | | | | | | | | | | |
| R↑ | $0.23 \pm 0.43$ | $0.41 \pm 0.55$ | $0.46 \pm 0.44$ | $0.82 \pm 0.54$ | $-0.20 \pm 0.51$ | $0.54 \pm 0.63$ | $-0.86 \pm 0.04$ | $0.04 \pm 0.22$ | $0.00 \pm 0.00$ | $0.00 \pm 0.00$ |
| S↓ | $0.30 \pm 0.01$ | $0.30 \pm 0.01$ | $0.29 \pm 0.01$ | $0.28 \pm 0.01$ | $0.29 \pm 0.01$ | $0.29 \pm 0.01$ | $0.30 \pm 0.01$ | $0.31 \pm 0.01$ | $0.31 \pm 0.00$ | $0.25 \pm 0.01$ |
| P↓ | $0.01 \pm 0.01$ | $0.06 \pm 0.01$ | $0.01 \pm 0.01$ | $0.02 \pm 0.01$ | $0.03 \pm 0.02$ | $0.01 \pm 0.01$ | $0.37 \pm 0.02$ | $0.03 \pm 0.01$ | $0.06 \pm 0.01$ | $0.22 \pm 0.02$ |
| **Concept: Smiling** | | | | | | | | | | |
| R↑ | $2.30 \pm 1.86$ | $5.05 \pm 1.22$ | $4.54 \pm 1.41$ | $5.87 \pm 1.28$ | $2.95 \pm 1.60$ | $1.45 \pm 1.46$ | $-4.70 \pm 0.47$ | $0.85 \pm 0.62$ | $0.99 \pm 1.13$ | $1.17 \pm 1.63$ |
| S↓ | $0.29 \pm 0.01$ | $0.29 \pm 0.01$ | $0.28 \pm 0.01$ | $0.28 \pm 0.01$ | $0.30 \pm 0.01$ | $0.31 \pm 0.01$ | $0.29 \pm 0.01$ | $0.31 \pm 0.02$ | $0.30 \pm 0.01$ | $0.29 \pm 0.01$ |
| P↓ | $0.02 \pm 0.02$ | $0.02 \pm 0.01$ | $0.00 \pm 0.01$ | $0.01 \pm 0.02$ | $0.02 \pm 0.01$ | $0.06 \pm 0.04$ | $0.20 \pm 0.01$ | $0.04 \pm 0.01$ | $0.06 \pm 0.01$ | $0.06 \pm 0.01$ |
| **Overall Score (Averaged across concepts)** | | | | | | | | | | |
| R↑ | $1.60 \pm 1.08$ | $2.54 \pm 0.78$ | $2.34 \pm 1.05$ | $2.85 \pm 0.86$ | $1.73 \pm 1.13$ | $0.93 \pm 0.80$ | $-0.92 \pm 0.74$ | $0.08 \pm 0.37$ | $0.38 \pm 0.48$ | $0.57 \pm 0.55$ |
| S↓ | $0.29 \pm 0.01$ | $0.28 \pm 0.01$ | $0.28 \pm 0.01$ | $0.28 \pm 0.01$ | $0.29 \pm 0.01$ | $0.28 \pm 0.01$ | $0.30 \pm 0.02$ | $0.29 \pm 0.01$ | $0.29 \pm 0.01$ | $0.27 \pm 0.01$ |
| P↓ | $0.03 \pm 0.02$ | $0.06 \pm 0.03$ | $0.01 \pm 0.01$ | $0.02 \pm 0.01$ | $0.03 \pm 0.02$ | $0.02 \pm 0.01$ | $0.18 \pm 0.04$ | $0.06 \pm 0.03$ | $0.06 \pm 0.03$ | $0.13 \pm 0.01$ |

Table 6: Comparison of video results with and without ASTD. Metrics include Range (R), Smoothness (S), and Preservation (P), each measured for Static (S) and Dynamic (D) aspects.

| Concept | Metric | CS w/o ASTD | CS w/ ASTD | TE w/o ASTD | TE w/ ASTD | Ours w/o ASTD | Ours w/ ASTD | PromptSliders w/o ASTD | PromptSliders w/ ASTD |
|---|---|---|---|---|---|---|---|---|---|
| Mountain Hiking | R(S)↑ | $2.58 \pm 0.43$ | $0.81 \pm 0.46$ | $0.94 \pm 0.25$ | $1.19 \pm 0.53$ | $1.08 \pm 0.44$ | $1.24 \pm 0.52$ | $-1.24 \pm 0.01$ | $-1.22 \pm 0.01$ |
| | R(D)↑ | $0.04 \pm 0.00$ | $0.01 \pm 0.01$ | $0.02 \pm 0.00$ | $0.01 \pm 0.00$ | $0.01 \pm 0.01$ | $0.02 \pm 0.01$ | $-0.01 \pm 0.01$ | $-0.01 \pm 0.01$ |
| | P(S)↓ | $0.10 \pm 0.02$ | $0.08 \pm 0.02$ | $0.06 \pm 0.02$ | $0.06 \pm 0.03$ | $0.03 \pm 0.01$ | $0.03 \pm 0.01$ | $0.09 \pm 0.01$ | $0.08 \pm 0.01$ |
| | P(D)↓ | $-0.54 \pm 0.26$ | $-0.44 \pm 0.20$ | $-0.39 \pm 0.27$ | $-0.46 \pm 0.31$ | $-0.28 \pm 0.16$ | $-0.28 \pm 0.16$ | $-1.02 \pm 0.01$ | $-1.00 \pm 0.01$ |
| | S(S)↓ | $0.33 \pm 0.01$ | $0.34 \pm 0.01$ | $0.34 \pm 0.01$ | $0.33 \pm 0.01$ | $0.34 \pm 0.01$ | $0.34 \pm 0.01$ | $0.35 \pm 0.01$ | $0.36 \pm 0.01$ |
| | S(D)↑ | $0.44 \pm 0.02$ | $0.44 \pm 0.01$ | $0.43 \pm 0.02$ | $0.44 \pm 0.01$ | $0.44 \pm 0.02$ | $0.43 \pm 0.02$ | $0.41 \pm 0.01$ | $0.43 \pm 0.01$ |
| Sailboat | R(S)↑ | $4.05 \pm 0.35$ | $4.98 \pm 0.79$ | $0.21 \pm 0.37$ | $0.68 \pm 0.43$ | $0.90 \pm 0.34$ | $2.05 \pm 0.44$ | $0.07 \pm 0.01$ | $-0.28 \pm 0.01$ |
| | R(D)↑ | $0.06 \pm 0.01$ | $0.08 \pm 0.02$ | $0.01 \pm 0.00$ | $0.01 \pm 0.01$ | $0.01 \pm 0.00$ | $0.03 \pm 0.01$ | $-0.01 \pm 0.01$ | $-0.01 \pm 0.01$ |
| | P(S)↓ | $0.08 \pm 0.03$ | $0.11 \pm 0.03$ | $0.02 \pm 0.00$ | $0.02 \pm 0.00$ | $0.02 \pm 0.00$ | $0.03 \pm 0.01$ | $0.09 \pm 0.01$ | $0.08 \pm 0.01$ |
| | P(D)↓ | $-3.54 \pm 1.54$ | $-3.89 \pm 1.56$ | $-1.40 \pm 0.49$ | $-1.35 \pm 0.46$ | $-1.19 \pm 0.47$ | $-1.56 \pm 0.62$ | $-1.51 \pm 0.01$ | $-1.53 \pm 0.01$ |
| | S(S)↓ | $0.29 \pm 0.01$ | $0.28 \pm 0.00$ | $0.32 \pm 0.01$ | $0.32 \pm 0.01$ | $0.31 \pm 0.01$ | $0.31 \pm 0.01$ | $0.33 \pm 0.01$ | $0.33 \pm 0.01$ |
| | S(D)↑ | $0.41 \pm 0.01$ | $0.41 \pm 0.01$ | $0.45 \pm 0.02$ | $0.44 \pm 0.02$ | $0.43 \pm 0.01$ | $0.44 \pm 0.02$ | $0.43 \pm 0.01$ | $0.47 \pm 0.01$ |
| Car Type | R(S)↑ | $-0.17 \pm 1.09$ | $2.08 \pm 0.89$ | $4.55 \pm 0.68$ | $4.42 \pm 1.01$ | $2.97 \pm 0.80$ | $2.93 \pm 0.78$ | $-0.21 \pm 0.01$ | $-0.28 \pm 0.01$ |
| | R(D)↑ | $0.00 \pm 0.01$ | $0.02 \pm 0.01$ | $0.06 \pm 0.01$ | $0.06 \pm 0.02$ | $0.05 \pm 0.02$ | $0.05 \pm 0.02$ | $0.01 \pm 0.01$ | $0.01 \pm 0.01$ |
| | P(S)↓ | $0.10 \pm 0.03$ | $0.14 \pm 0.03$ | $0.07 \pm 0.02$ | $0.07 \pm 0.02$ | $0.05 \pm 0.01$ | $0.05 \pm 0.01$ | $0.07 \pm 0.01$ | $0.07 \pm 0.01$ |
| | P(D)↓ | $-5.11 \pm 3.41$ | $-5.55 \pm 3.72$ | $-4.80 \pm 2.28$ | $-4.82 \pm 2.22$ | $-3.76 \pm 1.75$ | $-3.80 \pm 1.82$ | $-1.01 \pm 0.01$ | $-1.09 \pm 0.01$ |
| | S(S)↓ | $0.36 \pm 0.01$ | $0.32 \pm 0.01$ | $0.32 \pm 0.01$ | $0.31 \pm 0.01$ | $0.33 \pm 0.01$ | $0.32 \pm 0.01$ | $0.38 \pm 0.01$ | $0.39 \pm 0.01$ |
| | S(D)↑ | $0.46 \pm 0.02$ | $0.44 \pm 0.02$ | $0.43 \pm 0.02$ | $0.41 \pm 0.01$ | $0.43 \pm 0.01$ | $0.42 \pm 0.01$ | $0.45 \pm 0.01$ | $0.45 \pm 0.01$ |
| River Waves | R(S)↑ | $1.88 \pm 0.89$ | $0.79 \pm 0.51$ | $1.26 \pm 0.69$ | $0.22 \pm 0.31$ | $0.66 \pm 0.65$ | $0.61 \pm 0.30$ | $-0.53 \pm 0.01$ | $1.23 \pm 0.01$ |
| | R(D)↑ | $0.03 \pm 0.02$ | $0.01 \pm 0.01$ | $0.01 \pm 0.01$ | $0.00 \pm 0.01$ | $0.01 \pm 0.00$ | $0.01 \pm 0.01$ | $-0.02 \pm 0.01$ | $0.01 \pm 0.01$ |
| | P(S)↓ | $0.09 \pm 0.03$ | $0.06 \pm 0.02$ | $0.04 \pm 0.01$ | $0.03 \pm 0.01$ | $0.03 \pm 0.01$ | $0.03 \pm 0.01$ | $0.14 \pm 0.01$ | $0.10 \pm 0.01$ |
| | P(D)↓ | $-0.39 \pm 0.23$ | $-0.30 \pm 0.17$ | $-0.23 \pm 0.15$ | $-0.22 \pm 0.14$ | $-0.23 \pm 0.12$ | $-0.22 \pm 0.12$ | $-0.36 \pm 0.01$ | $-0.35 \pm 0.01$ |
| | S(S)↓ | $0.32 \pm 0.01$ | $0.33 \pm 0.01$ | $0.32 \pm 0.01$ | $0.33 \pm 0.01$ | $0.32 \pm 0.01$ | $0.32 \pm 0.01$ | $0.31 \pm 0.01$ | $0.31 \pm 0.01$ |
| | S(D)↑ | $0.46 \pm 0.02$ | $0.47 \pm 0.03$ | $0.44 \pm 0.02$ | $0.45 \pm 0.02$ | $0.44 \pm 0.01$ | $0.45 \pm 0.01$ | $0.43 \pm 0.01$ | $0.45 \pm 0.01$ |
| Butterfly Wings | R(S)↑ | $0.16 \pm 0.24$ | $0.06 \pm 0.11$ | $0.04 \pm 0.31$ | $0.20 \pm 0.30$ | $0.14 \pm 0.14$ | $0.01 \pm 0.12$ | $-0.01 \pm 0.01$ | $-0.23 \pm 0.01$ |
| | R(D)↑ | $0.00 \pm 0.00$ | $0.00 \pm 0.01$ | $0.00 \pm 0.00$ | $-0.00 \pm 0.01$ | $0.00 \pm 0.00$ | $0.00 \pm 0.00$ | $-0.01 \pm 0.01$ | $-0.01 \pm 0.01$ |
| | P(S)↓ | $0.04 \pm 0.01$ | $0.04 \pm 0.01$ | $0.05 \pm 0.02$ | $0.05 \pm 0.02$ | $0.04 \pm 0.02$ | $0.04 \pm 0.02$ | $0.21 \pm 0.01$ | $0.20 \pm 0.01$ |
| | P(D)↓ | $-0.82 \pm 0.33$ | $-0.80 \pm 0.32$ | $-1.05 \pm 0.41$ | $-1.06 \pm 0.36$ | $-0.82 \pm 0.32$ | $-0.83 \pm 0.32$ | $-3.15 \pm 0.01$ | $-3.14 \pm 0.01$ |
| | S(S)↓ | $0.30 \pm 0.01$ | $0.30 \pm 0.01$ | $0.30 \pm 0.01$ | $0.30 \pm 0.00$ | $0.30 \pm 0.00$ | $0.30 \pm 0.00$ | $0.30 \pm 0.01$ | $0.30 \pm 0.01$ |
| | S(D)↑ | $0.45 \pm 0.02$ | $0.46 \pm 0.02$ | $0.45 \pm 0.02$ | $0.46 \pm 0.02$ | $0.45 \pm 0.02$ | $0.45 \pm 0.01$ | $0.45 \pm 0.01$ | $0.47 \pm 0.01$ |
| Mountain Type | R(S)↑ | $2.46 \pm 0.91$ | $0.38 \pm 0.44$ | $2.46 \pm 1.10$ | $1.91 \pm 0.87$ | $1.59 \pm 0.63$ | $1.33 \pm 0.80$ | $1.52 \pm 0.01$ | $-0.52 \pm 0.01$ |
| | R(D)↑ | $0.02 \pm 0.01$ | $0.00 \pm 0.01$ | $0.04 \pm 0.01$ | $0.03 \pm 0.01$ | $0.03 \pm 0.01$ | $0.02 \pm 0.01$ | $0.02 \pm 0.01$ | $-0.01 \pm 0.01$ |
| | P(S)↓ | $0.08 \pm 0.04$ | $0.06 \pm 0.03$ | $0.05 \pm 0.03$ | $0.05 \pm 0.03$ | $0.03 \pm 0.01$ | $0.03 \pm 0.01$ | $0.20 \pm 0.01$ | $0.19 \pm 0.01$ |
| | P(D)↓ | $-0.89 \pm 1.46$ | $-0.70 \pm 1.14$ | $-0.81 \pm 1.46$ | $-0.78 \pm 1.39$ | $-0.69 \pm 1.23$ | $-0.64 \pm 1.11$ | $-0.50 \pm 0.01$ | $-0.48 \pm 0.01$ |
| | S(S)↓ | $0.33 \pm 0.01$ | $0.32 \pm 0.01$ | $0.31 \pm 0.01$ | $0.32 \pm 0.01$ | $0.31 \pm 0.01$ | $0.32 \pm 0.01$ | $0.31 \pm 0.01$ | $0.32 \pm 0.01$ |
| | S(D)↑ | $0.47 \pm 0.02$ | $0.48 \pm 0.01$ | $0.45 \pm 0.02$ | $0.45 \pm 0.02$ | $0.43 \pm 0.02$ | $0.43 \pm 0.02$ | $0.45 \pm 0.01$ | $0.43 \pm 0.01$ |
| Tree | R(S)↑ | $0.59 \pm 0.61$ | $3.01 \pm 0.00$ | $0.47 \pm 1.11$ | $0.15 \pm 0.38$ | $1.05 \pm 1.02$ | $2.34 \pm 1.19$ | $-0.98 \pm 0.01$ | $-0.51 \pm 0.01$ |
| | R(D)↑ | $0.02 \pm 0.02$ | $0.00 \pm 0.00$ | $0.02 \pm 0.01$ | $0.00 \pm 0.01$ | $0.02 \pm 0.01$ | $0.05 \pm 0.01$ | $-0.03 \pm 0.01$ | $-0.01 \pm 0.01$ |
| | P(S)↓ | $0.09 \pm 0.04$ | $0.09 \pm 0.04$ | $0.03 \pm 0.02$ | $0.03 \pm 0.01$ | $0.05 \pm 0.02$ | $0.05 \pm 0.02$ | $0.22 \pm 0.01$ | $0.19 \pm 0.01$ |
| | P(D)↓ | $-0.57 \pm 0.41$ | $-0.57 \pm 0.41$ | $-0.55 \pm 0.41$ | $-0.46 \pm 0.34$ | $-0.41 \pm 0.24$ | $-0.52 \pm 0.37$ | $-0.50 \pm 0.01$ | $-0.48 \pm 0.01$ |
| | S(S)↓ | $0.32 \pm 0.02$ | $0.27 \pm 0.01$ | $0.31 \pm 0.01$ | $0.32 \pm 0.01$ | $0.32 \pm 0.01$ | $0.31 \pm 0.01$ | $0.31 \pm 0.01$ | $0.32 \pm 0.01$ |
| | S(D)↑ | $0.47 \pm 0.02$ | $0.50 \pm 0.00$ | $0.44 \pm 0.02$ | $0.45 \pm 0.02$ | $0.44 \pm 0.02$ | $0.43 \pm 0.01$ | $0.46 \pm 0.01$ | $0.43 \pm 0.01$ |
| Cat Fat | R(S)↑ | $-0.27 \pm 0.49$ | $-0.60 \pm 0.70$ | $0.43 \pm 0.39$ | $0.00 \pm 0.00$ | $0.35 \pm 0.50$ | $0.21 \pm 0.34$ | $1.76 \pm 0.01$ | $0.15 \pm 0.01$ |
| | R(D)↑ | $-0.01 \pm 0.01$ | $-0.00 \pm 0.01$ | $0.01 \pm 0.01$ | $0.00 \pm 0.00$ | $0.01 \pm 0.01$ | $0.00 \pm 0.01$ | $0.01 \pm 0.01$ | $-0.02 \pm 0.01$ |
| | P(S)↓ | $0.06 \pm 0.03$ | $0.09 \pm 0.02$ | $0.03 \pm 0.01$ | $0.03 \pm 0.01$ | $0.02 \pm 0.01$ | $0.02 \pm 0.01$ | $0.11 \pm 0.01$ | $0.09 \pm 0.01$ |
| | P(D)↓ | $-0.70 \pm 0.26$ | $-0.87 \pm 0.36$ | $-0.60 \pm 0.24$ | $-0.60 \pm 0.24$ | $-0.45 \pm 0.18$ | $-0.46 \pm 0.17$ | $-0.80 \pm 0.01$ | $-0.71 \pm 0.01$ |
| | S(S)↓ | $0.30 \pm 0.02$ | $0.30 \pm 0.00$ | $0.30 \pm 0.01$ | $0.25 \pm 0.02$ | $0.30 \pm 0.01$ | $0.31 \pm 0.01$ | $0.30 \pm 0.01$ | $0.30 \pm 0.01$ |
| | S(D)↑ | $0.47 \pm 0.02$ | $0.47 \pm 0.02$ | $0.45 \pm 0.02$ | $0.50 \pm 0.00$ | $0.43 \pm 0.02$ | $0.45 \pm 0.01$ | $0.47 \pm 0.01$ | $0.47 \pm 0.01$ |
| Silhouette | R(S)↑ | $0.19 \pm 0.27$ | $0.18 \pm 0.23$ | $2.61 \pm 0.98$ | $2.46 \pm 1.05$ | $1.19 \pm 0.92$ | $2.85 \pm 0.94$ | $-0.04 \pm 0.01$ | $-0.02 \pm 0.01$ |
| | R(D)↑ | $0.01 \pm 0.01$ | $0.00 \pm 0.01$ | $0.04 \pm 0.01$ | $0.04 \pm 0.02$ | $0.02 \pm 0.02$ | $0.04 \pm 0.02$ | $-0.01 \pm 0.01$ | $-0.01 \pm 0.01$ |
| | P(S)↓ | $0.04 \pm 0.02$ | $0.05 \pm 0.01$ | $0.02 \pm 0.00$ | $0.03 \pm 0.00$ | $0.02 \pm 0.00$ | $0.02 \pm 0.01$ | $0.16 \pm 0.01$ | $0.17 \pm 0.01$ |
| | P(D)↓ | $-0.98 \pm 0.86$ | $-1.14 \pm 1.06$ | $-0.91 \pm 0.62$ | $-0.99 \pm 0.66$ | $-0.55 \pm 0.30$ | $-0.65 \pm 0.37$ | $-0.49 \pm 0.01$ | $-0.51 \pm 0.01$ |
| | S(S)↓ | $0.30 \pm 0.01$ | $0.30 \pm 0.01$ | $0.29 \pm 0.01$ | $0.29 \pm 0.01$ | $0.30 \pm 0.01$ | $0.29 \pm 0.01$ | $0.30 \pm 0.01$ | $0.31 \pm 0.01$ |
| | S(D)↑ | $0.45 \pm 0.02$ | $0.46 \pm 0.02$ | $0.42 \pm 0.02$ | $0.44 \pm 0.01$ | $0.44 \pm 0.02$ | $0.43 \pm 0.01$ | $0.44 \pm 0.01$ | $0.45 \pm 0.01$ |
| Lighthouse Water | R(S)↑ | $0.28 \pm 0.56$ | $0.01 \pm 0.33$ | $0.85 \pm 0.40$ | $1.73 \pm 0.68$ | $0.68 \pm 0.29$ | $1.62 \pm 0.59$ | $0.04 \pm 0.01$ | $-0.04 \pm 0.01$ |
| | R(D)↑ | $0.01 \pm 0.01$ | $-0.00 \pm 0.01$ | $0.01 \pm 0.00$ | $0.02 \pm 0.01$ | $0.01 \pm 0.01$ | $0.02 \pm 0.01$ | $0.00 \pm 0.01$ | $0.02 \pm 0.01$ |
| | P(S)↓ | $0.08 \pm 0.03$ | $0.06 \pm 0.02$ | $0.03 \pm 0.01$ | $0.04 \pm 0.01$ | $0.02 \pm 0.01$ | $0.03 \pm 0.01$ | $0.03 \pm 0.01$ | $0.02 \pm 0.01$ |
| | P(D)↓ | $-1.50 \pm 0.53$ | $-1.30 \pm 0.48$ | $-0.99 \pm 0.46$ | $-1.11 \pm 0.51$ | $-0.75 \pm 0.31$ | $-0.98 \pm 0.39$ | $-0.43 \pm 0.01$ | $-0.40 \pm 0.01$ |
| | S(S)↓ | $0.32 \pm 0.01$ | $0.30 \pm 0.01$ | $0.29 \pm 0.00$ | $0.30 \pm 0.01$ | $0.30 \pm 0.01$ | $0.30 \pm 0.01$ | $0.31 \pm 0.01$ | $0.32 \pm 0.01$ |
| | S(D)↑ | $0.47 \pm 0.02$ | $0.46 \pm 0.02$ | $0.44 \pm 0.02$ | $0.44 \pm 0.01$ | $0.44 \pm 0.02$ | $0.43 \pm 0.02$ | $0.43 \pm 0.01$ | $0.50 \pm 0.01$ |
| Overall Avg | R(S)↑ | $1.17 \pm 0.58$ | $2.29 \pm 0.45$ | $1.38 \pm 0.61$ | $1.30 \pm 0.56$ | $1.06 \pm 0.57$ | $1.52 \pm 0.60$ | $0.04 \pm 0.01$ | $-0.17 \pm 0.01$ |
| | R(D)↑ | $0.02 \pm 0.01$ | $0.02 \pm 0.01$ | $0.02 \pm 0.01$ | $0.02 \pm 0.01$ | $0.02 \pm 0.01$ | $0.03 \pm 0.01$ | $-0.01 \pm 0.01$ | $-0.01 \pm 0.01$ |
| | P(S)↓ | $0.08 \pm 0.02$ | $0.08 \pm 0.02$ | $0.04 \pm 0.01$ | $0.04 \pm 0.01$ | $0.03 \pm 0.01$ | $0.03 \pm 0.01$ | $0.13 \pm 0.01$ | $0.11 \pm 0.01$ |
| | P(D)↓ | $-1.50 \pm 0.93$ | $-1.55 \pm 0.94$ | $-1.18 \pm 0.68$ | $-1.19 \pm 0.66$ | $-0.91 \pm 0.50$ | $-0.99 \pm 0.54$ | $-0.93 \pm 0.01$ | $-0.93 \pm 0.01$ |
| | S(S)↓ | $0.32 \pm 0.01$ | $0.31 \pm 0.01$ | $0.31 \pm 0.01$ | $0.31 \pm 0.01$ | $0.32 \pm 0.01$ | $0.32 \pm 0.01$ | $0.32 \pm 0.01$ | $0.33 \pm 0.01$ |
| | S(D)↑ | $0.45 \pm 0.02$ | $0.46 \pm 0.02$ | $0.44 \pm 0.02$ | $0.44 \pm 0.01$ | $0.44 \pm 0.02$ | $0.44 \pm 0.01$ | $0.44 \pm 0.01$ | $0.46 \pm 0.01$ |

Table 7: Overall results for audio evaluation. R=Conceptual Range, S=Smoothness, P=Preservation.

| | CS | | Ours | | TE | | P2P | |
|---|---|---|---|---|---|---|---|---|
| | w/o ASTD | w/ ASTD | w/o ASTD | w/ ASTD | w/o ASTD | w/ ASTD | w/o ASTD | w/ ASTD |
| **Concept: Barking** | | | | | | | | |
| R↑ | $1.19 \pm 1.72$ | $1.21 \pm 0.96$ | $0.52 \pm 1.24$ | $2.03 \pm 1.92$ | $4.52 \pm 1.87$ | $4.52 \pm 1.87$ | $6.88 \pm 1.99$ | $8.61 \pm 3.07$ |
| S↓ | $0.34 \pm 0.03$ | $0.34 \pm 0.02$ | $0.34 \pm 0.03$ | $0.35 \pm 0.03$ | $0.34 \pm 0.02$ | $0.34 \pm 0.02$ | $0.36 \pm 0.03$ | $0.36 \pm 0.03$ |
| P↓ | $0.89 \pm 0.45$ | $0.72 \pm 0.33$ | $0.76 \pm 0.45$ | $1.04 \pm 0.32$ | $2.72 \pm 0.46$ | $2.72 \pm 0.46$ | $3.39 \pm 0.82$ | $3.50 \pm 0.73$ |
| **Concept: Car** | | | | | | | | |
| R↑ | $-6.48 \pm 4.61$ | $-2.32 \pm 1.35$ | $4.76 \pm 4.46$ | $7.06 \pm 4.88$ | $11.66 \pm 1.92$ | $3.27 \pm 2.12$ | $3.89 \pm 2.86$ | $1.31 \pm 3.22$ |
| S↓ | $0.42 \pm 0.02$ | $0.35 \pm 0.01$ | $0.38 \pm 0.04$ | $0.36 \pm 0.04$ | $0.39 \pm 0.02$ | $0.40 \pm 0.02$ | $0.39 \pm 0.03$ | $0.39 \pm 0.02$ |
| P↓ | $3.39 \pm 0.79$ | $5.55 \pm 0.35$ | $1.76 \pm 0.93$ | $2.19 \pm 0.72$ | $5.49 \pm 0.23$ | $3.40 \pm 0.41$ | $3.34 \pm 1.00$ | $3.94 \pm 0.75$ |
| **Concept: Cat** | | | | | | | | |
| R↑ | $1.99 \pm 1.85$ | $6.66 \pm 1.83$ | $1.16 \pm 1.38$ | $5.59 \pm 2.73$ | $4.35 \pm 4.30$ | $12.30 \pm 3.58$ | $1.63 \pm 1.45$ | $1.63 \pm 1.45$ |
| S↓ | $0.34 \pm 0.02$ | $0.34 \pm 0.02$ | $0.34 \pm 0.02$ | $0.34 \pm 0.04$ | $0.41 \pm 0.02$ | $0.44 \pm 0.03$ | $0.36 \pm 0.03$ | $0.36 \pm 0.03$ |
| P↓ | $1.27 \pm 0.44$ | $2.75 \pm 0.43$ | $0.94 \pm 0.59$ | $2.44 \pm 0.69$ | $4.51 \pm 0.30$ | $5.64 \pm 0.42$ | $3.34 \pm 0.36$ | $3.34 \pm 0.36$ |
| **Concept: Choir** | | | | | | | | |
| R↑ | $1.48 \pm 0.76$ | $6.91 \pm 0.40$ | $0.55 \pm 0.56$ | $2.84 \pm 0.82$ | $1.05 \pm 0.48$ | $2.72 \pm 0.53$ | $0.47 \pm 0.57$ | $0.26 \pm 0.80$ |
| S↓ | $0.38 \pm 0.03$ | $0.40 \pm 0.01$ | $0.38 \pm 0.02$ | $0.38 \pm 0.02$ | $0.37 \pm 0.02$ | $0.45 \pm 0.01$ | $0.41 \pm 0.03$ | $0.41 \pm 0.02$ |
| P↓ | $1.65 \pm 0.25$ | $3.61 \pm 0.38$ | $0.95 \pm 0.33$ | $2.45 \pm 0.32$ | $1.86 \pm 0.33$ | $5.44 \pm 0.15$ | $3.33 \pm 0.76$ | $3.39 \pm 0.72$ |
| **Concept: Clapping** | | | | | | | | |
| R↑ | $0.20 \pm 0.27$ | $-0.05 \pm 0.66$ | $0.33 \pm 0.31$ | $1.42 \pm 0.57$ | $0.33 \pm 0.34$ | $0.45 \pm 0.84$ | $0.24 \pm 0.50$ | $-0.65 \pm 0.47$ |
| S↓ | $0.36 \pm 0.02$ | $0.42 \pm 0.01$ | $0.35 \pm 0.02$ | $0.33 \pm 0.02$ | $0.46 \pm 0.01$ | $0.42 \pm 0.02$ | $0.37 \pm 0.03$ | $0.45 \pm 0.02$ |
| P↓ | $0.66 \pm 0.34$ | $2.52 \pm 0.21$ | $0.41 \pm 0.19$ | $1.89 \pm 0.31$ | $4.99 \pm 0.21$ | $4.06 \pm 0.22$ | $2.51 \pm 0.76$ | $4.54 \pm 0.45$ |
| **Concept: Crowd** | | | | | | | | |
| R↑ | $14.27 \pm 1.27$ | $13.74 \pm 1.39$ | $4.37 \pm 1.38$ | $9.34 \pm 2.20$ | $-1.58 \pm 0.77$ | $-1.61 \pm 0.77$ | $4.89 \pm 1.53$ | $12.29 \pm 2.21$ |
| S↓ | $0.38 \pm 0.02$ | $0.37 \pm 0.01$ | $0.36 \pm 0.02$ | $0.36 \pm 0.02$ | $0.47 \pm 0.02$ | $0.46 \pm 0.02$ | $0.48 \pm 0.01$ | $0.41 \pm 0.02$ |
| P↓ | $2.31 \pm 0.23$ | $2.08 \pm 0.38$ | $0.80 \pm 0.35$ | $1.98 \pm 0.44$ | $5.53 \pm 0.13$ | $5.53 \pm 0.13$ | $4.28 \pm 0.40$ | $4.58 \pm 0.30$ |
| **Concept: Door** | | | | | | | | |
| R↑ | $-0.50 \pm 1.20$ | $-2.49 \pm 2.59$ | $3.06 \pm 2.30$ | $2.89 \pm 1.97$ | $7.75 \pm 2.39$ | $11.93 \pm 2.70$ | $12.05 \pm 4.10$ | $11.46 \pm 3.55$ |
| S↓ | $0.39 \pm 0.02$ | $0.33 \pm 0.02$ | $0.37 \pm 0.02$ | $0.36 \pm 0.02$ | $0.34 \pm 0.02$ | $0.36 \pm 0.01$ | $0.35 \pm 0.03$ | $0.34 \pm 0.03$ |
| P↓ | $0.21 \pm 0.22$ | $0.92 \pm 0.39$ | $0.70 \pm 0.33$ | $0.61 \pm 0.32$ | $2.24 \pm 0.31$ | $3.25 \pm 0.35$ | $3.73 \pm 0.34$ | $3.69 \pm 0.33$ |
| **Concept: Guitar** | | | | | | | | |
| R↑ | $1.23 \pm 1.00$ | $3.02 \pm 2.16$ | $2.14 \pm 1.63$ | $8.84 \pm 1.63$ | $10.06 \pm 0.95$ | $11.49 \pm 1.30$ | $4.49 \pm 2.36$ | $4.49 \pm 2.36$ |
| S↓ | $0.35 \pm 0.02$ | $0.37 \pm 0.02$ | $0.34 \pm 0.03$ | $0.34 \pm 0.01$ | $0.34 \pm 0.00$ | $0.33 \pm 0.01$ | $0.36 \pm 0.04$ | $0.36 \pm 0.04$ |
| P↓ | $1.11 \pm 0.37$ | $3.28 \pm 0.32$ | $1.86 \pm 0.32$ | $4.62 \pm 0.35$ | $5.72 \pm 0.27$ | $5.06 \pm 0.42$ | $5.35 \pm 0.96$ | $5.35 \pm 0.96$ |
| **Concept: Ocean** | | | | | | | | |
| R↑ | $5.68 \pm 1.06$ | $13.96 \pm 1.41$ | $1.29 \pm 0.88$ | $4.48 \pm 1.90$ | $-6.12 \pm 1.74$ | $6.81 \pm 1.30$ | $11.69 \pm 2.79$ | $15.14 \pm 1.66$ |
| S↓ | $0.32 \pm 0.02$ | $0.31 \pm 0.01$ | $0.32 \pm 0.01$ | $0.32 \pm 0.02$ | $0.45 \pm 0.02$ | $0.35 \pm 0.01$ | $0.32 \pm 0.03$ | $0.35 \pm 0.02$ |
| P↓ | $1.31 \pm 0.19$ | $2.15 \pm 0.14$ | $0.24 \pm 0.13$ | $1.12 \pm 0.11$ | $4.68 \pm 0.12$ | $2.74 \pm 0.11$ | $4.31 \pm 1.04$ | $5.10 \pm 0.75$ |
| **Concept: Rain** | | | | | | | | |
| R↑ | $10.46 \pm 1.94$ | $17.19 \pm 1.46$ | $4.88 \pm 1.83$ | $15.93 \pm 1.75$ | $10.06 \pm 3.10$ | $9.39 \pm 2.23$ | $9.32 \pm 4.24$ | $13.01 \pm 3.53$ |
| S↓ | $0.26 \pm 0.02$ | $0.29 \pm 0.02$ | $0.28 \pm 0.01$ | $0.31 \pm 0.02$ | $0.33 \pm 0.03$ | $0.36 \pm 0.03$ | $0.37 \pm 0.06$ | $0.41 \pm 0.04$ |
| P↓ | $2.21 \pm 0.41$ | $3.72 \pm 0.24$ | $1.17 \pm 0.72$ | $2.98 \pm 0.67$ | $3.43 \pm 0.54$ | $4.26 \pm 0.49$ | $4.07 \pm 0.71$ | $4.42 \pm 0.57$ |
| **Overall Score (Averaged across concepts)** | | | | | | | | |
| R↑ | $2.95 \pm 1.57$ | $5.78 \pm 1.42$ | $2.31 \pm 1.60$ | $6.04 \pm 2.04$ | $4.21 \pm 1.79$ | $6.13 \pm 1.72$ | $5.56 \pm 2.24$ | $6.75 \pm 2.23$ |
| S↓ | $0.35 \pm 0.02$ | $0.35 \pm 0.02$ | $0.35 \pm 0.02$ | $0.34 \pm 0.02$ | $0.39 \pm 0.02$ | $0.39 \pm 0.02$ | $0.38 \pm 0.03$ | $0.39 \pm 0.03$ |
| P↓ | $1.50 \pm 0.37$ | $2.73 \pm 0.32$ | $0.96 \pm 0.43$ | $2.13 \pm 0.42$ | $4.12 \pm 0.29$ | $4.21 \pm 0.32$ | $3.76 \pm 0.71$ | $4.19 \pm 0.59$ |

