# OpenReview forum: "Training-Free Modality-Agnostic Concept Sliders: Fine-Grained Control via Diffusion Models of Images, Audio, and Video"
_ICLR.cc/2026/Conference — Submitted to ICLR 2026_

### Official Review · Reviewer_871i · 2025-10-30

**Soundness:** 1
**Presentation:** 2
**Contribution:** 2
**Rating:** 2
**Confidence:** 4

**Summary:**

The user propose a simple training-free and modality-agnostic method for concept slider task. Also, they provide three new metrics for concept slider (CS) for evaluating the range, smoothness and preservation of each method. Besides their inference time method, they also propose a method ASTD, which be used on top of other methods like CS, improve the performance.

**Strengths:**

- Proposed inference-time method is simple, easy to follow, and achieves better scores than CS on the benchmarks proposed by the authors.
- The evaluation principle (range, smoothness, preservation) for the CS task are reasonable, and using this idea to build a benchmark make sense.
- Compositional concept slider shown by authors is interesting.

**Weaknesses:**

- The author failed to adequately explore the proposed method in the paper. The authors simply present their method without any explanation or motivation. A simple and effective method is good, but could the authors provide some motivation and ablation for doing so and explain the underlying idea of the method?
- The method seems trivial. It apply $\epsilon_{\theta}(x_t, c_{base}, t)$ at first k steps, and $\epsilon_{\theta}(x_t, c_{base}, c_{+}, c_{-}, t)$ at the remaining steps, which all operators are presented in CS except It does not require fine-tuning and use $c_{base}$ for inference at the first K steps. As mentioned above, without proper analysis and motivation, I cannot be convinced that this is better than a method that requires fine-tuning (besides computational cost).Can the author explain why fine-tuning in CS is inefficient, even less so than a inference-time method?
- As models like clip/vclip often fails to assess fine-grained details about image/video, and sensitive to prompts. Could authors provide some user study about their comparison with other baselines? Also, the alignment between proposed metrics and human's preference should be assessed.
- Some qualitative cases are not good applications for concept slider, like adding a eyeglasses for a people, it's somehow a yes or no problem, rather than a concept of variability. Could authors provide some explaination about the potential applications of these cases or replace them?

**Questions:**

- How do author choose K? Is this variable depending on the concept/model architecture/specific model, or is it a constant value?
- Might be better to adjust the paper organization, move section 5 under method part would be better. Since it's placed after the experimental indicators section, it makes the paper seem somewhat disjointed.

---

> ### Author Response · Authors · 2025-11-23
>
> We thank the reviewer for the constructive feedback and for recognizing the effectiveness of our approach, the relevance of our benchmarking and metrics, and the contribution of compositional sliders. Below, we address the concerns raised, which helped us improve significantly our work.
>
> ---
>
> > W1 + W2 - The authors simply present their method without any explanation or motivation. Could the authors provide some motivation and ablation for doing so and explain the underlying idea of the method?
> > The method seems trivial... . As mentioned above, without proper analysis and motivation, I cannot be convinced that this is better than a method that requires fine-tuning (besides computational cost).Can the author explain why fine-tuning in CS is inefficient, even less so than a inference-time method?
>
> Thank you for these comments. We are somewhat unsure how to best address them, as they can be interpreted in multiple ways. Below we clarify our current intentions; We would be very grateful if the reviewer could refer to our questions below and helping us engage.
>
> > The authors simply present their method without any explanation or motivation. Could the authors provide some motivation and ablation for doing so and explain the underlying idea of the method?
>
> We lay the general motivation and gaps in the current literature in Section 1, motivating the development of a training-free, modality-agnostic method. In Section 3.1, we provide background and discuss the limitations of closely related methods that are *not* training-free or modality-agnostic. Then, in Section 3.2, we present our method and, at the end of that section, explicitly emphasize its benefits. Can the reviewer clarify what kind or an example for missing motivation? We would be happy to clarify any point.
> Regarding ablations, in the experimental section we compare against a text-embedding variant (Text Embedding CS), provide computational comparisons, and present results across multiple video backbones. Could the reviewer clarify which specific ablations they feel are missing?
>
> > The method seems trivial, without proper analysis and motivation, I cannot be convinced that this is better than a method that requires fine-tuning (besides computational cost).
>
> We respectfully disagree with the statement that our method is not convincingly better than Concept Sliders (which is very similar to our method just with training). While simple, our method is effective and robust, as evidenced by an extensive experimental section spanning 3 modalities, 30 concepts, multiple backbones, and several baselines. It consistently outperforms all other baselines despite being training-free, providing strong motivation to use our approach beyond its computational advantages.
>
> Regarding “triviality,” we are a bit confused, since the review also lists the simplicity of our method as a strength - a point with which we agree, as this simplicity is precisely what makes the method adaptable to different modalities, architectures, and samplers. Could the reviewer clarify this apparent tension?
>
> > Can the author explain why fine-tuning in CS is inefficient, even less so than an inference-time method?
>
> Thank you for asking; we are happy to clarify this point. Beyond the performance gains discussed above, our parameter-free, inference-only approach offers several advantages over fine-tuning-based methods such as LoRA CS (in addition to computational cost):
>
> 1. **Data efficiency.**
> Fine-tuning requires curating or generating a dataset for every new concept, which introduces pipeline complexity (data preparation, filtering, storage). Our method requires **no per-concept data collection**.
>
> 2. **Scalability and production utility.**
> In a production setting, LoRA-based CS requires training, storing, and managing a separate adapter for each concept, effectively fixing the system to a finite “concept menu.” In contrast, our training-free method supports **open-vocabulary** control: users can define new concepts on the fly via text prompts, without retraining or managing additional checkpoints.
>
> 3. **Architecture and sampler agnosticism.**
> LoRA implementations and their training procedures are often tightly coupled to specific architectures and sampling schemes. Our inference-time intervention operates at the score/function level and is therefore naturally compatible with different backbones and samplers, as demonstrated empirically.

---

> > ### Author Response · Authors · 2025-11-23
> >
> > > W3 - As models like clip often fails to assess fine-grained details about image/video and sensitive to prompts. Could authors provide some user study about their comparison with other baselines? Also, the alignment between proposed metrics and human's preference should be assessed.
> >
> > Thank you very much for raising these important suggestions. Following your comment, we conducted two **new** experiments:
> > (1) **Human Preference Evaluation**, where we ask subjects which slider output they prefer, and
> > (2) **Metric Alignment with Human Perception**, where we study how our metrics correlate with human judgments of range and smoothness.
> > We have added these experiments to the revised version and briefly summarize them below.
> >
> > **Human Preference Evaluation.**
> > In this experiment, each subject was shown two sliders side-by-side and was asked which slider they preferred. One generated by our method and one by Concept Sliders (CS), without knowing which was which. We collected a total of 336 pairwise preferences across 12 subjects and 6 different concepts on the image dataset. Overall, **54%** of the preferences (182/336) favored our method, while **46%** (154/336) favored CS. This **9% gap** is consistent with our quantitative metric results and supports the conclusion that, despite being fully training-free, our approach can outperform a method that relies on per-concept training, underscoring both its effectiveness and practical relevance.
> >
> > **Metric Alignment with Human Perception.**
> > In this experiment, we evaluate the two new metrics in our benchmark. Subjects were shown pairs of sliders (collected from multiple methods) and asked, for each pair: (1) which slider has better **range**, and (2) which has better **smoothness**. We collected 170 responses over 10 concepts from 17 subjects. We then counted how often the human choice matched the metric’s preferred slider (taking the maximum-scoring option for example, if the metric scores option (a) as 0.4 and option (b) as 0.6, we treat (b) as the metric’s choice). For range, there is **85%** agreement, and for smoothness **70%** agreement, providing additional evidence that our metrics meaningfully reflect human perceptual preferences.
> >
> > ---
> >
> > > W4 - Some qualitative cases are not good applications for concept slider, like adding a eyeglasses for a people, it's somehow a yes or no problem, rather than a concept of variability. Could authors provide some explaination about the potential applications of these cases or replace them?
> >
> > Thank you for raising this point and helping us improve our work. We agree with your remark. We initially used this concept because it was demonstrated in previous works; however, as you suggest, we will remove these examples and replace them with continuous concepts in the main paper. Thank you for pointing this out.

---

> > > ### Author Response · Authors · 2025-11-23
> > >
> > > > Q1 - How do author choose K? Is this variable depending on the concept/model architecture/specific model, or is it a constant value?
> > >
> > >
> > > That is an excellent question. For all models and concepts, we chose a **single default (k)** to maintain fairness and consistency. Across modalities, we selected values proportional to the total number of inference steps of the backbone; heuristically, we found that (k $\approx$ 30%) of the full trajectory offers a strong balance between preservation and expressive range across models.
> > >
> > > Additionally, in our experiments, we observed some heuristics relating the choice of (k) (the number of timesteps where the concept slider is active) to the trade-off between semantic preservation, conceptual range, and smoothness. When **reducing** the size of (k), semantic preservation degregate while conceptual range and smoothness improves. Intuitively, applying the concept direction only for a *small* portion of the trajectory gives the model limited opportunity to “anchor’’ the sample to the original semantic identity. At the same time, concept shift is being amplified, probabilty since its has more steps to effect the trajectory. In contrast, when **increasing** the size of (k), we observe the opposite behavior: preservation improves while range and smoothness degrade. This aligns with the interpretation that applying the concept direction over *many* steps repeatedly pulls the sample back toward the base manifold, suppressing large conceptual deviations but retaining more of the original semantics.
> > >
> > >
> > > We believe that exploring an *adaptive* or concept-dependent choice of (k), is a promising direction for future work. Following this question, we have added the above discussion to the main paper.
> > >
> > > ---
> > >
> > > > Q2 - Might be better to adjust the paper organization, move section 5 under method part would be better. Since it's placed after the experimental indicators section, it makes the paper seem somewhat disjointed.
> > >
> > > Thank you for this comment. Switching Section 5 with Section 4 is indeed a reasonable alternative, and we considered it while writing the paper. However, we chose the current order for two reasons: (1) ASTD (Section 5) is an independent component, and presenting it as a separate, later module reinforces our message that it can be used as a standalone add-on; and (2) the analysis and formulation of ASTD fundamentally rely on the formal definitions introduced in Section 4, which specify the aspects we aim to optimize - moving Section 5 earlier would therefore make many details appear out of context. We appreciate the suggestion and the opportunity to clarify this choice.

---

> > > > ### Author Response · Authors · 2025-11-27
> > > >
> > > > Dear 871i and reviewers,
> > > > Thank you for your suggestion for a user study in W3, to address this we have conducted an additional experiment consisting of a user study to compare an additional baseline.
> > > >
> > > > >  W3 - As models like clip/vclip often fails to assess fine-grained details about image/video, and sensitive to prompts. Could authors provide some user study about their comparison with other baselines? Also, the alignment between proposed metrics and human's preference should be assessed.
> > > >
> > > > The user study we conducted tests 30 sliders per concept, consisting of 7 scales each (overall 210 images) per concept. We replaced CLIP with BLIP as the alignment metric and computed their correlation. We found that there is an overall positive correlation between CLIP and BLIP scores. Approximately $r ≈ 0.65, p ≈ 0.000$, where $r$ represents the Pearson correlation coefficient (how much of the variance in one variable is explained by the other) and $p$ represents the statistical significance (the probability that the observed correlation occurred by chance). This suggests that despite the known biases, the overall trends captured by CLIP remain stable across different aligners. We attached a visual representation of this study in the supplementary material (correlation_blip_clip.pdf) as well as Section C.7 in the Appendix in the latest paper revision.
> > > > Regarding the alignment with human preference: this has been addressed in our previous response. Thank you for your constructive feedback, we appreciate your careful review and insightful suggestions.

---

### Official Review · Reviewer_EsD1 · 2025-11-01

**Soundness:** 3
**Presentation:** 3
**Contribution:** 3
**Rating:** 6
**Confidence:** 4

**Summary:**

This paper introduces a training-free, model-agnostic framework for applying Concept Sliders (CS) at inference time. The core idea is to partition the diffusion timesteps, using the base model for initial semantic generation and then applying a Classifier-Free Guidance (CFG)-like mechanism (positive-negative prediction difference) for the later, concept-forming steps.
A significant contribution of this work is the proposal of three new metrics to evaluate CS quality, addressing the limitations of existing methods: Conceptual Range (CR), Semantic Preservation (SP), and Conceptual Smoothness (CSM). Building on these metrics, the authors also identify the non-linear nature of concept application and propose a two-stage Automatic Saturation and Traversal Detection (ASTD) method, which reportedly optimizes quality by investing additional computation at inference time.

**Strengths:**

- Simple and Intuitive Methodology: The core method for applying CS at inference time is simple and well-motivated. Adapting the CFG mechanism by partitioning timesteps (base-only vs. base+CS) is an elegant way to achieve training-free application.

- Novel and Valuable Metrics: The proposal of the SP, CR, and CSM metrics is a strong contribution to the field. These metrics address clear limitations of existing evaluations (like $\Delta$ CLIP) by attempting to quantify crucial quality aspects—such as identity preservation, range of concept expression, and smooth transitions—which are vital for usability but often overlooked.

- Insightful Analysis (Non-linearity and ASTD): The paper insightfully identifies and demonstrates the non-linear behavior of concept sliders. Leveraging this observation to create the two-stage ASTD method is a clever extension that shows a deep understanding of the problem.

- Comprehensive Experiments: The method's effectiveness is demonstrated across a comprehensive set of experiments, including image, video, and audio generation, suggesting broad applicability of the core ideas.

**Weaknesses:**

While the contributions are notable, the paper has several weaknesses, primarily concerning practical trade-offs and the scope of validation, which position it as a borderline paper.

1. Significant Inference Cost & Questionable Trade-off: The "training-free" claim comes at a very high price. The method requires at least three model evaluations (base, positive, negative) per guided step, introducing significant inference latency and memory overhead. This trade-off is not sufficiently discussed. More critically, the proposed ASTD method is noted to require even more computation than training a CS from scratch. This makes the practical utility of ASTD highly questionable. The paper needs a much stronger justification for why a user would accept this massive inference cost over a one-time training cost.

2. Limited Conceptual Scope: The experiments are constrained to approximately ten concepts (e.g., 'age', 'smile'), which are well-established, common, and primarily relate to direct object manipulation or clear states. The paper lacks experiments on more abstract, nuanced, or rare concepts, which are often the main targets for creative exploration with sliders. This limits the demonstrated generalizability of the proposed metrics and methods.

3. Limited Model-Agnosticism Claim: The experiments appear heavily focused on noise-prediction and v-prediction models. It is unclear if the method directly applies to other diffusion frameworks, such as the increasingly prevalent flow-matching models. This omission weakens the "model-agnostic" claim.

4. Minor Issues:
  - Clarity on Multi-Concept Sliders: The paper demonstrates two-concept sliders, but the mathematical formulation and operational details (e.g., sequential vs. parallel application, potential for order-dependency) are not explained.
  - Presentation Inconsistencies: There are minor formatting issues, such as inconsistent references (e.g., 'Fig.' vs. 'figures' in Appendix C), missing experimental details (e.g., the exact sampler used), and unclear distinctions between which experiments used SD 1.4 vs. 1.5 ("some").

**Questions:**

I would appreciate it if the authors could clarify the following points in their rebuttal.

### Questions
1. Metric Formulation: Regarding the 'Overall Score' metric (CR / ($\epsilon$ + SP) + (1 - CSM)): The choice of $\epsilon$ = 1 seems disproportionately large compared to SP (which is < 1), potentially distorting the metric by heavily underweighting the Semantic Preservation term. Could the authors provide the rationale for this specific formulation and the value of epsilon? What intuition (e.g., related to saturation) guides this choice?

2. Applicability to Flow Matching: Could the authors comment on the method's applicability to flow-matching models? Specifically, since SD 3 is mentioned (which is a rectified flow model), how was the CFG-like guidance equation (predict_noise(...) - predict_noise(...)) adapted for a framework that predicts velocity?

3. Rationale for ASTD Cost: Given that ASTD is reportedly slower than training a new CS (and much slower than simple inference), what is the practical rationale for its use? In what specific scenario is this extreme trade-off (shifting a one-time training cost to every inference) justified?

4. Multi-Concept Slider Implementation: When applying two sliders simultaneously, how are the positive and negative pairs defined and combined? Is the process order-dependent (e.g., does applying 'age' then 'smile' differ from 'smile' then 'age')? How are potential interferences between sliders managed?

### LLM Disclosure
I have used an LLM to assist with improving the grammar, clarity, and polishing of this review. The content, analysis, and final judgments are entirely my own.

---

> ### Author Response · Authors · 2025-11-23
>
> We sincerely thank the reviewer for the thoughtful and constructive feedback. Your comments helped us refine several aspects of the paper and improve its clarity and significance. We appreciate your recognition of our simple, training-free CS methodology, the proposed SP/CR/CSM metrics, the non-linearity analysis leading to ASTD, and the comprehensive multimodal experiments. Below, we address all concerns and questions raised, and we would be happy to consider any further suggestions/questions you may have.
>
> ---
>
> > W1 - The computational tradeoff is high and not sufficiently discussed. The proposed ASTD method is noted to require even more computation than training a CS from scratch. This makes the practical utility of ASTD highly questionable. Needs a much stronger justification for why a user would accept this massive inference cost over a one-time training cost.
>
> Thank you for raising this important point and allowing us to emphasize aspects we somewhat failed to communicate. We would like to clarify two main points.
>
> **First.** **ASTD is optional** and operates **on top of** sliding methods. As shown in Table 3 of the main paper, both our method and CS are independent of ASTD and work without it. However, when ASTD is applied, **both** methods improve by a significant margin. This underscores the importance of selecting an appropriate range and scaling for an effective slider. Importantly, our method achieves superior results compared to CS in both with and without the ASTD in these evaluations.
>
> **Second.** CS requires **one-time training per concept**. In practice, this means that for every new concept, or even mild concept variations, a new CS must be trained from scratch. In contrast, our method (without ASTD) **requires no training** and can be run directly at inference time, which is often more practical for online, diverse creator workflows. We also emphasize that ASTD itself is a **one-time calibration per concept**, but it remains **optional** and can be applied to any sliding method that uses scaling, including ours and CS.
>
> Finally, to further improve ASTD’s usability, we develop a simple yet effective strategy that halves its runtime. Instead of sampling all diffusion steps, we sample only half , since high-fidelity final samples are not highly essential to estimate the range and scales that the ASTD should detect. We refer to this variant as **ASTD2x**. To validate this strategy, we apply ASTD and ASTD2x once each to improve our model on the image benchmark. The results are as follows: for **Semantic Preservation**, both achieve 0.02; for **Conceptual Range**, ASTD scores 0.299 while ASTD2x is only slightly lower at 0.285; and for **Conceptual Smoothness**, both score 0.28. Overall, this shows that ASTD can be accelerated by a factor of 2 with essentially no loss in quality.
>
> Following this discussion, and to improve the paper’s quality and clarity, we now highlight the practical utility of our inference-only method, ASTD, and the new ASTD2x as stand-alone components, and we have incorporated the above clarifications into the manuscript. We are happy to provide any further details if needed.
>
> ---
>
> > W2 - Limited Conceptual Scope: The experiments are constrained to approximately ten concepts which are well-established, common, and primarily relate to direct object manipulation or clear states. The paper lacks experiments on more abstract, nuanced, or rare concepts, which are often the main targets for creative exploration with sliders.
>
> Thank you for raising these important points. Following your suggestion, we have added a substantial number of more complex concepts to the final benchmark: 5 new concepts for video and 8 for images. The added concepts include:
>
> * **Video:** landscape flat vs. rolling terrain, river flow intensity, number of birds in a sky scene, meadow variation in a field, and car traffic speed.
> * **Images:** shore wave intensity, mountain snow coverage, forest season, crowd density on a beach, city vs. park background in a portrait, dog fur color, eye color, and hairstyle.
>
> This expansion makes the benchmarks significantly more diverse and challenging for both modalities. Additionally, we provide visual examples of these new concepts in the supplementary material (`additional_concepts.pdf`) for illustration.

---

> > ### Author Response · Authors · 2025-11-23
> >
> > We would also like to emphasize that, while for **video** and **audio** we constructed new benchmarks from scratch, for **images** we used the standard benchmark from Concept Sliders. Even without the new extensions, some attributes are genuinely **complex**. For instance, as shown on our anonymized website ([https://anonymous.4open.science/w/FreeSliders-4CD3/](https://anonymous.4open.science/w/FreeSliders-4CD3/)) and in the paper, our “car condition” slider spans from a brand-new car to a wrecked car, requiring coordinated changes such as color adjustments, geometric deformation, window integrity, and more. Similarly, in audio, we demonstrate controlling the pitch of a choir singing a classical piece and the “storminess” of ocean sounds - both of which go well beyond simple loudness changes.
> >
> > ---
> >
> > > W3 - Limited Model-Agnosticism Claim: The experiments appear heavily focused on noise-prediction and v-prediction models. It is unclear if the method directly applies to other diffusion frameworks, such as the increasingly prevalent flow-matching models. This omission weakens the "model-agnostic" claim.
> >
> > We would like to clarify this important aspect. In our paper, we already evaluate on a flow-matching model: **LTX-Video**, which uses a flow-matching denoising paradigm.
> >
> > To further address your concern, during the rebuttal we added **Stable Diffusion 3 (SD3)** to our benchmarking, which is also flow-matching–based (rectified flow). We attach qualitative results in the new supplementary file *`stable_diffusion_3.pdf`* and report the results below
> >
> >
> > |              | Conceptual Range ↑ | Conceptual Smoothness ↓ | Semantic Preservation ↓ | Overall Score ↑ |
> > | ------------ | ------------------ | ----------------------- | ----------------------- | --------------- |
> > | **SD3**  | 2.13           | 0.27                | 0.018                   | 1.36        |
> >
> >
> > These results further strengthen the experimental evidence that our method is agnostic to the denoising paradigm (and to the model architecture).
> >
> > Thank you for raising this point. Following the above, we added clarifications to the paper regarding the flow-matching nature of LTX-Video and the new experiments with SD3.
> >
> > ---
> >
> > > MW1 - Clarity on Multi-Concept Sliders: The paper demonstrates two-concept sliders, but the mathematical formulation and operational details (e.g., sequential vs. parallel application, potential for order-dependency) are not explained.
> >
> > Thank you for raising this point, which helps us sharpen this aspect. Composing multiple concepts is additive, and the formulation for combining several concepts, outlined in [1],  is:
> >
> > $\nabla_{x_t} \log p(x_t \mid c_{\text{base}}) + \eta \sum_{c \in \mathcal{C}} \big(\nabla_{x_t} \log p(x_t \mid c_+) - \nabla_{x_t} \log p(x_t \mid c_-) \big)$
> >
> > Intuitively, each term in the sum defines a “concept direction’’ (where each element of $\mathcal{C}$ is a base/positive/negative tuple) that pushes the sample toward $c_+$ and away from $c_-$. Thus, composing multiple concepts amounts to adding their directions in score space, making control along each axis potentially independent.
> >
> > To further clarify this, we have added a new experiment in the supplementary material: a matrix of concept combinations, provided in `compositions.pdf`, illustrating pairs of concept slider combinations.
> >
> > [1] Concept Sliders: LoRA Adaptors for Precise Control in Diffusion Models - R Gandikota, J Materzyńska, T Zhou, A Torralba, D Bau
> >
> > ---
> >
> > > MW2 - Presentation Inconsistencies: There are minor formatting issues, such as inconsistent references (e.g., 'Fig.' vs. 'figures' in Appendix C), missing experimental details (e.g., the exact sampler used), and unclear distinctions between which experiments used SD 1.4 vs. 1.5 ("some").
> >
> > Thank you for these comments; we have addressed all of them in the revised version of the paper now uploaded.

---

> > > ### Author Response · Authors · 2025-11-23
> > >
> > > > Q1 - Regarding the 'Overall Score' metric, the choice of epsilon = 1 seems large compared to SP (which is < 1), potentially distorting the metric by heavily underweighting the Semantic Preservation term. Provide the rationale for this specific formulation and the value of epsilon? What intuition guides this choice?
> > >
> > > During the construction of our benchmark, we sought a single overall metric to simplify comparisons and capture the trade-off between range, semantic change, and smoothness. Since range is a “higher is better’’ score and semantic preservation is “lower is better,’’ their ratio is a natural way to quantify this relationship. However, in our experiments we observed that preservation can become very small, approaching zero—for example, when a slider does not change the image at all, it yields an extremely low preservation score. In contrast, range scores, although theoretically able to reach very low values, rarely do so in practice, even when the perceptual range is essentially zero, likely due to the noisy nature of the underlying backbone model. The motivation and intuition for adding $(\varepsilon)$ was precisely to guard against such degenerate cases. In addition, we wanted the magnitude of the $(1 - \text{CSM})$ term to be comparable to the ratio term, so that it would not be numerically dominated or effectively ignored.
> > >
> > > ---
> > >
> > > > Q2 - Applicability to Flow Matching: Could the authors comment on the method's applicability to flow-matching models? Specifically, since SD 3 is mentioned (which is a rectified flow model), how was the CFG-like guidance equation (predict_noise(...) - predict_noise(...)) adapted for a framework that predicts velocity?
> > >
> > > That’s a good question. First, as noted in W3, the main paper already includes experiments with LTX-Video, which is a flow-matching model. Following your suggestion, we have expanded our experiments to also include SD3. For both models, we use the default CFG settings.
> > >
> > > Regarding the noise/velocity calculation, LTX-Video and SD3 follow the same logic: in flow-matching frameworks the model predicts a **velocity field** rather than a noise term. The guidance equation is applied analogously by linearly combining the base, positive and negative velocity predictions:
> > >
> > > $v(x_t) = v_{\text{base}}(x_t) + \eta \big(v_+(x_t) - v_-(x_t)\big).$
> > >
> > > ---
> > >
> > > > Q3 - Rationale for ASTD Cost: Given that ASTD is reportedly slower than training a new CS, what is the practical rationale for its use? In what specific scenario is this extreme trade-off justified?
> > >
> > > Please see our response to W1. We hope we have addressed all of your questions regarding this subject, and we will be happy to provide any further information if needed.
> > >
> > > ---
> > >
> > > > Q4 - When applying slider composition, how are the positive and negative pairs defined and combined? Is the process order-dependent? How are potential interferences between sliders managed?
> > >
> > > Please see our response to MW1 above; we will be happy to provide further details if needed.
> > >
> > > Regarding interference, this is an excellent question. All CS-style methods are subject to the underlying entanglement of the backbone diffusion model. For example, if changing “age” also changes “gender,” or altering “hair color” also shifts “gender,” the corresponding attribute vectors may interfere. Conversely, if the backbone is inherently disentangled, such interference should not occur. In our experiments, we did not observe clear cases of interference, but we agree this is an important avenue for future work. We have added this discussion in the relevant section of the paper to highlight potential issues, and we also mention it in the conclusion as an interesting direction to explore. Importantly, if such concept disentanglement exists in the backbone, our approach should not introduce additional interference. Thank you for raising this point and allowing us to clarify it.

---

> > ### Comment · Reviewer_EsD1 · 2025-11-24
> >
> > I thank the authors for the detailed response and the visual supplementary material. Most of my concerns have been addressed satisfactorily.
> >
> > However, to further strengthen the paper and clarify the theoretical/empirical contribution, I have a few remaining questions and suggestions regarding the new results. These are mostly suggestions for final polish or future insights, rather than demands for a major revision.
> >
> > ### 1. About `ASTD2x` Efficiency and Trajectory
> >
> > The authors demonstrated that ASTD can maintain performance metrics even with half the inference steps. This is an intriguing result.
> >
> > * **Inference Trajectory:** What makes the inference trajectory of ASTD robust even with fewer steps? Does ASTD implicitly help the denoising process follow a more "optimal path" (e.g., closer to the geodesic on the data manifold) toward the target latent?
> >     * *Suggestion:* While not strictly required, providing a visualization of the inference trajectory (ASTD vs. Baseline) or a brief theoretical intuition in the final version would significantly strengthen the paper.
> > * **Caching Strategy:** Since ASTD involves additional computation, have the authors considered caching strategies? For instance, techniques similar to [1] could potentially be adapted for the concept slider components to accelerate the process.
> > * **Distribution of Scales ($\eta$):** You stated that *"the scales found with the ASTD add-on vary per concept and per method."* It would be very insightful to see the actual empirical distribution of the found $\eta$ values. If the optimal $\eta$ ranges differ significantly across models (e.g., $\epsilon$-prediction vs. v-prediction vs. Rectified Flow), it would empirically prove why the fixed baseline `linspace(-3, 3)` fails. I suggest adding this analysis to the Appendix.
> >
> > ### 2. Rare or Complex Concepts (Composition)
> >
> > I appreciate the additional experiments in the supplementary material. I strongly suggest moving these results to the main paper or Appendix in the final version.
> >
> > * **Composition Method:** The current method sums the scores linearly (as shown in the provided equation). However, if the concept directions overlap, simple summation might lead to interference. Have the authors considered **orthogonal projection** or spherical interpolation (**SLERP**) instead of linear combination?
> >     * *Comment:* I am just curious about your intuition on whether ensuring orthogonality would yield better disentanglement in complex scenarios.
> >
> > ### 3. Applicability to Flow Matching (SD3, LTX-Video)
> >
> > I am glad to see the method works well with Flow Matching models. Since the velocity field $v_t$ and noise prediction $\epsilon$ share a linear relationship, the guidance equation holds analogously.
> >
> > * *Suggestion:* Although the authors explained this in the rebuttal, explicitly deriving the connection between the **general score function** formulation and the **velocity prediction** formulation in the Appendix would be very helpful for readers. It clarifies that your method is not limited to noise prediction but applicable to any vector-field-based generative model.
> >   * I do not expect this to be addressed during the tight discussion period; however, including this derivation in the camera-ready version (or future revisions) would significantly improve the paper's completeness.
> >
> > ### 4. Rationale for Overall Score (OS)
> >
> > Regarding the formulation $OS = \frac{CR}{\epsilon + SP} + (1 - CSM)$ with $\epsilon=1$:
> >
> > * **Metric Design (Additive vs. Multiplicative):** I suspect that the relative ranking of the methods remains stable regardless of the aggregation method. However, analytically, the first term $\frac{CR}{\epsilon + SP}$ tends to be significantly larger than 1 (dominating the score).
> > * **Scale Mismatch:** In contrast, $CSM$ acts as a variancemeasure, so the term $(1 - CSM)$ operates on a much smaller scale compared to the first term.
> > * **Dominance Issue:** Because of this scale mismatch, the metric is effectively dominated by the $CR$ term, potentially masking the contribution of $CSM$. A multiplicative formulation (e.g., geometric mean) might have been theoretically more robust in balancing these trade-offs without one term overpowering the others.
> > * *Note:* This is just a comment on metric design for future reference. I do not ask for new experiments since the qualitative conclusions likely hold.
> >
> > Based on the constructive rebuttal, I have maintained my positive rating.
> >
> > ### Reference
> > [1] Wimbauer, Felix, et al. "Cache me if you can: Accelerating diffusion models through block caching." CVPR 2024.

---

> > > ### Author Response · Authors · 2025-11-27
> > >
> > > Thank you very much for continuing to engage in this fruitful and impactful discussion. We greatly appreciate your comments and suggestions, which reflect a deep and broad understanding of our work. Following the previous exchanges—which have already helped us improve the paper significantly—and despite the limited time, we are keen to keep refining it further. Below, we respond to each point in detail.
> > >
> > >
> > > ### ASTD2x
> > > #### Inference Trajectory:
> > >
> > > > What makes the inference trajectory of ASTD robust even with fewer steps?
> > >
> > > This is a very good question. In diffusion models, reducing the number of sampling steps by half typically leads to lower sample quality (e.g., reduced sharpness and detail). However, our results show that the alignment and semantic-preservation measurements used in the different stages of ASTD remain robust even under this reduced-quality regime for the concepts in our benchmark. It would be interesting in future work to investigate how far this robustness extends and whether even fewer steps could be used. Our hypothesis is that this should be feasible—particularly for low-freqeuncy like concepts rather than high-frequency ones. We will add a brief remark about this interesting point in the paper.
> > >
> > > > Does ASTD implicitly help the denoising process follow a more "optimal path" (e.g., closer to the geodesic on the data manifold) toward the target latent?
> > >
> > > In some ways, yes. ASTD effectively finds the scales a slider should use to follow “optimal” paths that maximize conceptual alignment without distorting the original semantics. While we do not claim a formal theoretical guarantee beyond the geodesic intuition, we share the same view, that the metrics used in ASTD behave like geodesic distances in a “semantic space,” making the resulting scales more linear and straight to human perception in the final sampling outcome. We will also add this into the textual explanation of the illustration (see below)
> > >
> > > > Suggestion: While not strictly required, providing a visualization of the inference trajectory (ASTD vs. Baseline) or a brief theoretical intuition in the final version would significantly strengthen the paper.
> > >
> > > Thank you for this great suggestion — we will prepare an illustration that conveys the intuition behind ASTD and include it in the updated version of the paper.
> > >
> > >
> > > #### Caching Strategy:
> > > > Since ASTD involves additional computation, have the authors considered caching strategies? For instance, techniques similar to [1] could potentially be adapted for the concept slider components to accelerate the process.
> > >
> > > We have not explored this yet, but it seems like a very promising direction for significantly improving ASTD’s performance. We will add a short discussion of this possible extension in the paper’s discussion section, and we also plan to further investigate its potential. Thank you again for the excellent comment.
> > >
> > >
> > > #### Distribution of Scales
> > > > You stated that "the scales found with the ASTD add-on vary per concept and per method." It would be very insightful to see the actual empirical distribution of the found values. If the optimal ranges differ significantly across models (e.g., -prediction vs. v-prediction vs. Rectified Flow), it would empirically prove why the fixed baseline linspace(-3, 3) fails. I suggest adding this analysis to the Appendix.
> > >
> > > Thanks for this great suggestion. We are working on visualizing the distribution of scales to see whether we can identify interesting trends across different models, modalities, and methods.
> > >
> > >
> > > ###  Rare or Complex Concepts (Composition)
> > >
> > > > I appreciate the additional experiments in the supplementary material. I strongly suggest moving these results to the main paper or Appendix in the final version.
> > >
> > > Thanks for the recognition. In the new revision (prepared following this discussion), we have moved these examples to the appendix and added a reference to them from the main paper.
> > >
> > >
> > > > The current method sums the scores linearly (as shown in the provided equation). However, if the concept directions overlap, simple summation might lead to interference. Have the authors considered orthogonal projection or spherical interpolation (SLERP) instead of linear combination?
> > >
> > > These are very interesting suggestions, we also were curious to explore the impact of different composition, therefore, we replaced the simple linear combintation with SLERP as suggested, we quantitatively compare the two options in a new experiment in the appendix. Briefly, we did not observe differences, however, we also think that exploring interference should use quantifications metrics, similarly in the disentanglement literature. Nevertheless, this is a huge subject, and we keep it for future works. Thank you once again for deepening the dicussion regarding these intereseting subjects.

---

> > > > ### Author Response · Authors · 2025-11-27
> > > >
> > > > ### Applicability to Flow Matching (SD3, LTX-Video)
> > > > > Suggestion: Although the authors explained this in the rebuttal, explicitly deriving the connection between the general score function formulation and the velocity prediction formulation in the Appendix would be very helpful for readers. It clarifies that your method is not limited to noise prediction but applicable to any vector-field-based generative model.
> > > >
> > > > Following your suggestion, we have added a discussion in the appendix and a reference from the main page. Thank you for this suggestion which is improving significantly the paper contribution and clarifty.
> > > >
> > > >
> > > > ### Rationale for Overall Score (OS)
> > > >
> > > > Thank you for deepening the discussion around the OS. We share the same intuitions and appreciate your remarks. We agree that better balancing the magnitudes of the terms could further improve the overall score. At present, there is indeed some scale mismatch, which creates a slight bias toward the (CR) term. Although this mismatch is not extremely dominant under our current benchmark and concept set, it could lead to edge cases that might be mitigated through appropriate normalization or alternative formulations, as you suggest.
> > > >
> > > > It is also important to explore ways to weigh the metrics according to user preferences, which could be especially valuable for the creative community. As you noted, substantially revising this aspect is difficult within the short rebuttal timeline, but we take this point seriously and will work toward an improved version toward the final revision.
> > > >
> > > > ---
> > > > Thank you once again for this constructive discussion. We are more than happy to continue deepening the conversation if any remaining concerns or questions arise.

---

> > > ### Author Response · Authors · 2025-12-03
> > >
> > > Dear EsD1,
> > > As a follow-up to our previous comment regarding the distribution of scales in ASTD2x, we have added an analysis of the empirical distributions of the optimal scales in the supplementary files (optimal_scale_ranges.pdf). It includes scatter plots, violin plots, and box plots across Audio, Image, and Video modalities, comparing our method (with ASTD) to CS. These visualizations demonstrate how the optimal ranges vary across concepts and methods, supporting why a fixed baseline (e.g., `linspace(-3,3)`) is insufficient.
> > > > Distribution of Scales: You stated that "the scales found with the ASTD add-on vary per concept and per method." It would be very insightful to see the actual empirical distribution of the found values. If the optimal ranges differ significantly across models (e.g., -prediction vs. v-prediction vs. Rectified Flow), it would empirically prove why the fixed baseline linspace(-3, 3) fails. I suggest adding this analysis to the Appendix.

---

### Official Review · Reviewer_SUV9 · 2025-11-01

**Soundness:** 2
**Presentation:** 3
**Contribution:** 2
**Rating:** 4
**Confidence:** 4

**Summary:**

The paper proposes a training-free, modality-agnostic variant of Concept Sliders for fine-grained control in diffusion models. Instead of learning LoRA adapters per concept, the method computes the CS update at inference by doing three forward passes after an initial neutral phase, enabling plug-and-play control across images, video, and audio. The authors also extend Concept Slider benchmarking beyond images to a multi-modal suite, and propose ASTD to locate saturation points and reparameterize traversal for smoother, perceptually uniform edits. Experiments show competitive or better overall scores versus baselines, with modest extra inference cost and no training.

**Strengths:**

1. The idea is simple and straightforward. The inference-time estimation of the CS update removes per-concept learning and is architecture- and modality-agnostic; it just needs extra forward passes. This makes the approach useful as models evolve.

2. The authors extend the benchmark to video and audio, arguably the first unified multi-modal suite for fine-grained concept control.

**Weaknesses:**

1. While the author claims a training-free approach for concept slider and showcase applications across different modalities on video and audio, such formulation is actually not new. A related paper [1] also proposes a training-free approach to scale concepts by adjusting the classifier-free guidance during test-time. [1] is also model agnostic as it targets the general noise prediction of diffusion models. Moreover, scaling the loudness of audio and other applications has already been explored by [1]. This paper hasn't incorporated any discussion or comparison experiments on [1], which should be justified in the rebuttal. Otherwise, the contribution wouldn't be as significant as claimed by the authors.

[1] Scaling Concept With Text-Guided Diffusion Models

2. CR and CSM rely on CLIP/ViCLIP/CLAP; metric validity inherits biases/sensitivity of these models. Limited discussion on robustness to the choice of aligner. This is raised briefly in the discussion but not systematically evaluated.

3. For concepts like age, makeup, or waviness/loudness, perceptual smoothness and range are ultimately human-perceived. The paper lacks a human study to validate that lower CSM or higher CR correspond to better subjective control.

**Questions:**

1. How does the choice of intervention step k and CFG scale affect range, smoothness, and preservation? Any simple heuristics for picking k per model/modality?

2. Have you tried region- or attribute-specific preservation instead of pure global preservation?

3. When composing multiple sliders, how do you prevent vector interference?

---

> ### Author Response · Authors · 2025-11-23
>
> Thank you very much for the thoughtful and constructive review and comments. They helped us sharpen many aspects of the work and significantly improve the paper’s quality and clarity. We also appreciate your acknowledgement of the usefulness and effectiveness of our approach, as well as our contributions to benchmarking. Below, we address all concerns and questions raised, and we hope this resolves most of the concerns and gaps you identified. We would be happy to continue the discussion and consider any further suggestions or questions you may have.
>
> ---
>
> > W1 - While the author claims a training-free approach for concept slider and showcase applications across different modalities on video and audio, such formulation is actually not new. A related paper [1] also proposes a training-free approach to scale concepts by adjusting the classifier-free guidance during test-time. [1] is also model agnostic as it targets the general noise prediction of diffusion models. Moreover, scaling the loudness of audio and other applications has already been explored by [1]. This paper hasn't incorporated any discussion or comparison experiments on [1], which should be justified in the rebuttal. Otherwise, the contribution wouldn't be as significant as claimed by the authors.
>
> Thank you very much for bringing this important paper to our attention and helping us improve our work significantly. Following your comment, and to address your concern directly, we performed a quantitative comparison between the methods and added a more thorough discussion of the key differences in the related work section. We also revised the text to better reflect the current state of the field and explicitly reference [1]. These changes have been incorporated into the main paper, positioning our method more accurately and strengthening the evidence of its benefits relative to alternative approaches.
>
> Regarding **quantitative comparison**, we compare our method to [1] on the full image benchmark, reporting Semantic Preservation (SP), Conceptual Range (CR), and Conceptual Smoothness (CSM). The results are summarized below and show a clear quality gap in favor of our method, which outperforms [1] on all metrics, underscoring the effectiveness of our approach.
>
> | Method | SP ↓ | CR ↑ | CSM ↑ |
> |--------|------|------|-------|
> | Ours   | 0.01 | 2.27 | 0.36  |
> | [1]    | 0.15 | 0.40 | 0.31  |
>
> For the **conceptual comparison** between the works, we highlight the main differences, which we also summarize in the revised manuscript:
>
> 1. **Generation vs. editing.**
>    Our method is aligned with Concept Slider (CS) approaches, e.g., [2, 3], which focus on controlling concepts directly during generation from noise rather than on post-hoc image editing. In contrast, [1] is fundamentally an editing method and relies on a memory bank plus DDIM inversion. Out-of-the-box, it does not provide concept control when generating from scratch. To adapt [1] for comparison with our framework, we first generate images and then apply [1] as an editing step, which is less computationally efficient than CS-style approaches.
>
> 2. **Methodological differences.**
>    To reveal concepts of variation, [1] requires building a **bank** of intermediate diffusion states, making the method more complex and memory-intensive. Our approach does not depend on such a bank to discover concept directions. Moreover, [1] is developed specifically for DDIM sampling and is not directly applicable to other samplers without further modification. Conceptually, [1] uncovers directions via this memory bank, whereas we rely on positive/negative contrast, leading to different inference equations and an important conceptual distinction between the two methods.
>
> 3. **Multimodal benchmarking.**
>    We highly appreciate bringing to our attention that [1] also explores more than one modality, and we revised our text to acknowledge this. However, there remain substantial gaps that our work aims to bridge. While [1] shows qualitative applications to audio, it does **not** provide quantitative evaluation for this modality. In our work, we invest significant effort in building quantitative benchmarks, which are crucial for fair comparison and future progress of new methods. Additionally, we extend the benchmarking to **video**, which is not covered in [1].

---

> ### Author Response · Authors · 2025-11-23
>
> In summary, [1] is an important and closely related line of work, and we are grateful for the pointer. Our experiments and analysis indicate that our method achieves stronger quantitative performance, provides a simpler and more general mechanism (without memory banks or sampler-specific constraints), and offers a systematically benchmarked multimodal treatment across images, video, and audio. Conceptually, our approach is designed for direct **generation-time** control (following CS line of works), while [1] is fundamentally an **editing** method, which further differentiates the practical use cases and strengths of the two approaches.
>
> [1] Scaling Concept With Text-Guided Diffusion Models
>
> [2] Concept Sliders: LoRA Adaptors for Precise Control in Diffusion Models
>
> [3]  Prompt sliders for fine-grained control, editing and erasing of concepts in diffusion models
>
> ---
>
> > W2 - CR and CSM rely on CLIP/ViCLIP/CLAP; metric validity inherits biases/sensitivity of these models. Limited discussion on robustness to the choice of aligner. This is raised briefly in the discussion but not systematically evaluated.
>
> Thank you for raising this important point. Following your suggestion, we are now running an experiment to assess robustness to the choice of aligner and hope to have concrete results as soon as possible, ideally before the rebuttal period ends.
>
> ---
>
> > W3 - For concepts like age, makeup, or waviness/loudness, perceptual smoothness and range are ultimately human-perceived. The paper lacks a human study to validate that lower CSM or higher CR correspond to better subjective control.
>
> Thank you very much for raising these important points. Following your comment, we conducted a **Metric Alignment with Human Perception** experiment, where we study how our metrics correlate with human judgments of range and smoothness. We have added this experiment to the revised version and briefly summarize it below.
>
> In this experiment, we evaluate the two new metrics in our benchmark. Subjects were shown pairs of sliders (collected from multiple methods) and asked, for each pair: (1) which slider has a better **range**, and (2) which has better **smoothness**. We collected 170 responses over 10 concepts from 17 subjects. We then counted how often the human choice matched the metric’s preferred slider (taking the maximum-scoring option—for example, if the metric scores option (a) as 0.4 and option (b) as 0.6, we treat (b) as the metric’s choice). For range, there is **85%** agreement, and for smoothness **70%** agreement, providing additional evidence that our metrics meaningfully reflect human perceptual preferences.
>
> ---
>
>
> > Q1 - How does the choice of intervention step k and CFG scale affect range, smoothness, and preservation? Any simple heuristics for picking k per model/modality?
>
>
> That is an excellent question. In our experiments, we indeed observed some heuristics relating the choice of (k) (the number of timesteps where the concept slider is active) to the trade-off between semantic preservation, conceptual range, and smoothness.
>
> When **reducing** the size of (k), semantic preservation degrades while conceptual range and smoothness improve. Intuitively, applying the concept direction only for a *small* portion of the trajectory gives the model limited opportunity to “anchor’’ the sample to the original semantic identity. At the same time, concept shift is being amplified, probably since its has more steps to affect the trajectory.
>
> In contrast, when **increasing** the size of (k), we observe the opposite behavior: preservation improves while range and smootheness degrade. This aligns with the interpretation that applying the concept direction over *many* steps repeatedly pulls the sample back toward the base manifold, suppressing large conceptual deviations but retaining more of the original semantics.
>
> For all models and concepts, we chose a **single default (k)** to maintain fairness and consistency. Across modalities, we selected values proportional to the total number of inference steps of the backbone; heuristically, we found that (k $\approx$ 30%) of the full trajectory offers a strong balance between preservation and expressive range across models.
>
> Regarding **CFG**, our side experiments show that excessively high CFG values cause the base prompt to dominate, limiting fine-grained control, whereas too low CFG weakens semantic anchoring and leads to drift unrelated to the base prompt. We used the **default CFG** across all models and did not change it during evaluation.
>
> We believe that exploring an *adaptive* or concept-dependent choice of (k), is a promising direction for future work. Following this question, we have added the above discussion to the main paper.

---

> ### Author Response · Authors · 2025-11-23
>
> > Q2 - Have you tried region- or attribute-specific preservation instead of pure global preservation?
>
>
> Thank you for this question. As discussed in W1 above, region-based editing is currently outside the scope of CS methods, including ours, which are designed for controlling attributes directly during generation from noise rather than editing existing images. Nevertheless, we agree this is an important direction for future work, and we will include it in the final discussion as a key next step for CS methods in general and for our approach in particular.
>
> ---
>
> > Q3 - When composing multiple sliders, how do you prevent vector interference?
>
> That is a very good question. First, it is important to note that composing multiple concepts in CS is additive. The formulation for combining several concepts, as outlined in [1], is:
>
> $\nabla_{x_t} \log p(x_t \mid c_{\text{base}}) + \eta \sum_{c \in \mathcal{C}} \big(\nabla_{x_t} \log p(x_t \mid c_+) - \nabla_{x_t} \log p(x_t \mid c_-) \big)$
>
> Intuitively, each term in the sum defines a “concept direction’’ (each element of $(\mathcal{C})$ is a base/positive/negative tuple) that pushes the sample toward $(c_+)$ and away from $(c_-)$. Thus, composing multiple concepts amounts to adding their directions in score space, making control along each axis potentially independent.
>
> Nevertheless, all CS-style methods are ultimately subject to the entanglement structure of the backbone diffusion model. For example, if changing “age’’ also shifts “gender’’, or altering “hair color’’ also changes “gender’’, the corresponding attribute vectors may interfere. Conversely, if the backbone is inherently disentangled, such interference should not occur, consistent with the formulation above. In our experiments, we did not observe clear cases of such interference, but we agree this is an important avenue for future work. We have added this discussion to the paper in the relevant section, and we also mention it in the conclusion as a promising direction to explore. In addition, we include a **matrix of concept combinations**, which show two sliders grid in the new supplemntary materials in the file `compositions.pdf`.
>
> [1] Concept Sliders: LoRA Adaptors for Precise Control in Diffusion Models – R. Gandikota, J. Materzyńska, T. Zhou, A. Torralba, D. Bau.

---

> > ### Author Response · Authors · 2025-11-27
> >
> > Dear SUV9 and reviewers,
> > Following W2, thank you for pointing out the limitations of CLIP/ViCLIP-based evaluation and their sensitivity to prompts and fine-grained visual details. To address this concern, we conducted an additional study.
> >
> > >  W2 - CR and CSM rely on CLIP/ViCLIP/CLAP; metric validity inherits biases/sensitivity of these models. Limited discussion on robustness to the choice of aligner. This is raised briefly in the discussion but not systematically evaluated.
> >
> > This study spans across 10 concepts, using images from 30 sliders per concept. We replaced CLIP with BLIP as the alignment metric and computed their correlation. We found that there is an overall positive correlation between CLIP and BLIP scores. Approximately $r ≈ 0.65, p ≈ 0.000$, where $r$ represents the Pearson correlation coefficient (how much of the variance in one variable is explained by the other) and $p$ represents the statistical significance (the probability that the observed correlation occurred by chance). This suggests that despite the known biases, the overall trends captured by CLIP remain stable across different aligners. We attached a visual representation of this study in the supplementary material (correlation_blip_clip.pdf) as well as Section C.7 in the Appendix in the latest paper revision.
> > Thank you again for raising this important point, your feedback helped strengthen our evaluation and clarify the robustness considerations of our metric.

---

### Official Review · Reviewer_e12K · 2025-11-01

**Soundness:** 2
**Presentation:** 3
**Contribution:** 2
**Rating:** 4
**Confidence:** 3

**Summary:**

The original Concept Sliders are only applicable to images and require training LoRA models to learn semantic directions. In this work, the authors propose "Training-Free Modality-Agnostic Concept Sliders", a novel approach that directly computes concept directions during the inference phase by leveraging the noise sequences and conditional prompts generated by the base diffusion model, thereby eliminating the need for training. Furthermore, the authors extend this method to multi-modal tasks such as video and audio. The paper also introduces a unified evaluation metric, a reparameterization strategy, and empirically validates the effectiveness and composability of the method across different modalities.

**Strengths:**

1. The authors propose a method that eliminates the need to retrain a LoRA model for each individual concept, enabling unified control across images, videos, and audio. This addresses the limitations of traditional concept sliders, which require separate training for each concept and cannot be quickly adapted to new modalities.
2. To assess fine-grained concept control, the authors extend the original benchmark to include video and audio tasks. They also introduce three modality-agnostic evaluation criteria, facilitating fair comparisons in future research.
3. The experimental setup is thorough, and the authors have considered the problem from multiple angles, enhancing the credibility of the results.
4. The paper is well-written and easy to understand.

**Weaknesses:**

1. Although the proposed method no longer requires training a LoRA model for each concept, constructing concept directions still involves multiple noise predictions by the base model. This leads to significant computational overhead, especially in video and audio tasks, where a full diffusion network must be executed at each sampling step.
2. The dataset remains relatively limited, and the concept control tasks primarily focus on simple attributes. It remains to be validated through further experiments whether the concept sliders can maintain stability when dealing with complex actions or semantic variations.
3. The ASTD method relies heavily on saturated point detection and traversal-based reparameterization, which appear to be based on empirical heuristics. This may limit its generalizability to different models or concepts.

**Questions:**

1. The paper assumes that the noise difference between positive and negative concepts can effectively represent the concept direction, and that direct sampling and composition without additional training is sufficient. Is there any relevant mathematical theory to support the validity of this assumption?
2. Can you experimentally demonstrate the method’s performance in scenarios involving multiple overlapping concepts? For instance, when simultaneously controlling attributes such as age, expression, and background, would there be potential interference among these factors?
3. In terms of quantitative evaluation, could you supplement the results with scores generated by Vision-Language Models (e.g., Gemini or ChatGPT) to further validate the effectiveness of the proposed method?
4. Is the method applicable and effective when the base model is a diffusion model based on a DiT architecture (e.g., FLUX or Stable Diffusion 3)?

---

> ### Author Response · Authors · 2025-11-23
>
> We thank the reviewer for the thoughtful and constructive feedback, and for recognizing the simplicity and effectiveness of our approach, the novelty and importance of our metrics, the insightfulness of the analysis, and the comprehensiveness of our experiments. Below, we address all raised concerns and points, and we would be happy to continue the discussion if needed.
>
> ---
>
> > W1 - Although the proposed method no longer requires training a LoRA model for each concept, constructing concept directions still involves multiple noise predictions by the base model. This leads to significant computational overhead, especially in video and audio tasks, where a full diffusion network must be executed at each sampling step.
>
> Thank you for raising this important point, allowing us to clarify this subject. We would be happy to clarify a few points and offer a slightly different perspective on *computational overhead*.
>
> **Asymptotic complexity and constant factor.** It is important to note that, asymptotically, our computational overhead matches that of a trained Concept Slider (CS). That said, we agree that constant-factor differences matter in practice. In our setup, the additional cost arises from three extra network evaluations used in a cleaning step; however, they are invoked only after (k) steps (here (k=15) out of 50 steps in video/audio generation). Consequently, roughly the first 30% of the trajectory has essentially the same complexity as CS, mitigating part of the constant gap. Additionally, as reported in Table 2 of the main paper, our approach uses comparable memory and incurs only a factor of 1.4 more compute due to these extra evaluations. What constitutes a “big” gap can be subjective and dependent on the user’s system; however, we believe this trade-off can be very useful for many users in practice.
>
> **Optimization of the process.** Importantly, standard engineering optimizations—such as batching evaluations and/or parallelizing them across GPUs, which are widely used—can help close this gap (aside from hardware communication overhead), making the approach promising for scaling on modern systems. In this work we focus on the development of the method and leave such optimization directions for future work. Another relevant point is the progress in sampling techniques (e.g., [1, 2]) that reduce the compute overhead of diffusion/flow sampling; since our method is agnostic to the sampling procedure, these advances can further mitigate our computational overhead going forward.
>
> **In summary**, we agree the method is not “free” computationally; rather, we explicitly trade a modest constant-factor inference overhead for removing per-concept training and architectural dependencies, a particularly attractive trade-off when dealing with variaty of concepts, multiple modalities, or rapidly evolving backbones. We hope this clarifies our perspective and casts the computational overhead in a different light, and we are happy to add this clarification (or further details/suggestion) to the main paper to alleviate this concern following this discussion.
>
>
> [1] One step diffusion via shortcut models - K Frans, D Hafner, S Levine, P Abbeel
> [2] Inductive moment matching - L Zhou, S Ermon, J Song

---

> > ### Author Response · Authors · 2025-11-23
> >
> > > W2 - The dataset remains relatively limited, and the concept control tasks primarily focus on simple attributes. It remains to be validated through further experiments whether the concept sliders can maintain stability when dealing with complex actions or semantic variations.
> >
> > Thank you for raising this important concern. To address it directly, and following your suggestion, we have added a substantial number of more complex concepts to the final benchmark: 5 new concepts for video and 8 for images. The added concepts include:
> >
> > * **Video:** landscape flat vs. rolling terrain, river flow intensity, number of birds in a sky scene, meadow variation in a field, and car traffic speed.
> > * **Images:** shore wave intensity, mountain snow coverage, forest season, crowd density on a beach, city vs. park background in a portrait, dog fur color, eye color, and hairstyle.
> >
> > This expansion makes the benchmarks significantly more diverse and challenging for both modalities. Additionally, we provide visual examples of these new concepts in the supplementary material (`additional_concepts.pdf`) for illustration.
> >
> > Regardless of the above, we would like to emphasize that, alongside simple attributes, there are also genuinely **complex** attributes without the new extentions. For instance, as shown on our anonymized website ([https://anonymous.4open.science/w/FreeSliders-4CD3/](https://anonymous.4open.science/w/FreeSliders-4CD3/)) and in the paper, our “car condition” slider spans from a brand new car to a wrecked car, requiring coordinated changes such as color adjustments, geometric deformation, window integrity, and more. Similarly in audio, we demonstrate controlling the pitch of a choir singing a classical piece and the “storminess” of ocean sounds - both of which go well beyond simple loudness changes.
> >
> > ---
> >
> >
> > > W3 - The ASTD method relies heavily on saturated point detection and traversal-based reparameterization, which appear to be based on empirical heuristics. This may limit its generalizability to different models or concepts.
> >
> > Thank you for the opportunity to discuss this important subject. Indeed, ASTD is motivated by empirical observations and experiments. However, regarding its generalizability, we would like to refer to Tab. 3 in the main paper, which tests this exact question: we evaluate across **three** modalities (each with its own model architecture), **ten** concepts per modality, and **two** slider methods (ours and CS), to create a robust testing environment. As shown in this experiment, both methods consistently improve performance across all modalities and concepts.
> >
> > Additionally, to further strengthen the empirical evidence, and following your comment, we conducted an experiment with Stable Diffusion 3 (SD3), evaluating our method with and without the ASTD stage. The results are presented in the table below:
> >
> > |              | Conceptual Range ↑ | Conceptual Smoothness ↓ | Semantic Preservation ↓ | Overall Score ↑ |
> > | ------------ | ------------------ | ----------------------- | ----------------------- | --------------- |
> > | **w/ ASTD**  | **2.13**           | **0.27**                | 0.018                   | **1.36**        |
> > | **w/o ASTD** | 1.33               | 0.29                    | **0.017**               | 0.60            |
> >
> > We observe a relative improvement in both conceptual range and smoothness. Semantic preservation remains very similar in both cases, while the overall score with ASTD is more than **2×** higher. These results further support the empirical robustness of ASTD across methods, concepts, and generative backbones.
> >
> > In summary, we view this breadth and consistency as strong empirical evidence of generalizability. That said, we agree that formal theoretical guarantees are important, and we will add a note emphasizing this as a key direction for future work.

---

> ### Author Response · Authors · 2025-11-23
>
> > Q1 - The paper assumes that the noise difference between positive and negative concepts can effectively represent the concept direction, and that direct sampling and composition without additional training is sufficient. Is there any relevant mathematical theory to support the validity of this assumption?
>
> Thank you for raising this important question, shortly, the answer is **yes**. Diving into details, the assumption that the *difference* between noise predictions encodes a concept direction is exactly the modeling assumption underlying **Concept Sliders** [1] themselves.
>
> In CS , the target score for a concept slider is
> $$ \nabla_{x_t} \log p(x_t \mid c_{\text{base}}) + \varepsilon \big(\nabla_{x_t} \log p(x_t \mid c_+) - \nabla_{x_t} \log p(x_t \mid c_-)\big),
> $$
> and LoRA parameters are trained to approximate this *ideal* update. Our method simply **computes this expression directly at inference time** with the frozen model, instead of training an adapter to approximate it. Thus, we introduce *no new* assumption; we use the same quantity that CS aims to learn. From a probabilistic perspective,
>   $$
>   \nabla_{x_t} \log p(x_t \mid c_+) - \nabla_{x_t} \log p(x_t \mid c_-)
>   = \nabla_{x_t} \log \frac{p(x_t \mid c_+)}{p(x_t \mid c_-)},
>   $$
>   i.e., it is the gradient of the log-likelihood ratio between the two conditional distributions. This is the **steepest local direction that increases compatibility with ($c_+$) while decreasing compatibility with ($c_-$)**, which matches the intuitive notion of a semantic “direction” between opposing concepts. Similar contrastive score differences have been exploited in classifier guidance, classifier-free guidance, and recent steering methods such as NoiseCLR and related works we cite in Sec. 2.2.
>
>
> Regarding decomposition, this formulation makes the composition of several concepts natural, as also outlined in [1]:
>
> $\nabla_{x_t} \log p(x_t \mid c_{\text{base}}) + \eta \sum_{c \in \mathcal{C}} \big(\nabla_{x_t} \log p(x_t \mid c_+)  - \nabla_{x_t} \log p(x_t \mid c_-) \big)$
>
>
> Intuitively, each term in the sum defines a “concept direction’’ (each element of $\mathcal{C}$ is a tuple of base, positive, and negative concepts) that pushes the sample toward $c_+$ and away from $c_-$, so composing multiple concepts amounts to adding their directions in score space.
>
>
> We have added a short paragraph making this log-likelihood ratio interpretation explicit in the text to clarify the theoretical basis.
>
> [1] Concept Sliders: LoRA Adaptors for Precise Control in Diffusion Models - R Gandikota, J Materzyńska, T Zhou, A Torralba, D Bau
>
> ---
>
> > Q2 - Can you experimentally demonstrate the method’s performance in scenarios involving multiple overlapping concepts? For instance, when simultaneously controlling attributes such as age, expression, and background, would there be potential interference among these factors?
>
> Thank you for this comment. Following your suggestion, we added a new experiment in the supplementary material, `compositions.pdf`, where we present a matrix of edits combining two concepts.
>
> Regarding interference, we did not empirically observe such phenomena. Nevertheless, all CS-style methods are subject to the underlying entanglement of the backbone diffusion model. For example, if changing “age” also changes “gender,” or altering “hair color” also shifts “gender,” the corresponding attribute vectors may interfere; conversely, if the backbone is inherently disentangled, such interference should not occur. In our experiments, we did not find clear cases of interference, but we agree this is an important avenue for future work. We have added this discussion in the relevant section of the paper to highlight potential issues, and we also mention it in the conclusion as an interesting direction to explore. Importantly, if the backbone already provides disentangled concepts, our approach should not introduce additional interference.
>
> ---
>
> > Q3 - In terms of quantitative evaluation, could you supplement the results with scores generated by Vision-Language Models (e.g., Gemini or ChatGPT) to further validate the effectiveness of the proposed method?
>
> Thank you very much for this interesting suggestion. We are currently exploring it, and hope to have a concrete conclusion as soon as possible, ideally before the rebuttal period ends.

---

> > ### Author Response · Authors · 2025-11-23
> >
> > > Q4 - Is the method applicable and effective when the base model is a diffusion model based on a DiT architecture (e.g., FLUX or Stable Diffusion 3)?
> >
> > Thank you for the important question. In short, **yes** our method is agnostic to the backbone architecture (e.g., U-Net or DiT or other) **and** to the denoising paradigm (diffusion or flow-matching or other). In Section 6.2, we demonstrate seamless adaptation to two DiT-based video generators, **LTX-Video** and **CogVideoX**, and show that our approach works across both diffusion (CogVideoX) and flow-matching (LTX-Video) based training regimes. Following your question, we added a clarifying note to emphasize this architecture- and paradigm-agnostic design, which improves the presentation of our approach.
> >
> > Furthermore, we have also added SD3 to our benchmark, strengthening the evidence for the applicability and effectiveness of our approach across diverse generative paradigms. Please see the previous discussion for the quantitative results, and refer to the supplementary file *`stable_diffusion_3.pdf`* for qualitative generations with this model.

---

> > > ### Author Response · Authors · 2025-11-27
> > >
> > > Dear e12K and reviewers,
> > > Following Q3 we conducted an experiment to evaluate whether large vision-language models (VLMs) can make slider-quality judgments that align with human perception. Since time is short, these results are preliminary, and we emphasize that this direction deserves careful, dedicated investigation from multiple perspectives. Nevertheless, the experiment below sheds some initial light on the promise of this line of work. Briefly, the results suggest that a VLM makes decisions similar to humans, which could be very useful for future evaluation techniques.
> > >
> > > The experiment is as follows. We start from taking the results from the human preference study made earlier: for each pairwise comparison between two sliders, we take the **majority vote** as the “ground-truth” decision (which slider is preferred under the given concept and the definition of a concept slider). To reduce noise, we only keep comparisons where the majority share exceeds 60%. We then present the same slider pairs to ChatGPT 5.1, using essentially the same instructions given to the human evaluators, and measure how often the model’s choice matches the human majority. Each comparison is done independently. With 25 such comparisons (random baseline = 50%), we observe **81% agreement** between ChatGPT 5.1 and the human majority vote. This indicates that VLMs can make judgments that are reasonably consistent with human preferences in this setting.
> > >
> > > While more thorough experiments are needed, these findings suggest that VLM-based evaluation is a promising direction—we think especially if they will be combined with alignment techniques such as RLHF or DPO to further tune the judge to human preferences. We will include this experiment in the final revision and add a short discussion of this direction in the paper. Thank you again for this important and insightful question.

---

### Author Response · Authors · 2025-11-23
**General Comment**

We thank all reviewers for their thoughtful and constructive feedback, which has significantly improved the paper’s clarity and quality. In each rebuttal box, we respond point-by-point to all concerns and questions raised, and we are happy to continue the discussion to further refine the work. For your convenience, we have uploaded a revised version with changes marked in red and briefly summarized below. Due to time constraints, some items are still in progress, we list them as well, and hopefully, they will be completed and reported later in this rebuttal.

### List of changes


- We added a new experiment on the Stable Diffusion 3 model, showcasing qualitative examples (in `stable_diffusion_3.pdf` in the new supplementary), along with quantitative evaluation and an analysis of ASTD’s effectiveness.


- We added several more diverse and complex concepts to the benchmark - 5 for video and 8 for images. Illustrations of these new concepts are provided in the new supplementary material, in the file `additional_concepts.pdf`.


- We added to the paper a comparison with a new, related method. First, in the related work section, we provide a thorough conceptual comparison that clearly differentiates between the approaches. Second, in the experiments section, we present a quantitative comparison, demonstrating the superiority of our method.

- We show that ASTD can be accelerated to save almost 50%  of the computational cost while maintaining the same performance gains (ASTD2x). We added this result to the main paper and clarified that this stage can be viewed as a standalone component.

- We expanded the discussion in Section 6.2 on inference performance, emphasizing both the asymptotic equivalence between the approaches and the constant-factor differences in runtime.

- We expanded the final discussion to briefly highlight the importance of providing theoretical guarantees for ASTD and identified this as an interesting direction for future work.

- We clarified in the experiments section that our benchmark includes models with multiple generative denoising paradigms (flow matching vs. diffusion-based) and different backbones (U-Net vs. Transformers).

- We added theoretical background explaining the motivation for Concept Sliders theory in the appendix and referenced it from the main text.

- We added a discussion of interference between multiple concepts in the compositional experiments section. Additionally, we include in the new supplementary materials (`composition.pdf`) and in Section C a compositional grid illustrating concept combinations.

- We added a discussion regarding the choice of (k), highlight it as a promising direction for future work, and additionally provide visual illustrations of these heuristics in the new supplementary file (`k_comparisons.pdf`).

- We changed the middle example in Figure 2 with one illustrating the concept "Makeup" rather than "Glasses".

- In Section B, we clarified the scheduler types employed and detailed the distinctions between using SD 1.4 and SD 1.5 as backbones.

- We explore whether different alignment backbones produce different scoring distributions, and represent the results in the new supplementary file (correlation_blip_clip.pdf) and in Section C.

- We explore whether large multimodal models can be used to score the sliders.

- We add an analysis of the optimal scales found with ASTD for different concepts, modalities and methods in the new supplementary file (optimal_scale_ranges.pdf).

---

### Author Response · Authors · 2025-12-03

Dear AC, as the rebuttal concludes, we would like to thank all the reviewers for their valuable efforts and contributions throughout this process. During the rebuttal, we worked to address all comments and concerns raised. While some discussions were fully resolved and others remain open due to the information breach and the resulting changes, we believe we were able to address the majority of the issues brought up, significantly improving the quality and clarity of the paper. In the general comment submitted earlier, we summarized all changes made in response to the reviewers' feedback. Thank you.

---

### Meta-Review · Area_Chair_sgCf · 2026-01-06

**Summary:**

The paper proposes a training-free, modality-agnostic extension of Concept Sliders, enabling fine-grained semantic control in diffusion models across images, video, and audio without per-concept training. It introduces new evaluation metrics (Conceptual Range, Semantic Preservation, Conceptual Smoothness) and a two-stage optimization method (ASTD) for non-linear traversal. Reviewers acknowledged the method's simplicity, multimodal applicability, and novel benchmarking, but expressed concerns about high inference cost, limited conceptual scope, lack of thorough theoretical motivation, and insufficient validation against human perception.

**Reviewer Concerns:**

In the rebuttal, the authors addressed several key reviewer concerns by expanding the benchmark with more complex concepts (e.g., for video and images), adding experiments with Stable Diffusion 3 to demonstrate modality- and architecture-agnostic claims, providing human studies to align proposed metrics with perceptual judgment, comparing their method with a newly cited related work, and clarifying the computational trade-offs and theoretical motivation behind their approach. However, some issues remain unresolved or only partially addressed: significant computational overhead during inference (especially for ASTD) is still a practical limitation, the theoretical grounding for certain heuristics (like saturation point detection in ASTD) remains empirical, and broader validation on highly abstract or rare concepts is still limited. Additionally, concerns about potential interference in multi-concept composition and the robustness of CLIP-based metrics, while discussed, lack comprehensive empirical resolution.

**Reviewer Scores:**

The final ratings might have shifted slightly for Reviewers e12K and SUV9 (both initially rated 4: marginally below threshold) toward a weak accept, given the authors' substantial expansions to the benchmark, the human study aligning metrics with perception, and the competitive quantitative comparison against the newly cited method. However, the ratings would likely not have changed substantially to a clear accept. This is because core, unresolved concerns persisted: the high and arguably impractical inference cost (especially for ASTD), the empirical rather than theoretical basis for key components, and the limited scope of validation on truly complex or abstract concepts. Reviewer 8711 (initial rating 2: reject) might have been slightly swayed by the added human studies and theoretical clarification but would likely maintain a reject due to lingering doubts about the method's perceived triviality and insufficient motivation. Reviewer EsD1 (rating 6: marginally above threshold) provided the most constructive dialogue and maintained their positive rating, acknowledging the improvements while noting remaining open questions for future work. Overall, while the rebuttal improved clarity and addressed specific criticisms, it did not convincingly overcome the fundamental reservations about practicality, theoretical depth, and generalizability that initially placed the paper below the acceptance threshold for a majority of the reviewers.

---

### Decision · Program_Chairs · 2026-01-26

Reject